



# Behavior and Mechanisms of Doppler Wind Lidar Error in Varying Stability Regimes

Rachel Robey[1] and Julie K. Lundquist[2,3]

[1]Department of Applied Mathematics, University of Colorado Boulder, Boulder, Colorado, USA
[2]Department of Atmospheric and Oceanic Sciences, University of Colorado Boulder, Boulder, Colorado, USA
[3]National Renewable Energy Laboratory, Golden, Colorado, USA

**Correspondence:** Rachel Robey (rachel.robey@colorado.edu)

**Abstract.** Wind lidar are widespread and important tools in atmospheric observations. An intrinsic part of lidar measurement error is due to atmospheric variability in the remote sensing scan volume. This study describes and quantifies the distribution of measurement error due to turbulence in varying atmospheric stability. While the lidar error model is general, we demonstrate the approach using large ensembles of virtual WindcubeV2 lidar performing profiling doppler-beam-swinging (DBS) scans in

quasi-stationary large-eddy simulations (LES) of convective and stable boundary layers. Error trends vary with the stability regime, time-averaging of results, and observation height. A systematic analysis of the observation error explains dominant mechanisms and supports the findings of the empirical results. Treating the error under a random variable framework allows for informed predictions about the effect of different configurations or conditions on lidar performance. Convective conditions are most prone to large errors, driven by the large vertical velocities in convective plumes and exacerbated by the high elevation

angle of the scanning beams. The violations of the assumption of horizontal homogeneity due to filtered turbulent velocity variances dominate the error variance, with the vertical velocity variations of particular importance. Range gate weighting contributes little to the variability of the error, but induces an underestimating bias into the horizontal velocity near the surface shear layer. Error in the horizontal wind speed and direction computed from wind components is sensitive to the background wind speed but has negligible dependence on the relative orientation of the instrument. Especially during low winds and in

the presence of large errors in the $u$ and $v$ velocity estimates, the reported wind speed is subject to a systematic positive bias. Vector time-averaged measurements can improve the behavior of the error distribution with a predictable effectiveness related to the number of decorrelated samples in the time window. The approach in decomposing the error mechanisms with the help of the LES flow field extends to more complex measurement scenarios and scans.

## 1  Introduction

Effectively and efficiently collecting observations of atmospheric winds poses an ongoing, multifaceted challenge for the atmospheric science community. Wind-profiling lidar offers a cheaper, more easily deployable, and higher-profiling alternative to traditional meteorological towers while scanning lidar allows for collection of data over broad regions of the atmosphere. Over the last few decades, lidar technology has grown into maturity, with several commercial wind lidar systems becoming available since the late 2000s. Lidar systems are widely employed in scientific studies of atmospheric boundary layer meteorology



(Cheynet et al., 2017; Smith et al., 2019) and in assessments of wind resources (Gryning et al., 2017; Menke et al., 2020), wind turbine wake behavior (Aitken and Lundquist, 2014; Bodini et al., 2017), air quality (Liu et al., 2019), and fire meteorology (Clements et al., 2018). Lidar data, as opposed to 'point'-measurements collected by in-situ instruments like sonic anemometers, offer a more complex, indirect representation of the flow field which must be analyzed critically and in conjunction with an understanding of what is being measured and the extent of its limitations and potential biases.

All wind lidar instruments function on the fundamental basis of sampling the flow along an emitted beam. With a single lidar's beam, only a one-dimensional projection of the velocity can be measured. In light of this sampling limitation, dual- and triple- lidar methods have been explored to allow concurrent measurement of the necessary spanning wind vectors (Newsom et al., 2008; Stawiarski et al., 2013; Choukulkar et al., 2017; Menke et al., 2020). Use of single profiling or scanning lidar remains common and economic, so that quantifying their error behavior remains a high priority. Profiling lidar in particular make

additional assumptions about the flow (usually horizontal homogeneity) to reconstruct an estimate of the three-dimensional winds at various heights from a series of measurements pointing the beam in different directions (Bingöl et al., 2009; Lundquist et al., 2015).

The error of remote sensing instruments like lidar, sodar, and radar depends not just on the system itself but is a statistical distribution arising from the interplay of the system with the turbulent atmospheric flow. Sources of error in profiling lidar

measurement were distinguished by Gottschall and Courtney (2010). Uncertainties in the instrument hardware configuration (e.g. the beam angle) or in the alignment of the lidar on site (e.g. leveling and direction) can introduce error which can roughly be controlled by the calibration accuracy. Additional error is inherent to the measurement system, depending on the atmospheric conditions and character of the flow itself. Measurements of mean horizontal winds in favorable (flat, uniform) conditions have generally performed well in field assessments; the ten-minute averages of the horizontal wind have reported accuracy of 0.1-

0.2 m/s with wind direction within 2° (Lindelöw, 2008; Cariou and Boquet, 2010). Questions about measurements of vertical velocities and the ability of wind profiling lidar to measure turbulence remain areas of active research (Sathe et al., 2011; Sathe and Mann, 2012; Sathe et al., 2015; Newman et al., 2016).

The study of instrument error using numerical large-eddy simulations (LES) was introduced by Muschinski et al. (1999). The simulated flow, in conjunction with radio wave scattering theory, represented the action of a radar wind profiler in a flow field.

Analysis of the virtual instrument data provided valuable insights into field study results concerning vertical-velocity bias and primary sources of signal to noise ratio (SNR). Wainwright et al. (2014) leveraged LES in a similar way with a sodar simulator applied to a convective boundary layer. As wind lidar took off in popularity, interest grew for similar kinds of investigations and the insights the combination of a lidar model with LES could provide into the instruments.

LES enable the generation of realistic turbulent atmospheric flows with which to study likely interactions and resulting error

behavior of remote-sensing instruments. The spatial resolution of LES is typically on the order of 1-10s of meters, and is designed to explicitly capture the most critical length scales in the atmospheric boundary layer while parameterizing the effects of the smallest turbulent scales. The resolution is not sufficient to explicitly compute the underlying optical measurement of scattering; however, the salient effects of volume averaging and reconstruction over the scanning volume occur at a scale which can be supported by the LES data. Compared to field studies of instrument accuracy, studies with virtual instruments in LES





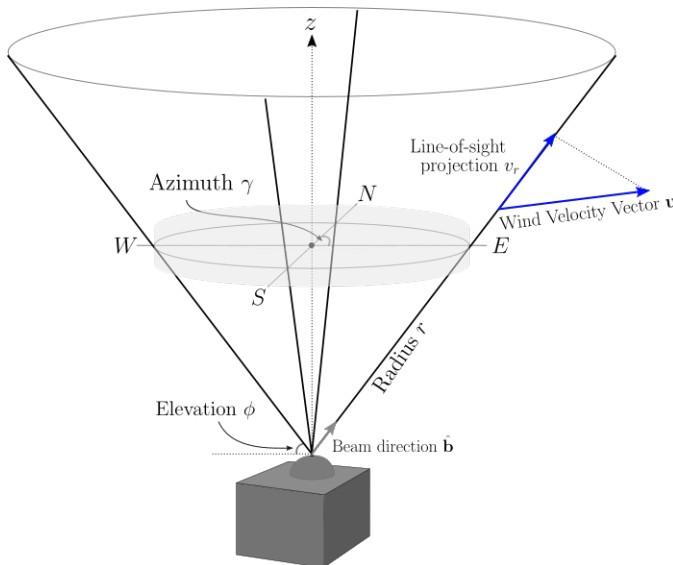

**Figure 1.** Geometry of a Doppler Beam Swinging (DBS) scan performed by a Leosphere WindcubeV2 to estimate a vertical profile of the three-dimensional wind velocity. At a frequency of $1s$, the sampling beam moves through a vertical and four angled positions corresponding to the cardinal directions. Scan ranges from 40-240m above the ground. The light grey cylinder demarcates a reference volume for the scan.

have unencumbered access to full knowledge of the flow field, allow for control over the the case parameters (terrain, forcing, boundaries), and can 'deploy' instruments in ways that are not physically or financially possible in reality (e.g. re-sampling the same flow field or testing many locations in a domain). The comprehensive flow field data also opens up the discussion about the appropriate reference 'truth' for lidar observations. Measurements may be better thought of as representing volume averages, which we cannot directly measure in the field but can compute a reference for in LES.

Earlier virtual lidar studies have generally considered complex lidar behavior and have been built on a range of different LES models. The coordinated use of multiple lidar to simultaneously probe spanning vectors of the wind in a volume was studied by Stawiarski et al. (2013) using the parallelized large-eddy simulation model (PALM) (Maronga et al., 2015). The dependence of profiling lidar on horizontal homogeneity complicates its use in complex terrain; Klaas et al. (2015) investigated observation deviations due to terrain and choice of instrument location. Gasch et al. (2019) implemented an airborne virtual lidar with

PALM and studied errors due to flow inhomogeneities. Wind energy applications have been a notable driver of virtual lidar studies. Simley et al. (2011) modeled scanning continuous wave lidar to optimize their upwind measurements for use in wind turbine control. The measurement of turbine wakes with profiling lidar was explored in Lundquist et al. (2015) (using SOWFA) and Mirocha et al. (2015) (using WRF-LES). Turbine wakes are also considered in Forsting et al. (2017) which focuses on the volume averaging along lidar beams in these high-gradient regions. Only recently has virtual lidar been employed for baseline

studies of profiling wind lidar in favorable (flat, uniform, quasi-stationary) conditions. Rahlves et al. (2021) used virtual lidar





in PALM LES to compare the bulk performance of various scan types (DBS and VAD at varying cone angles) across a suite of convective conditions.

We have developed a virtual lidar model tool in Python to run on output from the Weather Research and Forecasting large-eddy simulation (WRF-LES). WRF-LES boasts a user base of over 48,000 and is attractive for its accessibility as an open-source, documented model. It can be configured for ideal simulations or coupled with mesoscale nesting to simulate case studies of real sites (Mazzaro et al., 2017; Haupt et al., 2019), and offers a range of subfilterscale turbulence models for use in LES (Mirocha et al., 2010; Chow et al., 2005). In validations, WRF-LES has also compared well to observations of boundary layers in varying stabilities (Peña et al., 2021). Though the virtual lidar tool is targeted at WRF-LES, with minor adjustments to accommodate for the different output formats, it could be easily adapted for use with other LES models.

In this first demonstration of the virtual lidar tool, we consider a specific case of the Leosphere WindcubeV2 profiling lidar (Figure 1) measuring mean winds in ideal simulations of stable and convective conditions over flat terrain. As in Rahlves et al. (2021), the configuration allows for a baseline assessment of the lidar performance by omitting external sources of inhomogeneities, like complex terrain or wind turbines, and isolates the system error arising from complex but statistically stationary turbulent boundary layer flow. Depending on the quantity of interest, Rahlves et al. (2021) found that the configuration choices (scan type, cone angle, averaging length) have distinct effects on the lidar retrieval error. Additionally, profiling in strongly convective conditions, absent other sources of inhomogeneity, the lidar exhibited markedly larger errors than in more moderate convection. Our work extends that study for a single DBS profiling scan to include a range gate weighting in the lidar model, a further stable stability regime, and the disaggregation of the vertical profile heights. The idea of using ensembles to gauge the uncertainty of the error (Rahlves et al., 2021) is expanded to using larger ensembles to characterize an error distribution particular to the flow conditions and scan geometry. Further, we present an analytic treatment of the observation system error to explain the dominant error mechanisms and trends, supporting the findings of the empirical results. The framework further enables informed, a priori predictions of how different configurations or conditions might be expected to impact lidar performance, without relying on the full virtual lidar model.

Section 2 presents the lidar model and its configuration to represent the WindcubeV2 lidar. It then describes the WRF-LES cases against which ensembles of the virtual instruments were tested. The resulting observation error distributions are summarized in section 3, focusing on empirical trends across stability conditions, height, and time-averaging. Section 4 analyzes the mechanisms behind the errors and the conditions which drive the trends found by the virtual lidar model. Here we address the influence of the range gate weighting function and analytically represent the impact of violations of the horizontal homogeneity assumption. Further treatment investigates the wind speed and direction measurements derived from the wind components and time-averaged measurements.





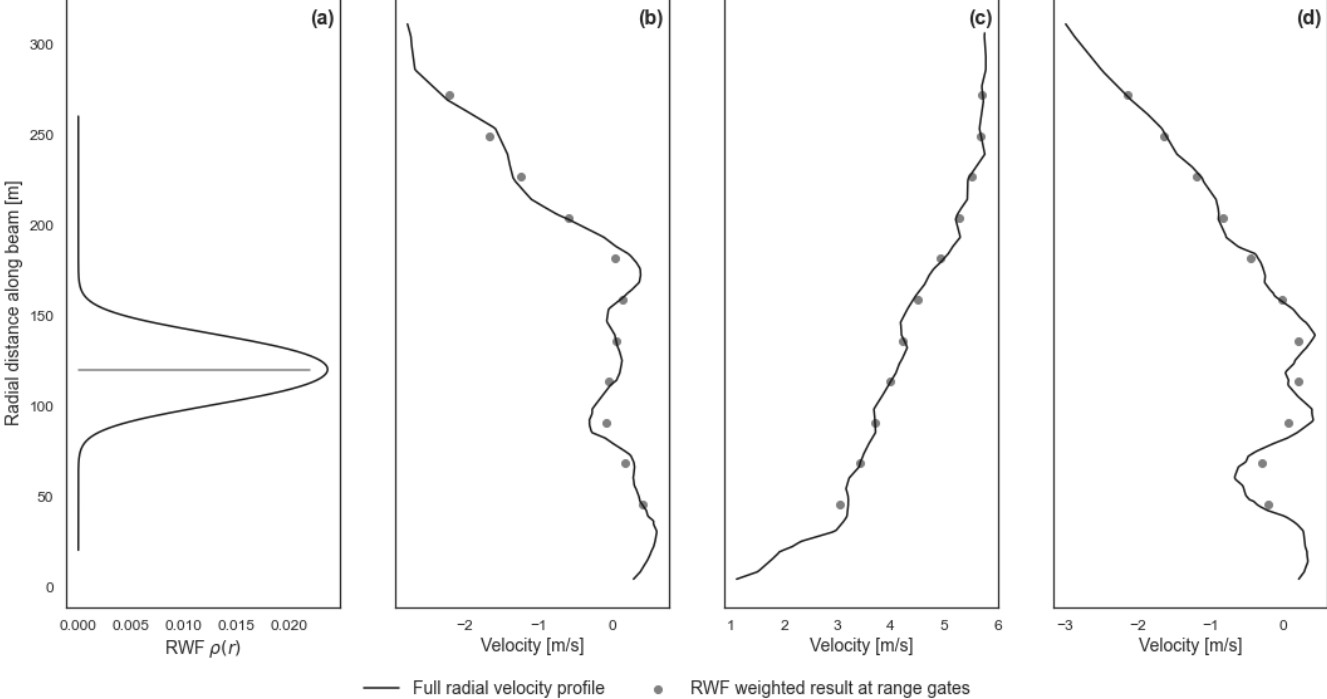

**Figure 2.** (a) Range gate weighting function (RWF) centered at the 120m range gate and examples (b, c, d) of full, unweighted radial velocity profiles along a beam, interpolated from LES winds, compared to the corresponding weighted result at the range gates.

## 2 Data and Methods

### 2.1 Generalized virtual lidar model

The virtual lidar was designed to create a configurable, general model that can be modified to replicate most lidar instruments. The observing system is decomposed into modular components common across lidar systems: the retrieval of radial wind velocities along an individual beam via a range gate weighting function, the scanning pattern the beam moves through, and the internal post-processing of these measurements. The handling of each of the components can be easily modified and new definitions substituted to allow for customization.

This initial study focuses on a common commercial system: the vertical profiling Leosphere WindcubeV2 performing a Doppler-Beam-Swinging (DBS) scan (Figure 1). Its parameters and geometry, summarized in table 2, form the basis of the description of the model stages. Thorough discussions of the Windcube, other wind lidar systems, and the underlying technology can be found in Courtney et al. (2008); Lindelöw (2008); Cariou and Boquet (2010), and Thobois et al. (2015).



### 2.1.1 Sampling along a single lidar beam

The basis of wind lidar is the retrieval of radial (line-of-sight) wind velocities along an emitted laser beam using the backscatter off aerosols entrained in the flow. Doppler lidar diagnose the shift in the frequency of the backscattered light to measure radial wind velocity (equation 1).

$$v_r = \hat{\boldsymbol{b}} \cdot \boldsymbol{u} = u \sin\gamma \cos\phi + v \cos\gamma \cos\phi + w \sin\phi \qquad (1)$$

The radial velocity, $v_r$, is the dot product of the wind velocity vector, $\boldsymbol{u} = (u, v, w)$ with $u$ along the zonal direction, and the beam unit direction vector $\hat{\boldsymbol{b}}$. The beam direction points along azimuthal angle, $\gamma \in [0°, 360°)$ clockwise from north, and elevation angle from the horizon, $\phi \in [0°, 90°]$. With this convention, positive radial velocities move away from the instrument. In the context of our model, we assume 'perfect' conditions in the sense of ignoring factors like aerosol type, size, and density distribution and conditions like humidity, fog, or precipitation that can affect the quality of the return signal in the optical measurement of the radial velocity (Aitken et al., 2012; Boquet et al., 2016; Rösner et al., 2020). We instead focus on the representation of the averaging introduced by the sampling process.

Although the scattering cannot be explicitly resolved on an LES scale, previous studies have found that the full sampling procedure (collection and internal processing of backscattered light) is well-approximated by the application of a range-gate weighting function (RWF) (Frehlich, 1997; Gryning et al., 2017; Simley et al., 2018). Prevalent lidar technologies employ either continuous wave (e.g. ZephIR) or pulsed (e.g. Windcube) lasers to target distances along the beam at which to retrieve velocities. Continuous wave systems set a focal distance to center returns, while pulsed lidar release a rapid sequence of pulses and separate the returns into a series of spatio-temporal 'range gates' along the beam. In both cases, the process acts like a weighted, volume average of radial velocities along the beam about the target distance. The cross-sectional area of the beam is negligibly small compared to the along-beam length scale, so that the averaging may be described by a one-dimensional line integral. The system-observed radial velocity at a distance $r$, $\overline{v}_r(r)$, is given by the convolution of projected wind velocities with the weighting function (equation 2).

$$\overline{v}_r(r) = \int_{-\infty}^{\infty} \rho(s) v_r(r+s) ds = \int_{-\infty}^{\infty} \rho(s) \boldsymbol{b} \cdot \boldsymbol{u}((r+s)\boldsymbol{b}) ds \qquad (2)$$

$\rho(s)$ is the normalized RWF satisfying $\int_{-\infty}^{\infty} \rho(s) ds = 1$, $\boldsymbol{b}$ is the beam direction unit vector, and $\boldsymbol{u}$ the velocity field.

For a pulsed lidar, the weighting function arises from the convolution of the range-gate profile with the beam pulse profile (Frehlich, 1997); equation 3 gives the integral in space.

$$\rho(r) = \int_{-\infty}^{\infty} g(r-s) \chi(s) ds \qquad (3)$$

A top hat, normalized indicator function, $\chi$, represents the time span of the range gate and the pulse shape, $g$, is assumed to be Gaussian (Banakh et al., 1997). In the lidar operation, the parameters for the pulse and range gate are temporal quantities. To transform into their representation for the spatial integral, assume that propagation is at the speed of light and note that the





originating signal must travel to a point in space and back to the instrument receiver to be collected (Lindelöw, 2008; Cariou and Boquet, 2010). The indicator function for the spatial range gate with temporal interval $\tau_m$ is given in equation 4.

$$\chi(s) = \begin{cases} \frac{1}{\Delta p}, & s \in \left[-\frac{\Delta p}{2}, \frac{\Delta p}{2}\right] \\ 0, & \text{else} \end{cases} ; \qquad \Delta p = \frac{c\tau_m}{2} \tag{4}$$

We can also express the spatial Gaussian pulse in terms of the temporal full-width half-maximum (FWHM) parameter, $\tau$ (equation 5).

$$g(s) = \frac{2\sqrt{\ln 2}}{\tau_s \sqrt{\pi}} \exp\left(-4\ln 2 \frac{s^2}{\tau_s^2}\right); \qquad \tau_s = \frac{c\tau}{2} \tag{5}$$

The convolution integral (equation 3) may be solved analytically in this case, yielding the expanded form (equation 6) found in some references (Cariou and Boquet, 2010; Lundquist et al., 2015; Forsting et al., 2017).

$$\rho(r) = \frac{1}{c\tau_m}\left[\text{erf}\left(\frac{4\sqrt{\ln 2}}{c\tau}r + \frac{\tau_m \sqrt{\ln 2}}{\tau}\right) - \text{erf}\left(\frac{4\sqrt{\ln 2}}{c\tau}r - \frac{\tau_m \sqrt{\ln 2}}{\tau}\right)\right] \tag{6}$$

Here, $c$ (0.29979 m/ns) is the speed of light, $\tau_m$ the temporal range gate, and $\tau$ the full width half maximum (FWHM) pulse duration. Other representations of the pulse and range gate (i.e. not top hat and Gaussian) are not necessarily valid under this approximation. Lindelöw (2008), for example, adapted the form to account for a focused beam which scales the RWF by the focusing efficiency. The basic, unadapted form presented here and implemented in our model for this study is also used in
several other virtual lidar models (Stawiarski et al., 2013; Lundquist et al., 2015; Gasch et al., 2019; Forsting et al., 2017).

Using parameters from the WindcubeV2 (table 2), the shape and extent of the RWF (equation 6) can be made concrete (figure 2a). The weights drop to half their peak value about 20 m to either side of the target distance and are non-negligible up to around 40 m. The choices of pulse and range gate parameters in a coherent lidar system must balance the desire for spatial locality (reducing the 'width' of the RWF) and the need for adequate accumulation time ($\tau_m$) to accurately resolve frequencies
used for the radial velocity measurement. The application of the RWF to the radial velocity profile may be thought of as a smoothing/low-pass filter (figure 2(b,c,d)).

The form of the pulsed lidar RWF is distinct from that of continuous wave lidar (Gryning et al., 2017; Forsting et al., 2017). A major difference is the consistency of the pulsed RWF over target distances; the RWFs of continuous wave lidar vary with distance, giving better locality close to the instrument than far away. Closer distances will be sampled at higher resolution by
a CW instrument and further distances by a pulsed lidar.

To compute the RWF-weighted retrieval from an LES flow field, the wind components are first interpolated to points along the beam and the projection onto the beam direction found. The virtual lidar uses a linear barycentric interpolation from a triangulation of the LES grid (i.e. linear interpolation on tetrahedrons using Virtanen et al. (2020), Spe). The numerical approximation of the convolution integral from the interpolated radial velocities treats the continuous weighted average as a
discrete weighted average (equation 7). The form is a slightly modified formulation of that used in Lundquist et al. (2015) and





Forsting et al. (2017) (see appendix E).

$$\overline{v}_r(r_0) = \int_{-\infty}^{\infty} \rho(s) v_r(r+s) ds \approx \int_{-T}^{T} \rho(s) v_r(r+s) ds \approx \sum_k \frac{h_k \rho(s_k)}{\sum_i h_i \rho(s_i)} v_r(r_0 + s_k) \tag{7}$$

Parameterizing along the beam, the $\{s_i\}$ nodes are the points where the winds have been interpolated ($s_i = 0$ is at $r_0$). If the nodes are taken as midpoints of intervals with corresponding lengths $\{h_i\}$ partitioning the integral range $[-T,T]$, then the quadrature formulation is a normalized midpoint rule. The normalization ensures the result is a weighted average (avoiding over- or under-estimation due to the numeric weights not summing to unity). The placement of the nodes $\{s_i\}$ is free to be chosen for convenience or, as recommended by Forsting et al. (2017), to optimize utility of the interpolated points in the convolution so that fewer points need be interpolated. Our current implementation uses equi-spaced points one meter apart along the beam (see appendix E for discussion).

Interpolation dominates the computational work in most virtual lidar models, making it a prime target for performance optimization. Some preliminary efforts have been made in our implementation, e.g. computing and saving interpolation weights to reuse for the different velocity components and for repeating beams on a static grid. Further improvements to the implementation and choice of nodes are possible and will be targeted in subsequent developments.

### 2.1.2 Time-resolved scanning patterns

Based on the type of scan they perform, lidars are categorized as profiling or scanning systems. Profiling lidars are designed to provide a vertical profile of the three-dimensional wind velocity, much as would be reported by a meteorological tower. To reconstruct a three-dimensional wind vector, the instrument needs spanning radial velocity samples from at least three different directions. The scanning head rotates through the necessary positions quickly, which limits the intervening evolution of the wind field. Scanning lidars perceive the atmosphere in a fundamentally different way; they move the beam more slowly through a slice of the atmosphere, resolving the radial velocity of features in the spatial selection. Common configurations include vertical slices (range height indicator or RHI scan) which fix the azimuthal angle while varying elevation; conical slices (vertical azimuthal display or VAD scan) which fix the elevation angle but slowly complete a full rotation, or a partial conical section (plan position indicator or PPI scan), as visualized in Clifton et al. (2015). Any scanning geometry, as described above or a more complex configuration, arises from 'pointing' the beam and is most naturally and compactly represented as a time series of elevation, $\phi$, and azimuthal, $\gamma$, angles in spherical coordinates with the beam source at the origin.

For the purposes of this study, we consider the Doppler-beam-swinging (DBS) profiling scan used by the WindcubeV2, which moves through the four cardinal directions, angled $28°$ from the vertical, before pointing the beam straight vertically (Figure 1). The total scan takes approximately 5 seconds, spending about a second at each of the scan positions (Bodini et al., 2019). The range gates correspond to equi-spaced heights above the ground. As the beam rotates through the scan, radial velocities are measured at the center and four points around the circular perimeter of the scanning cone for each given height. At each second in the post-processing stage, the most recent set of radial velocities is used to reconstruct an estimate of the vertical profile of three-dimensional velocities.





The beam accumulation time for the WindcubeV2 is about a second, while the LES model time steps are on the order of a

tenth of a second. The additional averaging due to the longer accumulation time is ignored in the current version of the virtual

lidar; it handles the scan by performing the beam sampling on snapshots of the flow field output at one second intervals. It

is assumed that in the WindcubeV2 the temporal average is less significant than the spatial averaging; future versions of the

model will account for accumulation times by performing this averaging step explicitly.

### 2.1.3   Internal processing: 3-D velocity reconstruction

In the WindcubeV2, the post-processing stage reconstructs the three-dimensional velocity from the radial velocities collected

across the scan cycle. Under the assumptions of horizontal homogeneity and invariance of the flow field over the scan duration,

the radial velocities collected by each of the beams at a given height are all projections of the same three-dimensional velocity

vector. Omitting the vertical beam, we solve for the vector components at a given range gate height (equation 8) as in, e.g.

Cariou and Boquet (2010).

$$\boldsymbol{u}_{lidar} = \begin{pmatrix} \dfrac{u_E - u_W}{2\cos\phi} \\ \dfrac{u_N - u_S}{2\cos\phi} \\ \dfrac{u_N + u_E + u_S + u_W}{4\sin\phi} \end{pmatrix} \tag{8}$$

At a fixed height, $u_{E,W,N,S}$ are the most recently measured radial velocities ($\overline{v}_r(r)$) from beams pointed in each of the

cardinal directions. The elevation angle of the beams from the horizon is $\phi = 62°$.

Later versions of Leosphere's Windcube instruments use a modified reconstruction (Krishnamurthy, 2020) for the vertical

velocity (equation 9), which weights the beams in the reconstruction using the estimated wind direction, $\Theta$, measured clockwise

from due north, as presented in Newman et al. (2016) and Sathe et al. (2011). Re-weighting emphasizes beams along the mean

wind direction, exploiting the fact that decorrelation distances along the mean wind direction are typically longer than along

the cross-stream direction.

$$\boldsymbol{u}_{lidar} = \begin{pmatrix} \dfrac{u_E - u_W}{2\cos(\phi)} \\ \dfrac{u_N - u_S}{2\cos(\phi)} \\ \dfrac{(u_N + u_S)\cos^2(\Theta) + (u_E + u_W)\sin^2(\Theta)}{2\sin(\phi)} \end{pmatrix} \tag{9}$$

When the mean wind direction is at a $45°$ angle to the lidar axes (delineated by the south-north and east-west beam pairs),

the weights reduce to the uniform $\frac{1}{4}$ in the original reconstruction (equation 8). When the mean wind aligns directly with one

of the lidar axes, only the two respective beams on that axis are used. For our tests, we use reconstruction with wind-direction

weighting to represent currently utilized versions of the instrument.





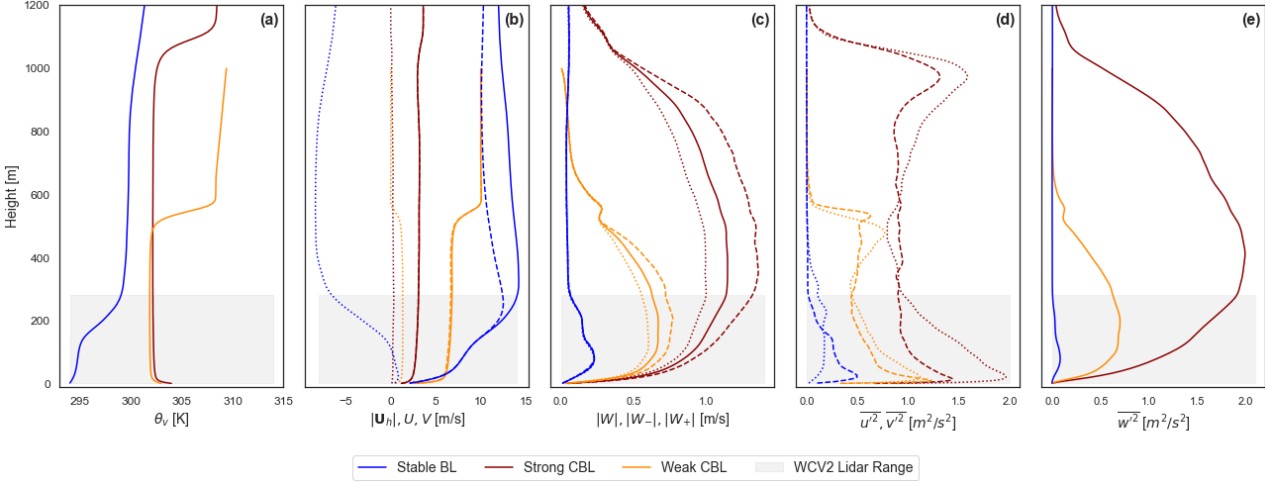

**Figure 3.** For each of the idealized LES cases, mean profiles of (a) virtual potential temperature; (b) horizontal winds: wind speed (solid), $U$ (dashed) and $V$ (dotted); and (c) mean vertical velocity magnitudes (solid) and the mean magnitudes over just the positive (dashed) and negative (dotted) vertical velocity values. Panels (d) and (e) show the turbulent velocity variances, $\overline{u'^2}$ (dashed), $\overline{v'^2}$ (dotted), and $\overline{w'^2}$ (solid). The grey region demarcates the WindcubeV2 vertical range including the region of influence for the range gate weighting.

## 2.2 Idealized atmospheric boundary layer simulations of varying stability

Realistic atmospheric flow fields were generated using large-eddy simulation (LES) configurations of the Advanced Research Weather Research and Forecasting (WRF-ARW) model v4.1 (Skamarock et al., 2019). WRF-ARW is a finite difference nu-

merical model which solves the flux form of the fully compressible, nonhydrostatic Euler equations for high Reynolds number flows. The model runs on a staggered, Arakawa-C grid with stretched, terrain-following hydrostatic pressure coordinates in the vertical. The simulations in this study employed a third-order Runge-Kutta time integrator and fifth- and third-order horizontal and vertical advection. All simulations used a nonlinear backscatter and anisotropy (NBA2) sub-filter scale stress model (Mirocha et al., 2010).

To establish a baseline reference for lidar operation in ideal conditions, all simulations in this study used uniform flat, grassy terrain (roughness length $z_0 = 0.1m$), periodic boundary conditions, with temporally and spatially invariant forcing. The idealized scenarios isolate fundamental characteristics of the atmospheric flows, removing potential influence from additional complexities and inhomogeneities from e.g. mesoscale forcing, varied terrain and land cover, and the influence of the diurnal cycle or nearby obstructions like wind turbines. None of the simulations in this study incorporated models for moisture, clouds,

radiation, microphysics, or land surface. The simulations were distinguished by varying stability: two convective cases and one stable stratification case, detailed in table 1 and figure 3. For each of the simulations, we used ten minutes of simulated time after spin-up was achieved, output at one second intervals.

For the convective boundary layers, we use data from precursor simulations in Rybchuk et al. (2021). Following the labeling therein, we designate the cases as the 'strong' and 'weak' convective boundary layers. Although both are considered strongly





**Table 1.** Parameters for WRF LES runs used to represent different stability regimes.

| Case | Strong CBL | Weak CBL | Stable BL |
|---|---|---|---|
| Domain Size ($L_x \times L_y \times L_z$) [km] | ($5 \times 5 \times 2$) | ($3 \times 3 \times 1$) | ($9 \times 3.99 \times 2.5$) |
| Cell Count ($N_x \times N_y \times N_z$) | ($500 \times 500 \times 200$) | ($500 \times 500 \times 160$) | ($1300 \times 570 \times 66$) |
| Horizontal Resolution [m] | 10 | 6.25 | 7 |
| Bottom Cell Height [m] | 3 | 3 | 5 |
| Surface Heating [K m s$^{-1}$] | 0.24 | 0.1 | – |
| Surface Cooling Rate [K h$^{-1}$] | – | – | -0.5 |
| Obukhov Length [m] | -5.44 | -12.32 | 78 |
| Friction Velocity [m s$^{-1}$] | 0.29 | 0.44 | 0.21 |
| Boundary Layer Height [m] | 1050 | 525 | 170 |
| Typical 100m winds ($\overline{|U|}, \overline{|V|}, \overline{|W|}$) [m/s] | (2.8,0.2,0.9) | (6.4,1.3,0.6) | (8.4,0.2,0.2) |

convective by their Obukhov length classification (Muñoz-Esparza et al., 2012) and are largely dominated by cell structures (Salesky et al., 2017), the cases differ meaningfully in the relative strength of the surface heating and geostrophic winds. The strong CBL case features stronger heating and slower winds than the weak CBL.

The stable boundary layer simulation closely follows the configuration of Sanchez Gomez et al. (2021). The surface condition for the stable case is driven with a cooling rate, rather than a negative heat flux (Basu et al., 2007). Spinning up a stable case
to relatively steady turbulence statistics can also be computationally expensive; a set of two one-way nested domains was used to reduce the computational demand. The parent domain has periodic boundary conditions and a horizontal resolution of 70m. It evolves for about 13.5 hours before the inner domain is started and simulated over the final 45 minutes. To reduce the fetch required to spin up fine turbulence in the nested interior grid, we employ the cell-perturbation method (CPM) from Muñoz-Esparza et al. (2014). The first 30 minutes of data from the fine-scale domain are discarded and only an interior region
excluding fetch and edge effects is used for the virtual lidar sampling.

Mean profiles, computed with data from the valid regions of the LES cases, are characteristic of the respective stability regimes (figure 3). Well-developed mixed layers, with consistent virtual potential temperature and wind speeds, account for the majority of the lidar observation range in the convective boundary layers. The bottom two reported range gates, however, incorporate values from the surface layer due to range gate weighting. The weaker convective case has significantly stronger
winds and the surface heating only supports a boundary layer about half as high as in the strong CBL case. The entrainment zone is out of lidar range in both cases. Vertical velocity magnitudes reach maximum values in the middle of the convective boundary layers, with a notable gap between the mean negative and positive values reflecting strong upward plumes and weaker, broader downdrafts. In the stable case, the boundary layer falls entirely within the lidar range. The distinctive temperature stratification is paired with strong winds that reach a maximum in a jet not far above the top of the boundary layer. Vertical velocities are
typically small and balanced and become negligible aloft.





**Table 2.** Parameters used in the model to configure a representative WindcubeV2 lidar performing a DBS scan.

|  | WindcubeV2 |
| --- | --- |
| DBS Elevation Angle $\phi$ [deg] | 62 |
| Minimum Range Gate [m] | 40 |
| Range Gate Resolution [m] | 20 |
| Number of Range Gates [#] | 11 |
| Frequency [Hz] | 1 |
| Range Gate Weighting Function (RWF) | Pulsed lidar (equation 6) |
| FWHM Pulse Width $\tau$ [ns] | 165 |
| Temporal Range Gate $\tau_m$ [ns] | 265 |
| Velocity Reconstruction Equation | Wind-direction weighted vertical (equation 9) |

Cariou and Boquet (2010); Bodini et al. (2019)

## 2.3 Configuration of virtual WindcubeV2 ensemble

A virtual WindcubeV2 is created in the lidar model as described in the previous section and summarized in table 2. To maximize realizations of the instrument sampling from each of the LES flows, a grid of forty-five instances of the virtual WindcubeV2 is placed throughout each domain. The locations are spaced such that their scanning volumes did not overlap. Each lidar scan

coincides uniquely with surrounding flow structures, comprising a statistical sample from which to diagnose general trends in how the instrument might interact with the distinctive atmospheric variability of each regime.

    The mean background states of the LES cases are spatially and temporally consistent across the domain, including the direction of the prevailing winds. To account for potential differences due to the relative orientation of the lidar axes in the flow, the ensemble of virtual lidar were re-oriented at three additional offset angles $\{15°, 30°, 45°\}$, and allowed to sample the

LES fields again. The small sensitivity of the error to the relative orientation is discussed in section 4.4.1.

    Determining the error in the lidar observation depends on defining a reference 'truth'. Profiling lidar are often thought of as replacing meteorological towers, returning a vertical profile of three-dimensional velocities similar to a tower fitted with instruments, but what value the lidar should actually be thought of as 'measuring' is not so straightforward. The samples used to estimate the wind components lack the precise locality of tower instruments; the beams collecting line-of-sight data span

an increasingly large area with height, each incorporating additional vertical extent via the RWF. These factors suggest that a volume average might be a more appropriate reference truth (as suggested in e.g. Courtney et al. (2008)). The lidar reflects pieces of both representations: it has the spatial spread of the volume average, but depends on only a handful of points on the edge of the volume which impart higher variability similar to a pointwise profile.

    Along with a 'pointwise', tower-style truth profile of interpolated velocities above the instrument, we determined a volume-

averaged profile for each lidar. The volume-average was computed as the mean of all LES points which fall inside cylinders tracing the lidar scan radius (figure 1). Centered at each range gate, the cylinders were defined to have radius equal to that of





the scan cone and height corresponding the the vertical projection of the RWF range resolution. For the WindcubeV2 the range resolution is the FWHM of the RWF ($\approx 40m$) so that the cylinders are $40\sin(\phi) \approx 35.3m$ tall. At the lowest levels with the smallest volumes, a minimum of around 80 LES points are used which increased to several hundred points in the top cylinders.

## 3    Observation error trends

The error incurred in any individual measurement does not necessarily represent general behaviors; deducing useful, generalizable trends entails focusing instead on distributions of the observation error. Each virtual instrument in the ensemble provides one instance of the ways the WindcubeV2 might interact with turbulent features in each flow regime, thereby sampling the error distribution. We characterize the resulting error distribution, and trends in its behavior, from the raw 1Hz lidar output as well as for time-averaged quantities.

We define the lidar error as the difference between the lidar-observed value and the reference truth, $u_{err} = u_{lidar} - u_{ref}$. Along with errors in each of the reconstructed wind components $(u, v, w)$ we consider errors in the horizontal wind speed and direction derived from the components (equation 10) which are also reported by the lidar and commonly used.

$$|\boldsymbol{u}_{h,lidar}| = \sqrt{u_{lidar}^2 + v_{lidar}^2}, \qquad \Theta_{lidar} = \arctan 2(-u_{lidar}, -v_{lidar}) \tag{10}$$

The wind direction, placed in the appropriate quadrant, is compactly represented by the two-argument inverse tangent (the sign and order of the arguments follows meteorological conventions with the angle measuring the wind source clockwise from north). Wind direction error is bound in the interval $(-180°, 180°)$ where positive values indicate the lidar reading an angle clockwise from truth and negative values an angle counter-clockwise from truth.

### 3.1    Raw 1-second velocities

A lidar reports a vertical profile of velocities each second for the ten minute duration of the simulation. Each distribution consists of ten minutes of data combined over the 45 ensemble members and four orientation angles, giving a total of 108,000 error samples. Disaggregating by height and stability, kernel density estimates (KDE) of the error histogram visualize the resulting distribution. Collating the KDEs into a ridgeline plot (figure 4), distinct variations in the distribution center, width, and shape appear. The distribution width in particular varies heavily with stability and height. Visual inspection confirms that the distributions are well-behaved and roughly normal with one central peak.

Statistical moments serve to summarize and quantify the properties of the distributions, facilitating intercomparison and the identification of trends. For each distribution, we computed the first four moments: unbiased estimators of the mean, centered variance/standard deviation, and standardized skewness and excess kurtosis (definitions in appendix B). The mean, $\mu$, represents the expected error, with non-zero values indicating a bias in the lidar observation. The centered variance, $\sigma^2$, measures the spread of the distribution about the mean, though the corresponding standard deviation, $\sigma$, can be easier to intuit as an indication of the distribution width and represents typical error magnitude in the original measurement units. For centered (zero mean) distributions, the standard deviation is comparable to the root-mean-squared-error (RMSE or RMSD) metric.





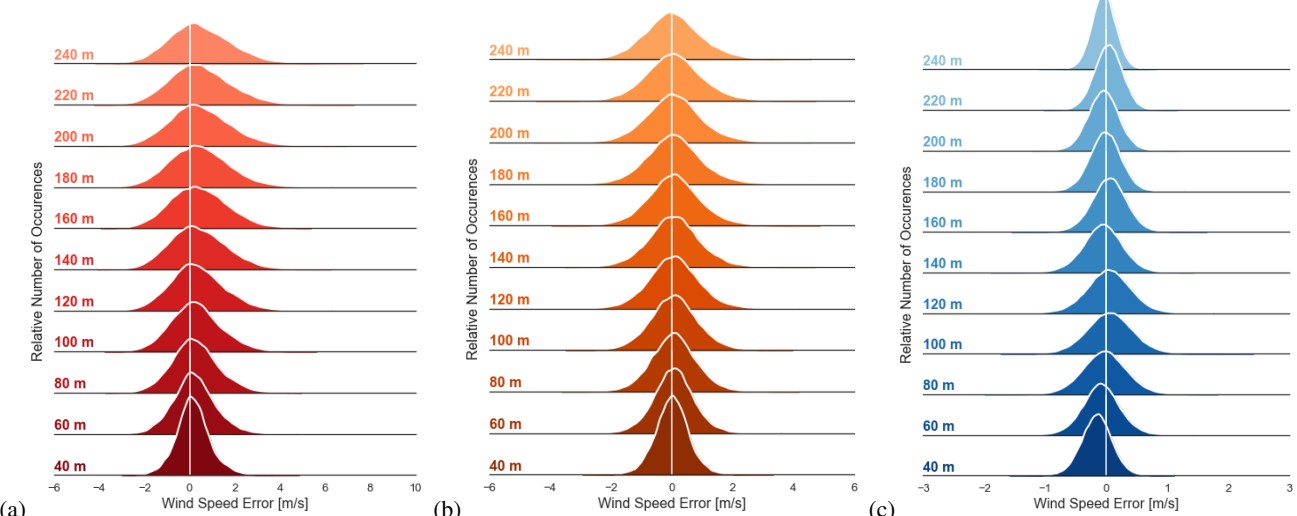

**Figure 4.** Kernel density estimates (KDEs) of aggregated wind speed error distributions. Errors are computed with respect to the volume average at each height for the (a) strong CBL (b) weak CBL, and (c) stable BL. Each distribution consists of 108,000 data points.

The higher-order moments of skewness and kurtosis, normalized by the standard deviation, are non-dimensional descriptors of the the distribution shape, namely asymmetries and decay of the tails. With a few exceptions, the metrics suggest that the distributions do not differ substantially from normal. There is a slight positive skewness (long tail on the positive side of the distribution) in wind speed estimates (0.25 in strong convection). The bottom range gates, influenced by the surface layer, also deviate from normal under strong convection: there is evidence of slight positive skewness (0.4) at the surface in wind direction and negative skewness (0.4) in the vertical velocity. Generally the excess kurtosis, though not extreme (+1-2), indicates more slowly decaying tails than those found in a normal distribution. The effect is more pronounced near the surface. Wind direction at the surface exhibits the most extreme behavior with an excess kurtosis of +8 at the surface. The distributions should be regarded as having more 'outliers' with respect to the width than would be found in a normal distribution.

The rest of discussion focuses on trends in the the mean and standard deviation of the error. For the sake of space, they are only shown for the wind speed and direction (figure 5) and for the vertical velocity (figure 6); see appendix A for the first four moments for all variables. Patterns in the error behavior are considered over stability cases, with respect to height, and as they appear in particular measured quantities.

In all cases, the distribution of the error with respect to the pointwise truth displays larger standard deviation than the error using the volume-averaged truth. With the beam measurements at the perimeter of the scan volume, the lidar reconstruction has no way to predict small-scale variations at the center of the volume where the pointwise truth resides. It can only reconstruct an average representation, and comparison with the point value incorporates additional uncertainty into the error through the pointwise truth reference.



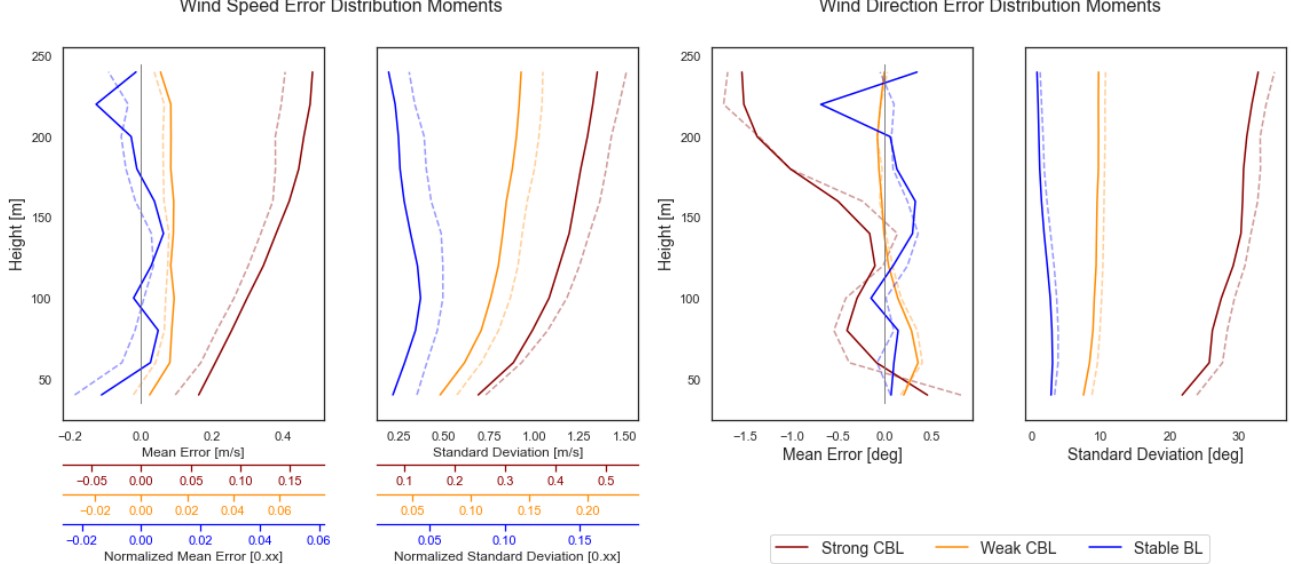

**Figure 5.** Mean and standard deviation of error in wind speed and wind direction using both volume-averaged truth (dotted) and pointwise 'tower' truth (dashed). Stability cases are distinguished by line color. For wind speed, the primary axis gives absolute error and colored secondary axes designate relative error with respect to 100m values for the respective LES case.

The stability case and corresponding structure of the boundary layer powerfully influence the error behavior. The strong convective case consistently suffers from more significant error, with greater bias and standard deviation in the wind speed, wind direction and vertical velocity error. It is followed by the weak CBL, with the stable case being the best behaved. The primary mechanisms identified in section 4 justify the relative ordering. The heterogeneity, especially in the vertical velocity,

of the convective plumes strongly drives the error while high wind speeds in the stable case help to reduce it.

Height trends depend on the change in volume circumscribed by the scan, but also on the vertical structure of the boundary layer (figure 3) and corresponding scale and character of the turbulent structures as noted by Wainwright et al. (2014) for sodar measurements. In the test convective conditions, the standard deviation of the error grows with height up to the top of the Windcube range. The error growth tracks the growth in the vertical velocities from the lower layers the middle of the boundary

layer where the convective plumes are strongest. In the stable case, the standard deviation reaches a maximum just below the boundary layer height at 170m. The peak in the error spread seems to occur close to the infection point in the $v$ mean velocity and the maximum vertical velocity magnitude. In these quasi-stationary conditions, the height trends depend strongly on the vertical structure of the flow, not just the size of the scan volume.

Some error traits are particular to the quantity being measured, outside of the general trends identified above. The wind speed

measurement exhibits a bias toward over-estimation (positive error mean), particularly in convective conditions. In the stable case, the distribution is more centered except close to the surface where the mean becomes negative (underestimate). Under strong convection, the wind direction observations also suffer from a bias; the direction lists more and more counter-clockwise





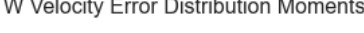

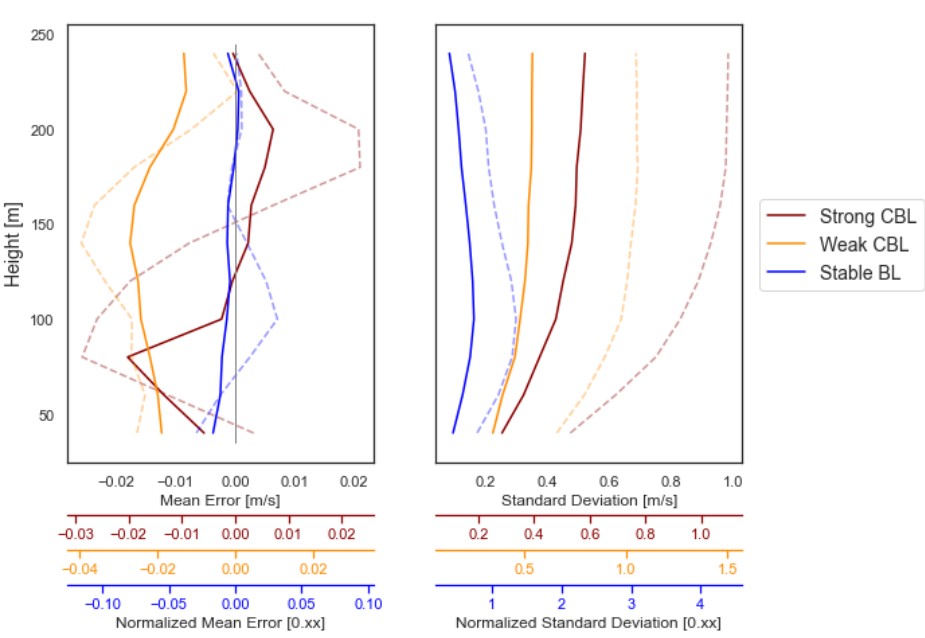

**Figure 6.** Mean and standard deviation of error in vertical velocity reconstruction (equation 9) using both volume-averaged truth (dotted) and pointwise 'tower' truth (dashed). Stability cases are distinguished by line color. The primary x-axis gives absolute error and colored secondary x-axes designate relative error with respect to typical, 100m values for the respective LES case.

(southward) from truth with height. In the vertical velocity reconstruction, the relative error can grow to large fractions of typical magnitudes in all stability cases, obscuring the actual signal in the measurement. Note the inversion of the usual

stability ordering in the vertical velocity mean error: measurements in weak convection experience larger bias than in strong convection. The vertical velocity error distributions using point or volume-averaged truth references do not mirror one another as they do for the other variables. Since the variability in vertical velocity occurs on a shorter scale, the pointwise value is less representative of the volume average which explains the greater discrepancy. The error in the WindcubeV2 reconstruction of vertical velocity (equation 9) shown here is compared against that in the equally-weighted reconstruction (equation 8) and

direct measurement by the vertical beam in section 4.6.

     The aggregate distributions described here are the result of large ensembles over turbulence realizations. They describe the distribution of the error in a single measurement made by an instrument randomly dropped into each stability regime. A time series of measurements does not randomly sample the aggregate distribution, rather it does so selectively based on the character of the few turbulent structures encountered. The selective sampling implies that the distribution of error in measurements made

by a single lidar over a finite time window (e.g. ten minutes) can have its own sub-structure (figure 7). The distributions in this case can display non-normal shapes as well as limited shifts in the mean and variance compared to the ensemble distribution. For example, if only one or two convective plumes move over the instrument, there may be multi-modality based on the strength





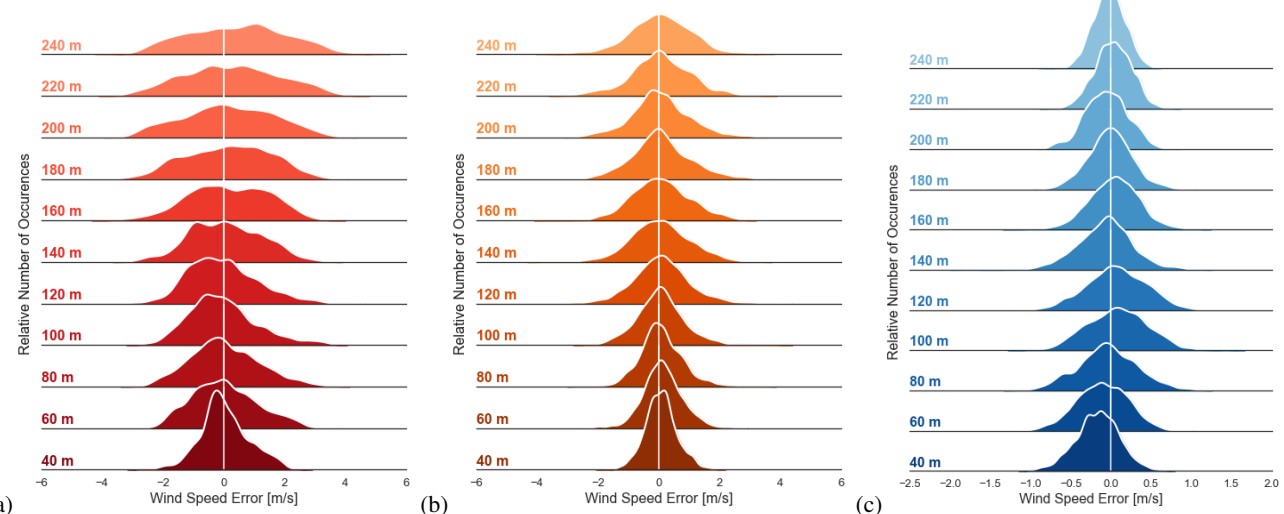

**Figure 7.** Kernel density estimates (KDEs) of the wind speed error distribution for a single lidar over 10 minutes. Errors are computed with respect to the volume average at each height for the (a) strong CBL (b) weak CBL, and (c) stable BL. Each distribution consists of 600 data points. The distributions are not representative.

of the vertical velocity of the plumes. Over long enough times, the instrument encounters more and more turbulent structures and the error will converges toward an ensemble distribution (figure 7). In light of the selective sampling, caution should be

exercised in assuming that the ensemble error distributions presented here can be used to represent the distributions of error in a specific time series of measurements.

Though the error magnitudes are not generally large, the behavior is far from uniform and exhibits strong dependence on the flow itself, even with respect to height within the same stability regime. The mechanisms driving the observed trends in error are broken down in depth in section 4.

**3.2 Ten-minute time averaged velocities**

Time averaging is often used to filter out random error 'noise' in the raw measurements output with each beam update. Common averaging intervals may be over two, ten, or even thirty minutes, with experimental evaluations of the system accuracy often reported in terms of the ten-minute average in the wind energy context. As with the error in the high(er) frequency wind measurements, we characterize the error distribution of the ten-minute averaged measurements (figure 8).

Velocity component observations are averaged over the ten-minute simulation span and compared to the time-averaged truth references. The wind speed and direction are not averaged directly but re-computed from the time-averaged wind components (equation 10) so that they correspond to a vector average of the wind. Wainwright et al. (2014) and Gasch et al. (2019) distinguish the 'scalar' average from this kind of 'vector' average while more detailed analysis comparing the averages have



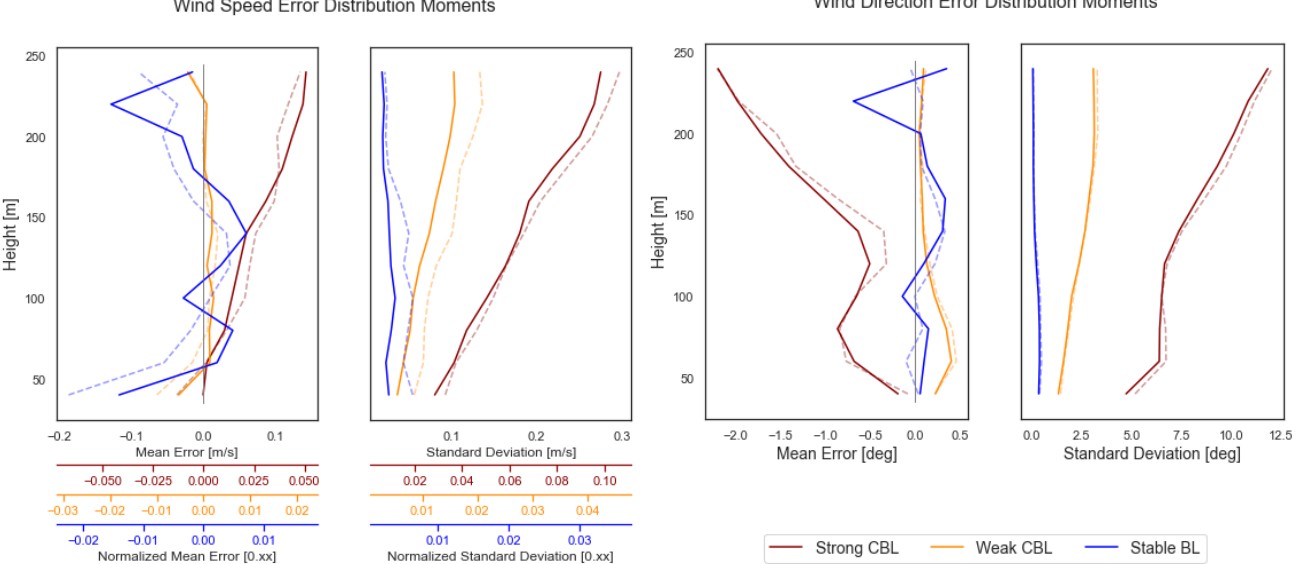

**Figure 8.** Statistical moments as in figure 5, but error distributions for the 10-minute averages.

been undertaken by Clive (2008) and Rosenbusch et al. (2021). We omit time-averaged vertical velocity error here. Each

distribution in the ten-minute average consists of an ensemble of a total of $180$ values (ensemble and offset angles).

Under quasi-stationary conditions, the notions of pointwise and volume-averaged truth start to converge to a general spatio-temporal average, which is reflected in the merging of the two error distribution metrics. The correspondence suggests that field studies comparing against time-averaged 'point' tower measurements can effectively reflect the error with respect to the volume-average as well. The overall error magnitudes found by the virtual lidar are consistent with those in field deployments

of lidar compared against tower measurements. In select, flat conditions typical mean discrepancies in the range $\pm 0.2$ m/s with standard deviations of 0.20 m/s (Courtney et al., 2008) have been reported, which encompass all but the more extreme errors in the strong convection case. The virtual ten-minute averaged wind speeds have mean errors within $\pm 0.2 m/s$ with standard deviation $< 0.3 m/s$ and wind direction have mean error bias within $2°$ and standard deviations in $2.5°$ except for in the strongly convective case where it can reach up to $12°$.

Many of the qualitative trends identified in the raw, 1 Hz measurement error persist in this averaged assessment; the ordering of relative error between stability conditions and their respective height trends are similar to the previous section. While respecting these general trends, the form of the moment curves shift a little compared to the original distribution, e.g. the curvature of the standard deviation with height.

The major benefit of the time average lies in the marked reduction (by a factor of around 5) of the error standard deviation

across cases and heights. The erroneous rotation in the strong CBL wind direction mean remains although the time average lessens the over-estimation bias in the wind speed. The distribution shapes in the ten-minute average also grow closer to a normal distribution shape.





## 4 Analysis of error mechanisms

The following analysis systematically addresses how turbulent variations induce error in the wind reconstruction and how that
error propagates into derived quantities. Much of the analysis presented in this section is quite general and applies to any DBS
reconstruction of the form in equation 8 and in any flow condition. The approach can be extended to different scan types as
well in decomposing their error.

Two elements directly introduce error into the observation model: the application of the range gate weighting function
in the radial velocity measurement and the assumption of horizontal homogeneity in the reconstruction. Using a random
variable model, we identify the contributions of the RWF and homogeneity violations to the error in the wind component
reconstructions. Accumulation time and time-staggering of the beams are not addressed. The behaviors exhibited in the virtual
lidar data are tied back to the separate elements based on the character of the flow regime.

Values computed using the estimated wind components can take on their own, non-trivial error behavior. We characterize
the error in wind speed and direction in terms of the $u$ and $v$ error distributions and examine its dependence on the background
wind speed and direction. A time average is also a function acting on the raw velocity data and we trace and quantify its
expected and observed effect on the error distributions.

Finally, an empirical comparison is made between the error in the different Windcube vertical velocity reconstruction techniques and the direct measurement made by the vertical beam.

### 4.1 Random variable model of turbulent variation in wind component error

To break down the driving sources of error in the reconstruction of the velocity components, we start by deriving the form
of the error. The formulation allows the contribution to the error due to the turbulent variations to be explicitly delineated
and tracked. For a fixed height, let $\boldsymbol{U} = (U, V, W)^T$ be the mean velocity across the scan volume, i.e. the volume-averaged
truth, and assume that it is constant through the five second scan duration. (Different notions of the volume average could
be used here, but the disk seems the most useful average representation to measure). Each angled beam samples a perturbed
velocity, $\boldsymbol{u}_{E,W,N,S} = \boldsymbol{U} + \boldsymbol{u}'_{E,W,N,S}$ where the subscript denotes the cardinal direction of the beam azimuthal direction. The
measurement of the projection of the perturbed, point velocity is subject to an additional perturbation, $r_{E,W,N,S}$, due to the
range gate weighting function. Then the radial velocity measured by the beam pointed east, for example, is given by:

$$u_E = \cos(\phi)(U + u'_E) + \sin(\phi)(W + w'_E) + r_E$$

Carrying the forms through the reconstruction (equation 8), the error in the wind components is given by the difference with the
volume-averaged value $\boldsymbol{U}$. (This analysis is similar to that of Newman et al. (2016), which extends the derivation to turbulent
variances). For the horizontal $u$ and $v$ components, the resulting representation of the error is given in equations 11 and 12.
The vertical velocity error form depends on whether the wind-direction weighting is used; we limit our analysis to the equally

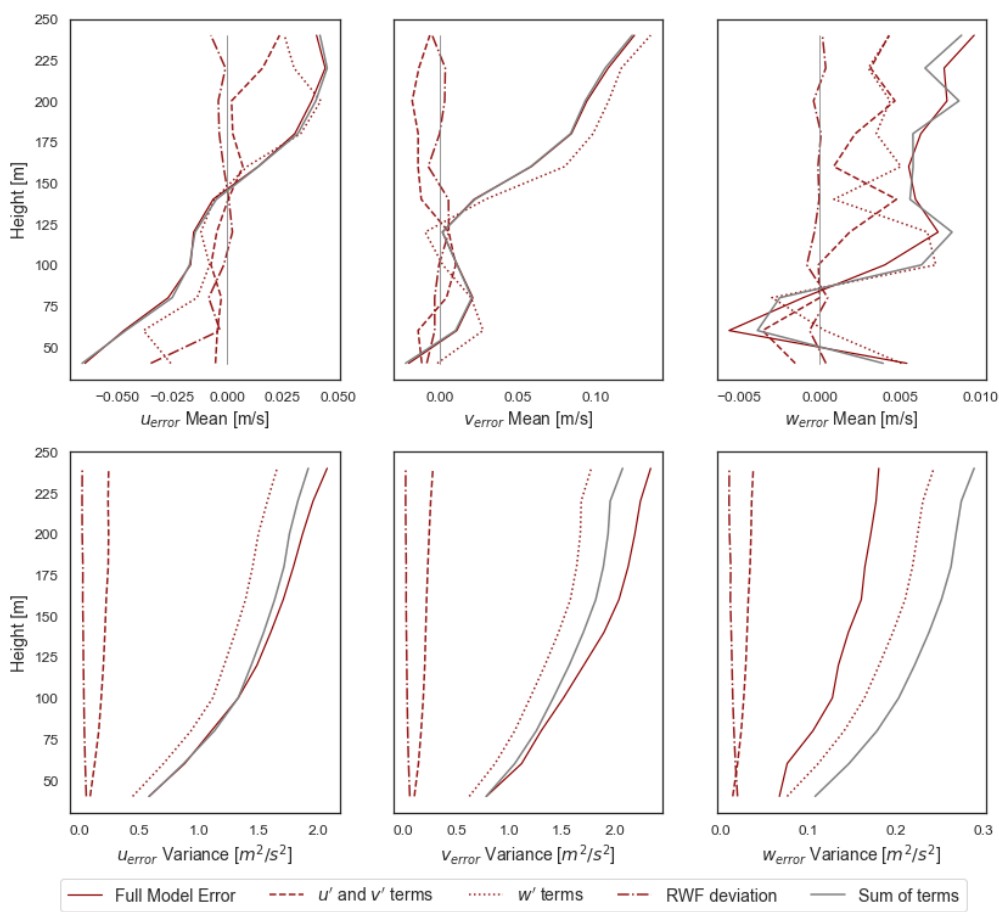

**Figure 9.** Decomposition of error for each wind component in the strong convective stability case. The mean and variance are partitioned according to equations 14-17 into contributions from perturbations in horizontal $(u', v')$ and vertical $(w')$ velocities and the RWF measurement $(r')$. Only the data from virtual lidar aligned with the LES $U, V$ axes are used. The sum of the plotted components is given in grey. Discrepancy in the variance between the full model and the sum of terms is due to omitted covariance terms.



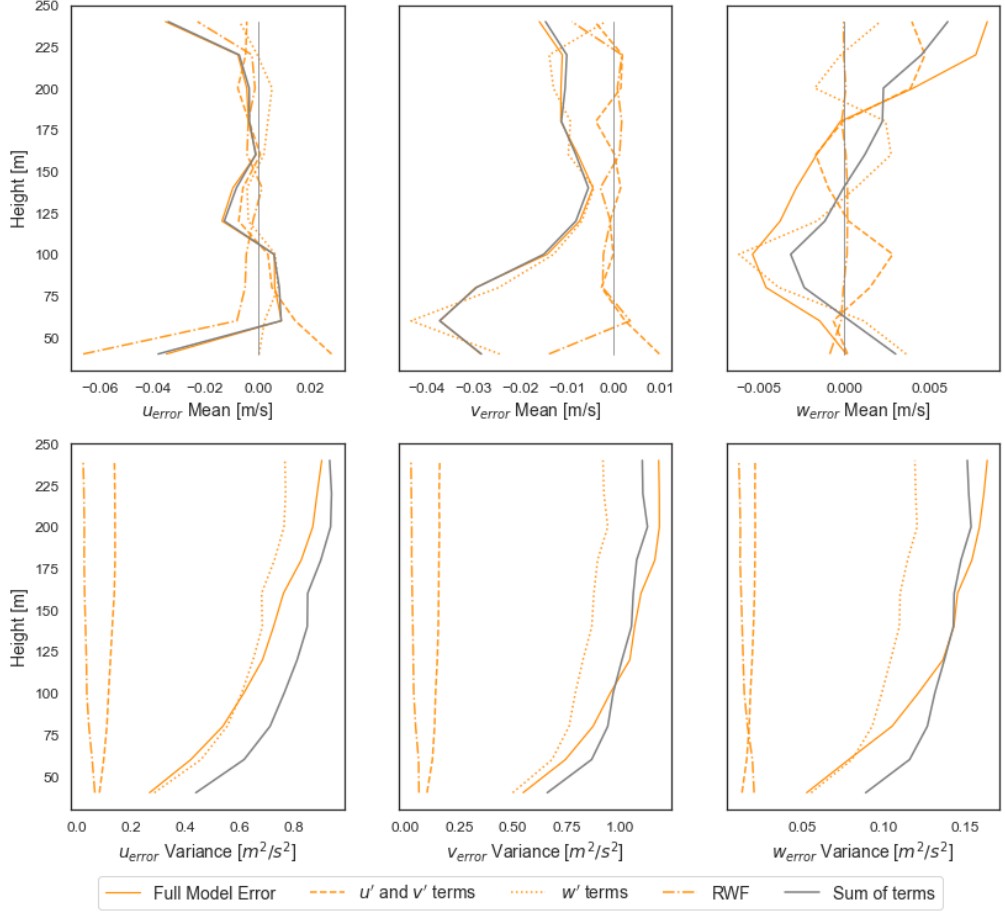

**Figure 10.** Decomposition of wind component error under the weak convection case, as in figure 9.

weighted version (equation 13).

$$u_{err} = u_{lidar} - U = \frac{1}{2} \left[ (u'_E + u'_W) + \tan(\phi)(w'_E - w'_W) + \sec(\phi)(r_E - r_W) \right] \tag{11}$$

$$v_{err} = v_{lidar} - V = \frac{1}{2} \left[ (v'_N + v'_S) + \tan(\phi)(w'_N - w'_S) + \sec(\phi)(r_N - r_S) \right] \tag{12}$$

$$w_{err} = w_{lidar} - W = \frac{1}{4} \left[ (w'_N + w'_E + w'_S + w'_W) + \cot(\phi)(u'_E - u'_W + v'_N - v'_S) + \csc(\phi)(r_N + r_E + r_S + r_W) \right] \tag{13}$$

In a perfectly horizontally homogeneous wind field, the velocity perturbation values all individually vanish (i.e. not due to cancellations). However, even in that perfectly horizontally homogenous case, a non-linear vertical profile can still induce nonzero error through the RWF. Further, in the presence of turbulence in the flow, homogeneity is violated and the perturbation values may be regarded as random variables with distributions resulting from the character of the atmospheric variations and the lidar scan geometry. Under this model, spatial trends in the background flow would be expressed in shifted perturbation mean





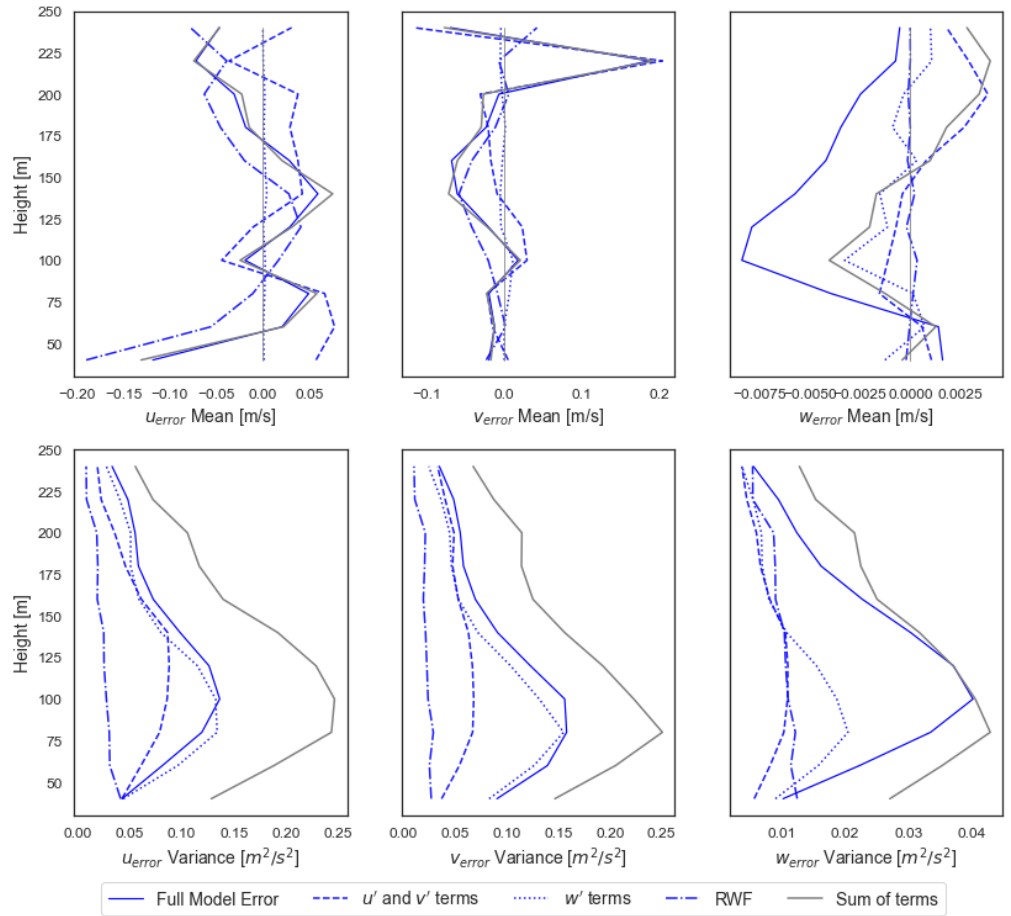

**Figure 11.** Decomposition of wind component error under the stable case. as in figure 9.

values at the respective beam locations. The error formulation defines how the inhomogenieties in separate wind components and effects of the RWF combine to produce the total error.

As functions of the random perturbations, the wind component errors are themselves random variables. The mean, $\mu$, and

450 variance, $\sigma^2$, of the error distributions can be expressed in terms of the perturbation distributions through algebra of random variables. The mean operator is linear and directly decomposes the overall error mean into constituent parts for the horizontal (equation 14) and vertical (equation 15) wind components.

$$\mu(u_{err}) = \frac{1}{2}[\mu(u'_E) + \mu(u'_W)] + \frac{\tan(\phi)}{2}[\mu(w'_E) - \mu(w'_W)] + \frac{\sec(\phi)}{2}[\mu(r_E) - \mu(r_W)] \tag{14}$$

$$\mu(w_{err}) = \frac{1}{2}[\mu(w'_E) + \mu(w'_W) + \mu(w'_N) + \mu(w'_S)] + \frac{\cot(\phi)}{2}[\mu(u'_E) - \mu(u'_W) + \mu(u'_N) - \mu(u'_S)]$$

$$+ \frac{\csc(\phi)}{2}[\mu(r_E) + \mu(r_W) + \mu(r_N) + \mu(r_S)] \tag{15}$$





The variance ($\sigma^2$) can also be decomposed into a linear combination of constituent terms, but introduces covariance terms (equations 16-17). (The variance is preferred here to the standard deviation ($\sigma$) reported in section 3 so that the contributions are additive , i.e. no square root). We do not explicitly expand the covariance terms which quantify correlations between the perturbations.

$$\sigma^2(u_{err}) = \frac{1}{4}[\sigma^2(u'_E) + \sigma^2(u'_W)] + \frac{\tan^2(\phi)}{4}[\sigma^2(w'_E) + \sigma^2(w'_W)] + \frac{\sec^2(\phi)}{4}[\sigma^2(r_E) + \sigma^2(r_W)] + \text{covariance terms} \qquad (16)$$

$$\sigma^2(w_{err}) = \frac{1}{4}[\sigma^2(w'_E) + \mu(w'_W) + \sigma^2(w'_N) + \sigma^2(w'_S)] + \frac{\cot^2(\phi)}{4}[\sigma^2(u'_E) + \sigma^2(u'_W) + \sigma^2(u'_N) + \sigma^2(u'_S)]$$

$$+ \frac{\csc^2(\phi)}{4}[\sigma^2(r_E) + \sigma^2(r_W) + \sigma^2(r_N) + \sigma^2(r_S)] + \text{covariance terms} \qquad (17)$$

As before, the mean of the error distribution describes offsets, or biases, in the error; it represents how quantities are consistently over- or underestimated. The variance of the error describes the spread in the error values; it reflects the expected magnitude of errors on top of any systematic, mean bias.

The relative weighting of the perturbations is controlled by the scan cone elevation angle from the horizon, $\phi$ (figure 1), and describe the result of the relative projection of the perturbations on the beam. For the WindcubeV2 cone angle, $\tan\phi \approx 1.88$, so that the vertical velocity perturbations are weighted almost twice as heavily as the horizontal velocity perturbations. The radial velocity perturbations are similarly heavily weighted, with $\sec\phi \approx 2.1$. The asymmetric weighting is further exacerbated in the variance which uses the squares of these values. The cone angle also controls the spatial separation of the beams and can thus implicitly influence the distributions of the perturbations themselves. The beam separation can be of particular importance in the presence of background spatial variation in the flow, potentially producing error bias.

The virtual lidar model uses the LES to predict the perturbation distributions and the complex ways in which the perturbations can be interrelated with each other and with respect to the volume averages. Random variable theory can be used to describe the propagation of uncertainty into the error from the attributes of the perturbation distributions. The virtual lidar and LES data are used to compute the terms in the decomposition of the error mean (equations 14 and 15) and variance (equations 16-17) over each of the stability cases (figures 9, 10, 11). The error mean in the overall model should be equal to the sum of the decomposed terms. The variance decomposition omits the covariance terms, the combined effect of which accounts for the remaining gap between the sum and the full model error. The explicitly tracked terms explain how much of the total error variability can be ascribed to each set of (weighted) perturbations. In the following sections, the behavior of the decomposed wind component errors in the test cases are explored.

## 4.2 Effects of range gate weighting

The size of discrepancies in the radial velocity measurement due to weighting by the RWF may be analytically bound. The bound serves to illuminate the conditions under which the perturbations from the point value can become large.

Assume any RWF, $\rho(s)$, is a non-negative, symmetric function which monotonically decays as $s \to \pm\infty$ and satisfies $\int_{-\infty}^{\infty} \rho(s) = 1$. Let $v_r(r)$ be the radial velocity profile along the beam and $R > 0$ a threshold distance from the target $r_0$. Then under the RWF model, the size of the discrepancy between the range-gate-weighted measurement, $\overline{v}_r(r_0)$, and the actual,





pointwise radial velocity, $v_r(r_0)$, is constrained in equation 18 (derivation in appendix D).

$$| \, \overline{v}_r(r_0) - v_r(r_0)| \leq \left[ 1 - \int_{|s| \leq R} \rho(s)ds \right] \left( \max_{s > |R|} |v_r(r_0 + s)| + |v_r(r_0)| \right) + \left[ \frac{1}{2} \int_{|s| \leq R} \rho(s)s^2 ds \right] |v_r''(r_0 + \xi_*)| \tag{18}$$

The bounding terms can be forced to be small by selecting $R$ to manipulate the coefficients. The magnitude of the radial velocities in the atmosphere can practically be expected to be finitely bound. Taking $R$ to be large drives the integral of $\rho(s)$ to one and thus the first term to zero. The second term does the opposite: the coefficient grows rapidly with $R$ and is small for small $R$. The tension between the requirements picks out the conditions which allow for potentially large deviations in the weighted measurement from the true point value.

Radial velocity profiles with constant gradient do not incur error in the RWF application; symmetry leads the linear contributions to cancel. Indeed, visual inspection confirms this behavior in regions of constant gradient about $r_0$, which incur only small discrepancies (figure 2(b,c,d)). The competition between the remaining bounding terms (equation 18) places requirements on the radial velocity behavior itself; in the absence of large curvature in the radial velocity profile ($|v_r''|$ small), the error can be expected to be negligible. Data collected by the virtual lidar affirms such behavior in practice, showing the largest 500 misrepresentations in areas with sharp bends in the radial velocity profile (figure 2).

The impact of the RWF on the total error in the test cases is generally limited compared to that of the horizontal inhomogeneities, except where the RWF interacts with the surface layer. Its contribution ($r'$) to bias and variability of the wind component error is separated in each stability case according to the model in the previous section. In the convective regimes, the proportion of error variance from RWF perturbations is negligible across heights and wind components (figures 9,10). The 505 relative impact is larger in the stable case, especially for vertical velocity, though the absolute magnitude is still small (figure 11). The most prominent influence of the RWF is near the surface layer (due to strong shear) and manifests as a negative shift in the mean error in the horizontal velocities. The weak CBL and stable BL, with strong winds and surface shear, exhibit the effect most strongly. Assuming a logarithmic profile, a large second derivative is to be expected close to the surface, which our analysis suggests is conducive to larger error in the radial velocity measurement. The logarithmic curve in $u$ has a consistent 510 shape which induces an repeated underestimates by the RWF, $r_E < 0$. Accounting for the negative projection of $u$ on the west beam, $r_W > 0$ so that the difference $r_E - r_W$ is systematically negative, leading to a negative bias in $u$ measurements. It follows that if the the mean winds were easterly, the sign of the bias would also flip ($r_E - r_W > 0$) so that the magnitude of the wind component would still be underestimated. Our findings linking the effect of shear on lidar error due to the RWF echo those found by Courtney et al. (2014), and would be applicable in other strong-shear regions of the atmosphere such as at the top of 515 a wind turbine wake.

### 4.3 Horizontal homogeneity violations

Violations of horizontal homogeneity across the scan volume are directly represented in the error model by the velocity perturbations in the wind component errors (equations 11,12,13). Horizontal variations may also be reflected in the radial velocity along the beam and are assumed to be encapsulated in the treatment of the RWF. The perturbations in the velocity around the

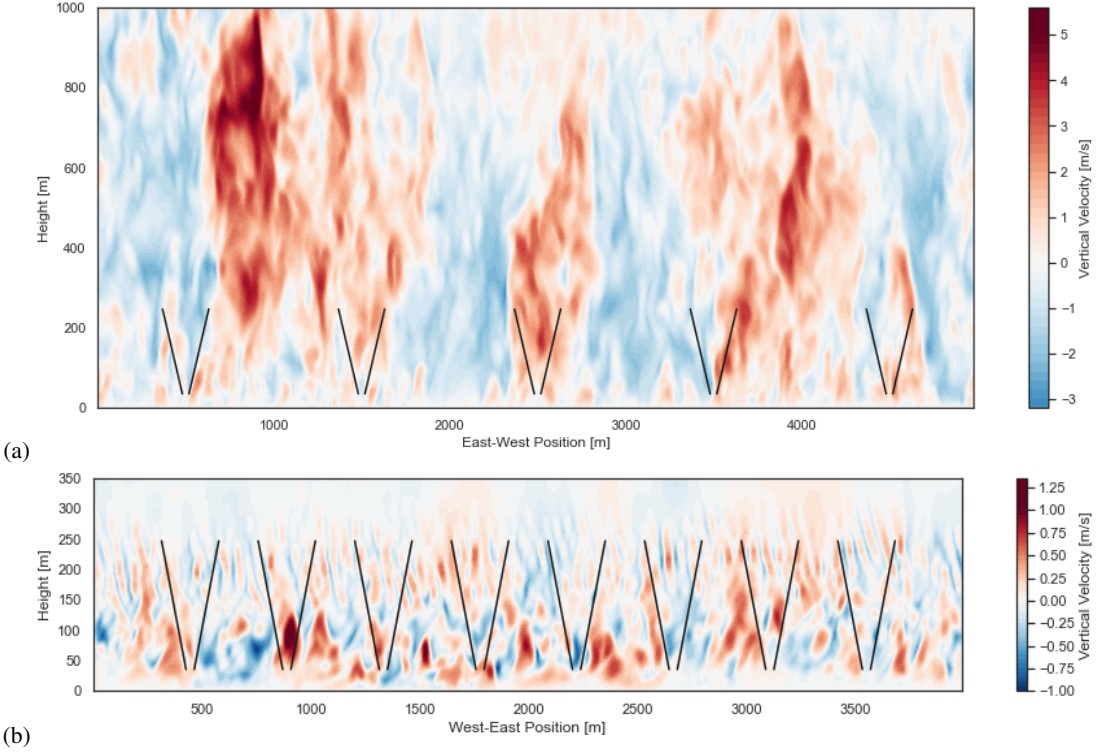

**Figure 12.** Instantaneous slice of vertical velocity across the west-east plane in (a) the strong CBL and (b) the stable BL. The WindcubeV2 scan geometry is shown for reference by the beams in black.

scan radius occur due to heterogeneity on a larger spatial scale than those along a single beam, potentially permitting larger turbulent structures with larger variations.

To describe the error due to the perturbation terms, we consider what they represent and how they relate to the turbulence in the flow. In the lidar model, the velocity perturbations ($u'_{beam}$) are taken with respect to the average over the scan volume; they are a kind of turbulent fluctuation under the high-pass filter based on the scale of the scan volume (i.e. filtering out turbulence at

the larger scales). The variance of the perturbations ($\sigma^2(u'_{beam})$), which determines their magnitude, is a filtered fraction of the full turbulent Reynolds velocity variance (e.g. $\overline{u'u'}$). As the size of the scan volume increases, the volume average approaches an ensemble mean so that the perturbations are Reynolds fluctuations. The variance expected in the lidar perturbations is determined by the proportion of turbulent variances in each direction above the filter scale (figure 13), with the full variance constituting a 'cap' on the total possible variance.

It is tempting, with usual conventions about turbulent perturbations, to assume that the lidar velocity perturbations will have zero mean and identical distributions at each of the beam locations. Under these assumptions, the mean error due to the the horizontal homogeneity violations would be zero. However, the volume average mean is neither the direct mean of the beam velocities nor is it the turbulent ensemble mean. The velocity perturbations can produce a non-zero mean because of





consistently occurring spatial patterns in how the velocities vary at the edges of the scan volume with respect to the average
velocity over that volume. The spatial structures at play in the LES cases with respect to the lidar scan volume can be seen in
the cross-sections in figure 12. If the turbulent structures are small enough and the scan volume large enough, then the volume
average does approach the turbulent ensemble mean and the beams sample separated, independent turbulent fluctuations. Under
such conditions, the assumption of zero mean and identically distributed perturbations at each beam are appropriate. However,
when there are coherent turbulent structures on the order of the scan volume, the volume average no longer represents a
turbulent ensemble mean. Without dissecting the mechanics more closely, we simply note that large coherent structure like
turbulent plumes characteristic of the convective boundary layer can induce repeated, non-symmetric patterns in the relative
perturbations of the beams, leading to non-zero means.

We can now see how these behaviors play out in the virtual lidar data. Using the derived forms (equations 14-17), the
contributions from the horizontal $(u', v')$ and vertical $(w')$ perturbations to the wind component errors are delineated for each
test case (figures 9, 10, and 11). For the most part, the error mean biases can be attributed to RWF effects and the velocity
perturbation terms do have close to zero mean, but important deviations from that assumption do arise. The largest deviations
from zero mean error (except for the RWF) occur in the strong CBL case and stable BL case, though they do arise elsewhere.
We attribute these deviations to the presence of large coherent structures on the order of the scan volume as discussed above,
which agrees with the presence of the large convective plumes as well as the larger-scale structures above the boundary layer
in the stable BL.

In convective conditions, the weighted perturbations in vertical velocity dominate the other sources of variance in the error,
indicating that the vertical velocity perturbations become larger than the other terms and dominate the occurrences of large
magnitude error. The stronger the convection, the more dominant the role of the vertical velocity terms. By contrast, the error
in the stable BL arises from a more even interplay in the horizontal and vertical velocity perturbations. The covariance between
the beam perturbations also generally serves to temper the overall error variances.

The relative contributions of the vertical and horizontal perturbation terms is due to the variance of the perturbations them-
selves, which we identified as the filtered portion of the full turbulent velocity variance, and the weighting of the terms in the
error (equations 16 and 17). Physically, we might expect convective plumes to violate the desired horizontal homogeneity in
the flow 12), but it is an even more outsized effect that creates the error. The convective plumes induce large but localized
vertical velocity variances so that the lidar scan volume does not filter out the bulk of vertical velocity variance while more of
the horizontal turbulence is filtered out (figure 13). The cone angle then over-weights the vertical velocity variance terms. The
compounding effects conspire to make the vertical velocity dominant in creating large errors (figures 9 and 10). Even in the
stable boundary layer, which features much smaller vertical velocity variances, the vertical velocity perturbations contribute
significantly to the error. The smaller scales of the vertical velocity variations in stable conditions (figure 12) coincide with
smaller turbulent variance, but also allow a larger portion of the variance to be filtered into the lidar perturbations (figure 13).
The result is again over-weighted according to the cone angle. The contributions from the horizontal perturbations, meanwhile,
are lessened by more of the variations being filtered out by the scan volume and lack of relative weighting.





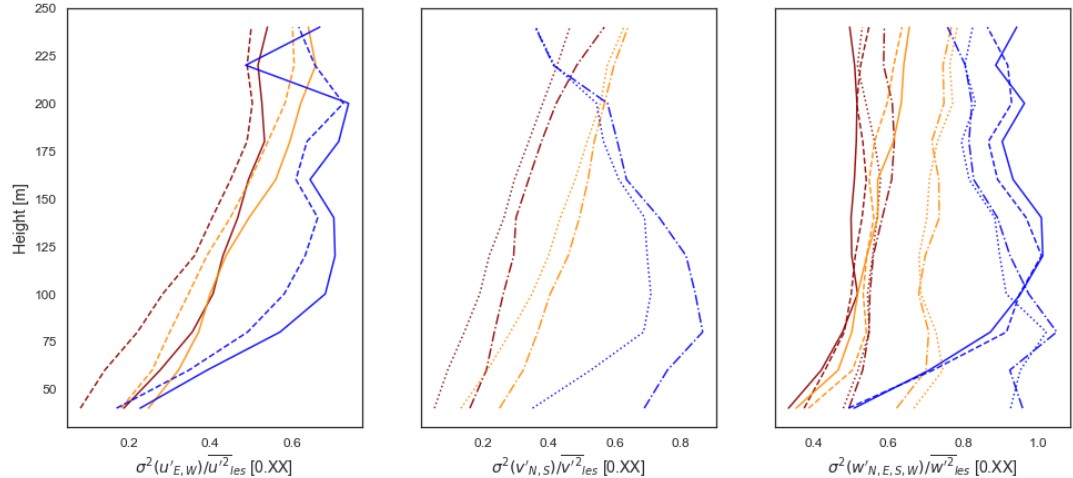

**Figure 13.** The proportion of the full turbulent velocity variances $(\overline{u'^2}, \overline{v'^2}, \overline{w'^2})$ present in the turbulent perturbations from the lidar scan volume average (e.g. $u'_E$). Ratios shown using the perturbation variances at the east (solid), west (dashed), north (dash-dotted), and south (dotted) beams.

The coupling of the error with the turbulent structure, and the vertical velocity in particular, helps explain the cause of the correlation in the error height trends with the boundary layer structure. Variance in the horizontal perturbations is more
effectively limited by the scan volume filtering, especially near the surface. In the convective cases, as velocity variance $(\overline{w'w'})$ grows from the surface to the middle of the mixed layer (figure 13(a)), so too does the variance in the $w'$ perturbations, driving a greater spread of error with increasing height. In this example, the lidar range does not extend to the top of the convective boundary layer, but without other background sources of heterogeneity the error variance might diminish higher up, even with the large scanning radius. The error variance in the stable boundary layer peaks near the center of the boundary layer where the
turbulent vertical velocity variances are large and more of the horizontal variances are filtered. The diminishing error variance at higher altitudes relies on the decay of both the horizontal and vertical perturbations at the top of and above the boundary layer and the increase in turbulent length scales which combat any increase in scan volume size.

Interpreting the velocity perturbations as turbulent fluctuations filtered by the scan volume, the connection between the lidar error variance and the structure of turbulence in the boundary layer flow becomes clear. The vertical velocity variance takes
on a more prominent role in the error due to the smaller spatial scales in the turbulent variations and the additional weighting on the terms from the beam projection angle. And while the perturbations are turbulent fluctuations, the spatial structure of the scan volume and beam samples can induce non-zero perturbation means in the presence of cohesive structures with sizes on the order of the scan volume scale. The large vertical variances and cohesive structures in convective regimes can induce larger lidar measurement errors.





### 4.4 Secondary effect on derived quantities: wind speed and direction

An often preferred representation of the observed horizontal wind vector is as a direction, $\Theta$ (meteorological convention), and magnitude, $|\boldsymbol{U}_h|$. The WindcubeV2 internally computes and reports these derived quantities by solving for them with the reconstructed velocity components (equation 10). The computed values do not inherit the error from the wind component errors directly; rather the quantities should be thought of as functions of the $u$ and $v$ errors treated above, taking on their own distinct,

related error distribution behavior.

As non-linear functions of $u$ and $v$, the error in the wind speed and direction do not drop out as cleanly as they did for the individual wind components. We can, however, still estimate the breakdown of contributions so we can analyze the behavior. Consider the error of the squared wind speed,

$$|\boldsymbol{u}_{h,lidar}|^2 - |\boldsymbol{U}_h|^2 = 2U u_{err} + 2V v_{err} + u_{err}^2 + v_{err}^2$$

To estimate the wind speed error itself, suppose that the lidar-derived estimate is close to the actual volume-averaged wind speed, $|\boldsymbol{U}_h|$. Then we can factor and approximate (first order in $|\boldsymbol{U}_h|_{err}$) as follows.

$$|\boldsymbol{u}_{h,lidar}|^2 - |\boldsymbol{U}_h|^2 = (|\boldsymbol{u}_{h,lidar}| + |\boldsymbol{U}_h|)(|\boldsymbol{u}_{h,lidar}| - |\boldsymbol{U}_h|)$$
$$= (2|\boldsymbol{U}_h| + |\boldsymbol{U}_h|_{err})(|\boldsymbol{U}_h|_{err}) = 2|\boldsymbol{U}_h||\boldsymbol{U}_h|_{err} + |\boldsymbol{U}_h|_{err}^2 \approx 2|\boldsymbol{U}_h||\boldsymbol{U}_h|_{err}$$

Combining with the squared speed error and solving for the $|\boldsymbol{U}_h|_{err}$, we find the error in the computed wind speed with respect

to the horizontal velocity component errors (equation 19).

$$|\boldsymbol{U}_h|_{err} = |\boldsymbol{u}_{h,lidar}| - |\boldsymbol{U}_h| \approx \frac{U}{|\boldsymbol{U}_h|}u_{err} + \frac{V}{|\boldsymbol{U}_h|}v_{err} + \frac{u_{err}^2 + v_{err}^2}{2|\boldsymbol{U}_h|} = \frac{\boldsymbol{U}_h}{|\boldsymbol{U}_h|} \cdot \boldsymbol{u}_{h,err} + \frac{|\boldsymbol{u}_{h,err}|^2}{2|\boldsymbol{U}_h|} \qquad (19)$$

The wind speed error consists of a projection of the horizontal wind errors and a strictly positive term. The more closely the mean wind aligns with an axis, the more heavily the corresponding component error is weighted. We observed slightly wider error distributions in cross-stream velocity estimates compared to streamwise (figure A2). In that case, the potentially larger

error is weighted less heavily.

The squared terms account for the positive bias observed in the wind speed error distributions (figure 5) since it consistently shifts errors to be more positive. We can explicitly find a theoretical mean of the wind speed error (appendix C) and simplify by assuming the $u$ and $v$ error random variables have a mean of zero (although this assumption can break down, e.g. in the presence of coherent structures or due to the RWF near the surface).

$$\mu(|\boldsymbol{u}_h|_{err}) \approx \frac{\sigma^2(u_{err}) + \sigma^2(v_{err})}{2|\boldsymbol{U}_h|} \qquad (20)$$

Note that this does not mean the total error (equation 21) is always positive – the weighted component errors can cause it to be negative – but that on average the reported wind speed will be greater than the truth. The bias should be less severe the smaller the errors in the $u$ and $v$ wind component estimates and the higher the wind speed. The difference in the variance of the $u$ and $v$ errors across the stability regimes along with the respective wind speeds accounts for the degree of bias observed between

the cases, most significant for the strong CBL and least for the stable BL.





Without explicitly computing the variance forms, we can estimate the error magnitude. The wind speed error should generally be on the order of the individual component errors (i.e. their standard deviation), though the bias term term has the potential to become more prominent in adverse conditions (large $u, v$ errors, slow winds). The standard deviations of the wind speed error from the virtual lidar data (figure 5) do match up with the those of the $u$ and $v$ error distributions, maintaining
respective height trends in each of the stability cases as described in the previous section.

Now consider the wind direction error. To simplify the analysis, we set aside the quadrant correction and consider just the traditional inverse tangent function to find the angle in $[-90°, 90°]$. Then the wind direction error, in radians, is the difference,

$$\Theta_{err} = \Theta_{lidar} - \Theta = \arctan\left(\frac{u_{lidar}}{v_{lidar}}\right) - \arctan\left(\frac{U}{V}\right)$$

As with the wind speed, the mean value does not directly cancel. Applying the difference identity for $\arctan$ and simplifying,

$$\Theta_{err} = \arctan\left(\frac{V u_{err} - U v_{err}}{|\boldsymbol{U}_h|^2 + V v_{err} + U u_{err}}\right) \tag{21}$$

Turn this into looser bound that is simpler to interpret. The derivative of $\arctan$ is continuous and bound above by one so that we may bound, $|\arctan(x)| \leq |x|$. For the error,

$$|\Theta_{err}| \leq \left|\frac{V u_{err} - U v_{err}}{|\boldsymbol{U}_h|^2 + V v_{err} + U u_{err}}\right| = \left|\frac{|\boldsymbol{u}_{h,err}|\sin(\theta_{\boldsymbol{U}_h, \boldsymbol{u}_{h,err}})}{|\boldsymbol{U}_h| + \frac{\boldsymbol{U}_h}{|\boldsymbol{U}_h|} \cdot \boldsymbol{u}_{h,err}}\right| \tag{22}$$

The bound is tighter the smaller the error and the sign of the bounding expression should match that of the error creating
an 'envelope' for the error. As opposed to the wind speed, in the wind direction error form, the mean velocity components are cross-multiplied with the velocity component errors. Depending on the orientation of the wind, one of the terms in the numerator is more likely to dominate: the stronger wind component multiplied with the larger cross-stream error.

Under the error approximation, we can similarly estimate the mean error (equation 23). If we again assume the $u$ and $v$ reconstruction errors to have zero mean, then the wind direction mean error should also be zero.

$$\mu(\Theta_{h,err}) \approx \frac{1}{|\boldsymbol{U}_h|}\left(\sin(\Theta)\mu(v_{err}) - \cos(\Theta)\mu(u_{err})\right) \tag{23}$$

As seen in the strong CBL (figure 5), the approximation can break down in significant ways, introducing bias and listing. Patterns in coherent structures were identified as a likely cause of non-zero means in $u$ and $v$, but the magnitude of the biases in the strong CBL resemble those in the stable BL at some heights. The combination in the strong convection case of the coherent structures and the weak winds allows the non-zero means in $u$ and $v$ to be amplified to create the wind direction error
bias. The stable BL, on the other hand, tempers the effect by the strength of the wind speeds. If the lidar ensemble included instruments rotated at offsets spanning a full $360°$ rather than $45°$, the signs of the errors would cancel (flipping the signs of $u$ and $v$ in the lidar coordinate system flips the signs of the error). Within the ensemble of offsets computed with the virtual instrument, however, the signs are consistent and the bias persists across the rotated lidar measurements. For measurements made in relatively steady, slow winds it cannot be expected that bias will not emerge as it does in the strong CBL data.





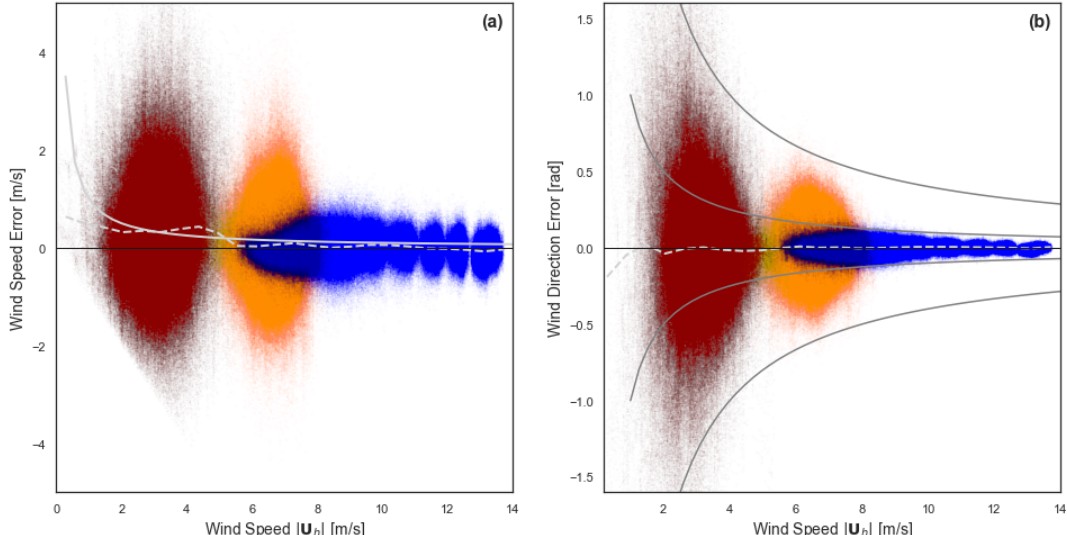

**Figure 14.** Trends in (a) wind speed and (b) wind direction errors with respect to the volume-averaged wind speed for the observation. Data points are colored by stability case: strong CBL (red), weak CBL (orange), and stable BL (blue). Reference lines are given in (a) for the mean scatter error at each wind speed (dashed) and $1/|\boldsymbol{U}_H|$ decay (solid). In (b), reference lines trace the $1/|\boldsymbol{U}_H|$ and $4/|\boldsymbol{U}_H|$ envelopes.

### 4.4.1 Effects of wind speed and orientation angle

The idea that lidar might manifest smaller error at higher winds seems intuitive. On top of potential implicit effects on the homogeneity over the scan volume, the derived error forms draw out explicit dependencies on wind speed and direction. The trends predicted by the analytic form convincingly describe those observed in the virtual lidar data. We find that wind speed powerfully influences error in the lidar observations, while the orientation of the lidar with respect to the mean wind has negligible impact.

The wind speed trends in the virtual lidar data align with theoretical assessments (figure 14). The wind speed error magnitudes visibly contract from the slow, strong CBL data to the fast winds of the stable BL; the decrease has to do with the identified mechanisms of driving the error variance in each of the stability regimes. A trend line for the mean error in the wind speed measurement is computed with respect to the true volume averaged wind speed. It exhibits a positive bias that diminishes rapidly with increasing wind speed. Variations in the magnitude of the $u$ and $v$ error variances in the numerator as well as contributions from their deviation from zero means introduce noise on the predicted $1/(2|\boldsymbol{U}_h|)$ (equation 19) decay.

According to the wind direction error form (21), we expect the magnitude of the error to decay on the whole with at least $1/|\boldsymbol{U}_h|$. Indeed, allowing for potentially different scales of $u$ and $v$ error in the numerator, the data falls nicely along the reference envelopes, especially along the tail of the stable BL. The additional decay is likely accountable to the inverse tangent which curtails the size of the largest errors compared to the bounding estimate as well as improved correlation at the heights with the strongest winds reducing the component errors.





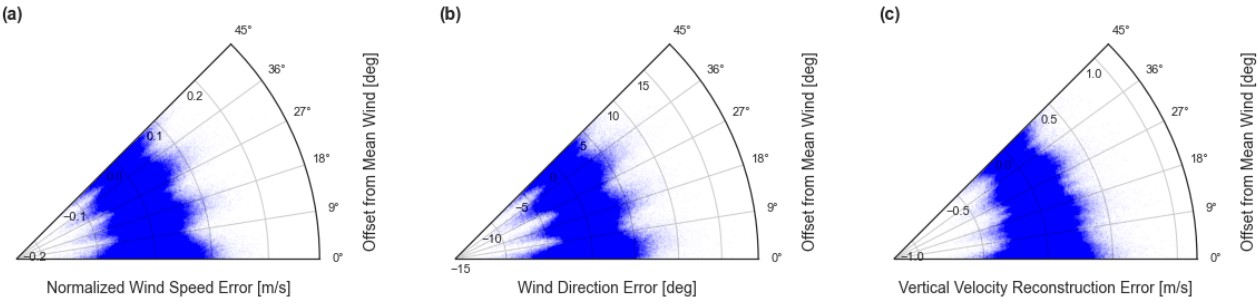

**Figure 15.** Errors in (a) wind speed, (b) wind direction, and (c) vertical velocity in the stable BL case plotted against the offset angle between the wind direction and the nearest lidar axis. Data clusters around the offsets for the discrete offset angles tested.

Potential differences in error as a function of the lidar orientation with respect to the mean wind direction come from slight differences in the weighting in implicit differences in correlation effects in the streamwise and cross-stream directions. In the virtual lidar error data, only small discrepancies could be distinguished between errors in the stream-wise and cross-stream velocity component estimates, likely due to the relatively poorer correlation in the cross-stream velocity component. Comparing the disaggregated rotated lidar ensembles (figure A2), a slight amplification in error variance shifts from one component to the other depending on which has a smaller projection of the prevalent wind.

In practice, changes in orientation produce a negligible effect on the error which is washed out by other, more significant sources. (figure 15). A small improvement in the error spread for $|\boldsymbol{u}_H|, \Theta$ and $w$ might be discerned when the lidar axes are closer to an even $45°$ offset from the prevailing wind (in the stable case; this variability erodes completely in the convective cases) which makes the relative error between the horizontal wind components more consistent. The trend with offset angle in the vertical velocity error may be because of a combination of the correlation and the full, even incorporation of all four beams in the reconstruction (equation 9), each with similar error distribution characteristics. None of the trends are particularly significant, however. Comparing moments across the rotated ensembles (figure A2) confirms that there is little meaningful difference in the error behavior between offset angles (with some exception for peak variance of the stable BL vertical velocity error).

It should be noted that although no strong trends were found with respect to the relative offset between the mean wind and the lidar axes, the signs of some of the biases can change with the signs of $u$ and $v$. The derived error forms for the wind speed and direction and discussion of the RWF (section D) explain these dependencies.

## 4.5 Reducing error through time averaging

Time averaging reduces the 'noise' of the error in the raw, high frequency measurements made by the lidar, leaving a more reliable mean measurement. Under conditions in which the background flow continues to evolve in time, the utility of time averaging must be weighed against the length of the interval during which 'quasi-stationary' conditions exist and sacrificing



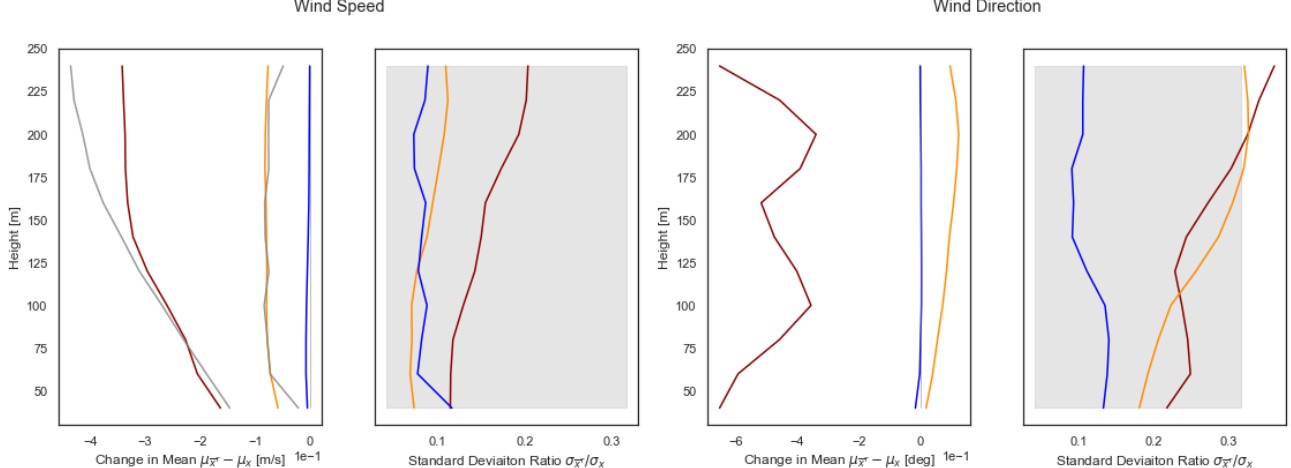

**Figure 16.** Change in time-average error distribution compared to the error distribution of the 1 Hz samples for wind speed and direction. Left panels show shift in the distribution means $\mu(u_{err}) - \mu(\overline{u}_{err}^T)$ and right panels show the ratio of the standard deviation of the time-averaged error distribution to the original $\sigma(\overline{u}_{err}^T)/\sigma(u_{err})$. Colors indicate stability case: red (strong CBL), orange (weak CBL), and blue (stable BL). Expected ranges are demarcated by the shaded box.

the resolution of shorter time-scale dynamics. Making an informed assessment of an appropriate time window length rests on

quantifying the expectation of the improvement of the measurement.

The lidar error varies along with the 'random' turbulence in the flow which we have reflected by describing the error as a random variable drawn from a distribution dependent on the character of the turbulence and the lidar scan geometry. In preceding analysis we considered how the physical and mathematical mechanisms might influence the distribution of the error in the raw lidar measurement. Now we consider how time averaging acts on the full error distribution.

Time averages were performed over the wind components (which is mathematically equivalent to averaging over the beam radial velocities when the reconstruction is linear with respect to them as in equation 8). Let $T$ be the length of time window average and suppose the instrument samples at a constant interval of $\tau_s$ (every second for the WindcubeV2). Then there are $T_s = \lfloor T/\tau_s \rfloor$ samples in the discrete average over the window. Assume quasi-stationary conditions so that the volume average at one any given time is close to the time averaged value, e.g. $U \approx \overline{U}^T$.

Start with the effect of the average on the error in the wind component reconstructions. The following derivations are given in terms of $u$, but hold identically for all the wind components. Expanding the lidar estimate of the time-averaged truth, $\overline{U}^T$, the error in the time averaged measurement emerges as the arithmetic mean of each of the sample errors (equation 24)

$$\overline{u}_{lidar}^T = \frac{1}{T_s}\sum_{t=1}^{T_s} u_{lidar}(t) = \frac{1}{T_s}\sum_{t=1}^{T_s}[U(t) + u_{err}(t)] = \overline{U}^T + \frac{1}{T_s}\sum_{t=1}^{T_s} u_{err}(t) \implies \overline{u}_{err}^T = \frac{1}{T_s}\sum_{t=1}^{T_s} u_{err}(t) \quad (24)$$

That is, the error in the time-averaged measurement can be expressed as the sum of (scaled) random variables drawn from the

distribution of the original sample errors. Using linearity, the mean of the time-averaged error distribution, $\overline{u}_{err}^T$, is equal to





the mean for the original sample errors; i.e. time averaging does not change the mean ensemble error of the wind components. The result holds for any linear reconstruction. The mean error in individual measurements may benefit from time averaging, however, as the time average approaches the potentially less biased 'ensemble' error from the selective sub-sample (e.g. figure 7).

The time average does, however, reduce the error spread, as observed in the virtual lidar data. The variance of a sum of independent, identically distributed variables is well known; the variance of the original random variable (the un-averaged error) is scaled by one over the number of samples in the average. In a time series, the samples cannot simply be treated as independent since subsequent samples are highly correlated. In the lidar, the pattern of turbulent structures that gave rise to a particular error continue to influence the error in the following samples as well so that the errors are quite similar. The reduction

factor on the error standard deviation may be estimated by a range. The lower bound is given by treating all of the $T_s$ samples in the time average as independent, even though they are not. A rough upper estimate can be formed by simply discarding the highly-correlated data. Let $\tau_c$ be the decorrelation time for the time series (which should be similar to the decorrelation time for the winds themselves). A time average computed only with $T_c$ samples spaced at least $\tau_c$ apart from one another can safely treat the samples as independent. The combination of the estimates provides a range of factors by which we can expect

to reduce the error standard deviation in the time-averaged estimate of the wind components (equation 25)

$$(T/\tau_s)^{-1/2} \approx T_s^{-1/2} \le \frac{\sigma_{\overline{u}_{err}^T}}{\sigma_{u_{err}}} \lesssim T_c^{-1/2} \approx (T/\tau_c)^{-1/2} \tag{25}$$

The standard deviation reduction factor (assuming $\tau_s, \tau_c$ fixed) scales proportionally to $T^{-1/2}$. The marginal utility of longer time average in terms of reducing the standard deviation shrinks rapidly; just four independent samples are needed to halve the standard deviation but 100 are needed to bring the standard deviation to a tenth of its original value.

The error in the derived quantities of wind speed and direction were derived in terms of the wind component errors. For the time-averaged quantities, we may simply use the modified component distributions, which we determined above had (1) no change in mean and (2) scaled variance.

    The wind speed error experiences an improvement not only in the spread of the error but also in the mitigation of the positive bias. In the mean, the first terms in the wind speed error (19) remain unaltered, but the positive bias term is proportional to the

sum of the $u$ and $v$ error variances and is accordingly scaled by the factor $(T/\tau_c)^{-1}$. Though the mean estimates of the wind components themselves do not improve under the time average, the reduction in the spread of their errors leads to a marked improvement in the mean error of the wind speed. The variance for the wind speed was not explicitly computed but estimated to be roughly equal to a weighted combination of the $u$ and $v$ error variances. The reduction factor for the wind speed variance (and thus standard deviation) is about that of the wind components. Using the rough approximation for the decorrelation time,

the projections for the change in the time averaged behaviors hold up against the virtual lidar data (figure 16).

    The mean wind direction error should experience negligible change under the time average since it arises from the component error means. The estimate of the error magnitudes project it to be proportional to the component standard deviation over the mean horizontal wind speed. Since we assume the wind speed to be consistent through the time window, the wind direction





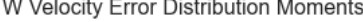

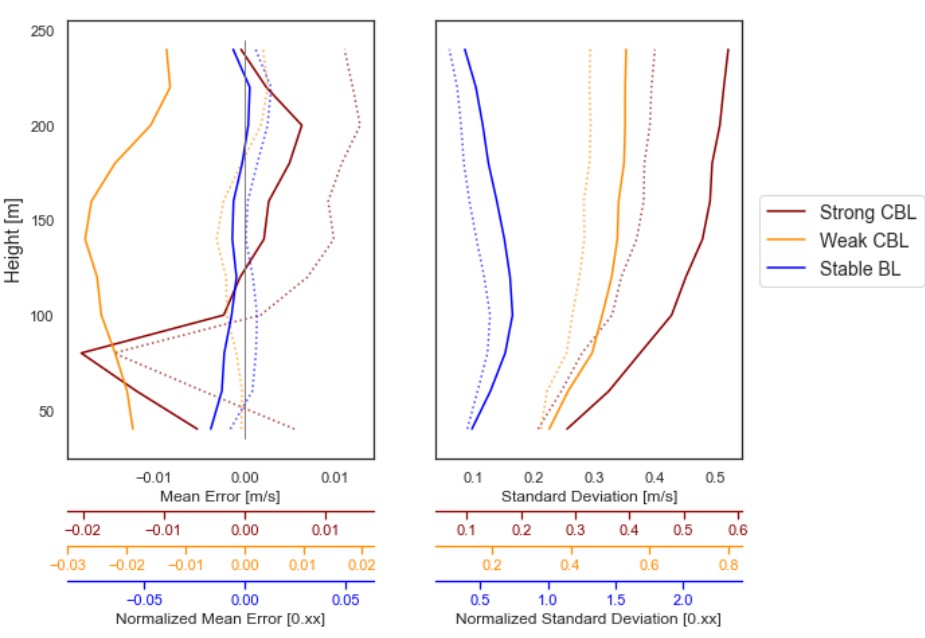

**Figure 17.** Comparison of the error in vertical velocity measurement using the evenly weighted vertical velocity reconstruction (equation 8), dotted line) and the wind-direction weighted reconstruction (equation 9), solid line). Both use the volume-average as truth.

standard deviation should also be reduced by the same factor as the wind components, $(T/\tau_c)^{-1/2}$. Both predictions hold up

in the empirical results (figure 16).

### 4.6 Note on vertical velocity

The vertical velocity measurement demands separate treatment from the horizontal winds. The vertical velocity itself behaves distinctly from the horizontal winds: it varies much more rapidly and the the features of interest occur at smaller spatial and temporal scales, the background signal being close to zero. The WindcubeV2 offers two possibilities to reconstruct the vertical

velocity from the measured radial velocities by either equally weighting the beams or using the wind direction to selectively weight them (equations 8,9). The measurement from the vertical beam is a third option that samples just the vertical velocity and does not require reconstruction.

  Since the variation in the vertical velocity generally occurs at a smaller scale than the scan volume, the most interesting information will tend to be lost in the reconstruction process. Even if the volume average were perfectly recovered, the average

itself loses information. The error induced in the different sensing cases were separately examined from the virtual instrument data. Figure 17 compares the two reconstruction techniques (equations 8 and 9). The equally-weighted reconstruction was examined in depth with the analytic break down of the error; here we empirically consider the effect of instead using the wind direction weighting.





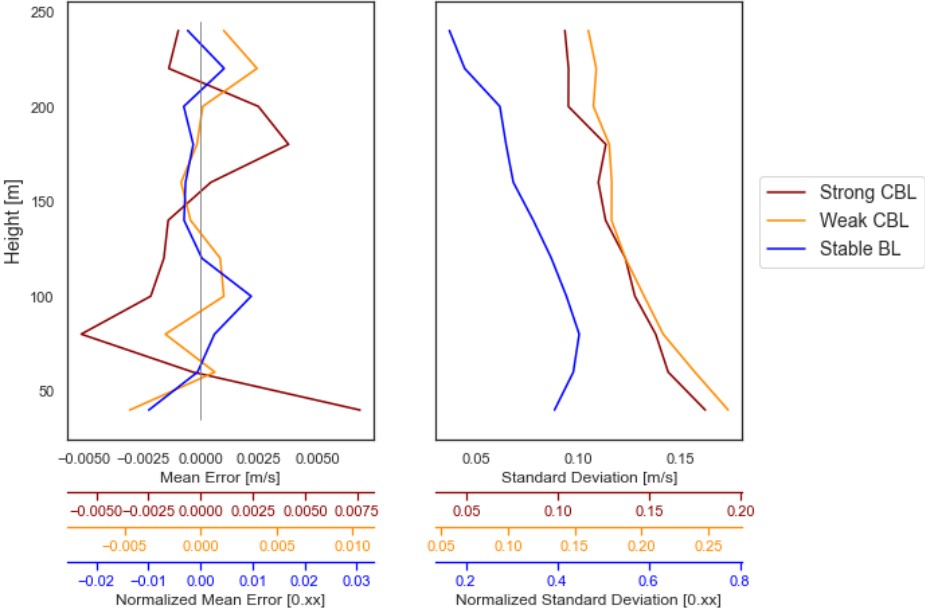

**Figure 18.** Vertical velocity measurement error moments for the vertically pointed beam. Error is with respect to the pointwise truth.

At least with respect to the disk-averaged truth, the lessened dependence on the full four beams using wind direction weighting seems to outweigh the beneficial effects. When the wind is directed between the lidar axes (45 degree angle), the two reconstructions are identical; the difference arises when the wind direction lies more closely with one of the axes so that two beams are weighted more heavily, shrinking the contribution from the other two beams. In our test cases, the reconstruction relies primarily on the east and west beams. With respect to the volume-averaged vertical velocity, the bias and standard deviation of the error is on the whole larger under the reconstruction using wind direction weights than with equally weighted beams (figure 17). In the context of the random variable error model (equation 13), weakening or removing dependence on some of the beams removes the chance for cancellation of the perturbations; the mean over two points will tend to be a poorer representation of the volume mean than that over four points. The concentrated dependence can also magnify the influence of variations experienced the two beam locations. In light of the empirical error behavior, the benefits of incorporating more points seems to supersede the benefits of streamwise correlations.

The vertical beam sidesteps the implicit volume average as well as the need for any reconstruction. In this case, the error incurred in measuring a pointwise vertical velocity arises purely from the effects of the range-gate-weighting in the measurement process. The RWF produces errors with magnitude and character that are distinct from the reconstruction errors (figure 18). Across stability cases, the bias shrinks to be negligible. The difference in standard deviation between the convective cases vanishes and the overall magnitude is significantly reduced compared to the error in the reconstruction. As opposed to the error in the reconstruction estimate, the standard deviation with only the RWF decreases with height. In the stable case, the RWF was a larger portion of the error in the vertical velocity reconstruction (figure 11). Under stable conditions, the difference between





the reconstruction and the vertical beam is less dramatic; the standard deviation magnitudes are about halved and the decrease in the standard deviation with height exists with both but is more pronounced with the vertical beam. Even using the vertical beam, the standard deviation can still be relatively quite large compared to typical values.

## 770  5  Conclusions

Atmospheric variability inextricably influences error in wind lidar measurements. By using virtual instruments acting on LES flows, error mechanisms can be isolated and explicitly tracked and analyzed to better understand the error behavior as a whole. In this study, we considered profiling lidar measurements in quasi-stationary conditions. Even in the absence of explicit sources of inhomogeneity, observation error can arise, tightly coupled to the character of turbulence in the flow. The error distributions

of a virtual WindcubeV2 lidar were estimated from ensembles of virtual lidar run in uniform, ideal WRF-LES scenarios in convective and stable boundary layer regimes. An analytic error model leverages random variable representations to describe how the turbulent variability propagates into the lidar error, decomposing the contributions from velocity perturbations at each beam from the volume-average and from deviations in the point measurement due to range-gate weighting.

The quantification of the error aligns with values found in field studies (e.g. Courtney et al. (2008)) and in similar estimates

of virtual lidar performance in baseline convective conditions (Rahlves et al., 2021). To define the error, we compared virtual measurements against both a pointwise truth reference and the average over the scan volume. Under ten-minute averages in the quasi-stationary conditions of the test cases, the distinction between the kinds of spatio-temporal averages fades and the two error distributions seem to converge. The magnitudes of the overall errors in the virtual measurements fall generally within experimentally determined ranges in favorable conditions: ten-minute averaged wind speeds have mean errors within $\pm 0.2 m/s$

with standard deviation $< 0.3 m/s$ and wind direction have mean error bias within $2°$ and standard deviations in $2.5°$ except for in the strongly convective case where it can reach up to $12°$.

The character of the error in the reconstructed wind vector components is driven by the form of the turbulence, so that the lidar accuracy is dependent on the flow regime and vertical structure of the boundary layer. This derivation explains findings from other sensitivity studies (Klaas and Emeis, 2021; Rahlves et al., 2021) in that unstable conditions are prone

to larger error than stable conditions. The error variances were connected to weighted, spatially filtered turbulent variances in the horizontal and vertical velocities. The vertical velocity variances are of particular importance. Since the variations in vertical velocities tend to occur on smaller spatial scales, fewer of these variations are filtered out by the lidar scan volume scale. The resulting velocity perturbation variance is then weighted more heavily in the error compared to horizontal perturbations, i.e. the 'projection' of the vertical velocity variations is greater on a more narrow scan cone. In convective conditions, the compounding

mechanisms lead to vertical velocity perturbations almost single-handedly accounting for the error. Under stable conditions, the error magnitudes are notably smaller and result from a more balanced interplay of the vertical and horizontal velocity inhomogeneities. Height trends in the error arise from the interaction of the changing filtering scale and the vertical structure of the turbulence in the boundary layer. The magnitudes grow with the increasing vertical velocity variance up into the middle of the convective boundary layers. The shallow stable boundary layer is fully encompassed in the lidar range and the error





trends strongly with its profile; the error magnitudes are maximized in the mid-boundary layer where the filtered turbulence
is strongest. Previous work has investigated the relationship of lidar error with aggregate turbulence intensity (Courtney et al.,
2014). Our findings reflect the connection, with additional separation of the turbulent fluctuations into each of the components
to allow for the difference in spatial scales and weighting to come through.

The range-gate weighting in the radial velocity measurement has minimal relative effect on the total lidar error except for
in high-shear regions near the surface layer. Deviations incurred in the radial velocity measurement by the weighted volume-
average along the beam should vanish under constant gradients but can grow in the presence of large second derivatives in the
radial velocity projection along the beam. For the most part, the impact of the larger variations over the scan volume dominate
any RWF effects. In the bottom few range gates near the surface, however, the virtual lidar data reflect a prominent interaction
of the RWF with shear near the surface layer leading to measurement bias. The persistent curvature of an approximately
logarithmic profile results in significant (around $0.6 m/s$ in the stable case) and consistent underestimation of the magnitude of
along-wind horizontal component(s). Our findings are consistent with previous studies that have identified the key interaction
of the RWF with shear resulting in error bias (e.g. Lindelöw et al. (2008) and Clive (2008) and Courtney et al. (2014)).

Within the class of DBS/VAD profiling scans, any control over the reconstruction error comes from adjusting the cone
angle, $\phi$, and the number and spacing of the scan azimuthal angles, $\{\gamma_i\}$. The virtual instrument tests in our study held these
parameters fixed to match the WindcubeV2, but other studies have explicitly investigated the sensitivity of the lidar error with
respect to the scan configuration. The decomposition of the error in our study according to the derived error form, however,
allows for insights into the relative impacts of the configuration choices on the error behavior.

The cone angle determines the degree of projection of the horizontal and vertical perturbations (manifest in the weighting
in the error form) as well as the spatial separation of the sampling beams. In the strong and weak CBL test cases in particular,
the error demonstrates strong adverse impacts of resulting heavy weighting on the vertical perturbations. The dominance of
the vertical perturbation terms can be tempered by reducing the elevation angle. Teschke and Lehmann (2017) proposed a
shallow elevation angle ($\phi \approx 35.26°$) based on an analytic minimization of reconstruction error in a horizontally homogeneous
and stationary wind field. Rahlves et al. (2021) tested the low elevation angle with a virtual instrument in quasi-stationary
convective conditions with favorable results compared to more typical, larger elevation angles. Improvements achieved by
reducing the angle to lessen the weighting on the vertical velocity perturbations may be offset by the corresponding effects of
increasing in the separation of the beams. In the presence of background flow gradients, larger separation can induce greater
mean error, as explored in terms of linear variations of $w'$ in Bingöl et al. (2009). In quasi-stationary cases without background
gradients, the effect of the scan radius is primarily one of filtering the turbulence in the flow. Only a (high-pass) filtered portion
of the full turbulent velocity variance is reflected in the variance of the velocity perturbations from the volume average which
appear in the error form. Decreasing the elevation angle lengthens the filter scale, allowing for larger error variances, up to
a cap determined by the full, unfiltered turbulence. We propose that the competing effects of the cone angle lead to different
optimal angles depending on the flow conditions and that reasonable estimates can be performed to minimize error if rough
numbers for any gradients and the magnitudes of the turbulent variances / spectra are known.





Some profiling scans use a different number of off-vertical beams to diagnose the mean winds. The beams are usually
preferred to be symmetrically spaced to remove potential bias (Sathe et al., 2015; Teschke and Lehmann, 2017). Common
reconstructions of the three-dimensional velocity fit sinusoids to the radial velocity measurements using a least squares process
(Newsom et al., 2015)). The result is a linear operation on the beam radial velocities of which the DBS scan presented in this
study is a special case. The error in these alternative profiling scans should take on a similar error form to that derived here,
with perturbations averaged over a greater number of beams. Under a simple stochastic model, Teschke and Lehmann (2017)
showed the standard deviation of the error should be proportional to $N^{-1/2}$ with $N$ the number of beams. The form our error
model also suggests that a greater number of independent samples in the scan volume should help reduce error. We do not
explicitly account for the time required to complete the scan, however, which increases with the number of beams. Rahlves
et al. (2021) tested scans using different numbers of beams without finding a universal trend in error.

The error in wind component reconstructions propagates into the error in the corresponding computation of horizontal wind
speed and direction. The error was formulated in terms of the $u$ and $v$ reconstruction errors. A systematic positive bias in the
wind speed estimate emerges from a strictly positive term that scales with $u$ and $v$ error variance and inversely with wind
speed, which is corroborated by the virtual lidar data. The standard deviation of the wind speed error is estimated to be on
par with the $u$ and $v$ error standard deviations. The form of the wind speed error derived here is similar to that in Rosenbusch
et al. (2021) (but only expanded to first order in the wind speed error), with similar consequences. The leading order terms are
the same and a positive bias of the same scale results. The formulation used here relies on the wind component errors rather
than directly using $u'$ and $v'$ perturbations from the mean horizontal wind. The $u$ and $v$ component errors behave differently
than just turbulent perturbations, e.g. incorporating vertical velocity variations as shown in our analysis. These findings of a
systematic overestimation do not contradict the mechanisms of possible wind speed underestimation studied in Bingöl et al.
(2009) which arise from specific gradient in the flow inducing an underestimation in the first order terms.

The wind direction has no explicit bias except that arising from the $u$ and $v$ reconstructions. The standard deviation is
roughly that of the $u$ and $v$ errors over the average wind speed (i.e. the error magnitudes are reduced at higher winds). The
observed error magnitudes strongly depend on mean wind speeds (especially the wind direction) but are only weakly related
to the relative orientation of the lidar. As in Rahlves et al. (2021), our results suggest no predominant direction of the random
$(u_{err}, v_{err})$ vector.

Individual measurements can suffer from large error which can be reduced through time averaging. While time averaging
cannot correct for biases in the wind component measurements, the standard deviations of the error are reduced by a factor
proportional to $T^{-1/2}$. The longer the decorrelation time in the error time series, the less the reduction. The use of time
averaging on wind speed measurements has elicited particular interest; Rosenbusch et al. (2021) examines both scalar and
vector averaging of the winds and the different behaviors presented. We recomputed the vector average of the wind speed and
direction using the time-averaged $u$ and $v$ horizontal wind components. Although the bias in the wind component estimations
cannot be improved, the systematic overestimation bias in the wind speed is reduced through vector time averaging (by a factor
proportional to $T^{-1}$).





Vertical velocity, with features of interest existing on smaller spatial and temporal scales, is a greater challenge to measurements by lidar. A vertically pointing beam omits the need for reconstruction or the implicit large scale spatial average over the
scan volume. Instead, only the smaller-scale averaging from the range-gate is applied. The errors associated with the vertical beam with respect to the pointwise values are significantly smaller and represent a more useful value that captures more of the small-scale variability in $w$.

Fully leveraging the access to the flow field afforded by an LES model, virtual lidar tools allow for not only predicting instrument error but also for separating and analyzing potentially competing mechanisms which rise to the error. The results
would benefit from a comparison with field data from an actual Windcube instrument and investigation of ways to identify the mechanisms and possible behavior of error in the data. For specifically targeted quantities/heights, optimizations of the scan using knowledge of likely mechanisms should be tested to confirm expected behaviors. Working from this baseline study, additional complications to the flow field could be introduced, e.g. complex terrain, heterogeneous flows like turbine wakes or canopy flows, or deployment of lidars on moving platforms such as ships, buoys, vans, or aircraft.

**6 Code and data availability**

Virtual lidar code may be found at https://gitlab.com/raro0632/virtual-lidar. Ensemble data collected from the virtual lidar for the LES test cases analyzed here are archived at https://doi.org/10.5281/zenodo.6112629.





## Appendix A: Error distribution moments with orientation disaggregation

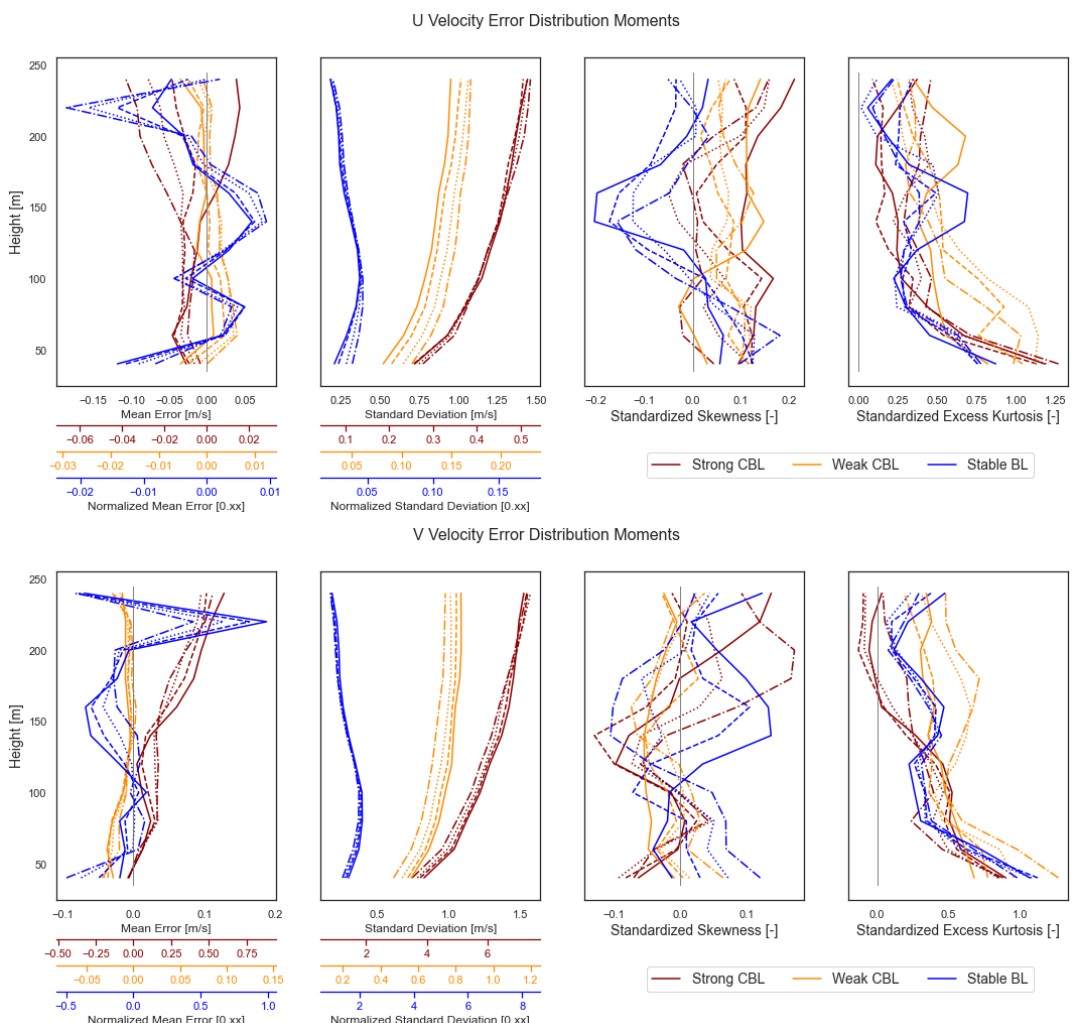

**Figure A1.** Comparison of $u$ and $v$ error distribution moments over disaggregated lidar orientation angles. No offset, same axes as LES domain (solid line), rotated $15°$ CCW from LES domain axes (dashed), rotated $30°$ CCW (dotted), and $45°$ CCW (dash-dot).

## Appendix B: Sample-based estimates of statistical moments

For a random variable $X$ which has been sampled with observed values $\{x_i\}$, the definitions used for the unbiased estimators of the full population/distribution moments are as follows.





**Figure A2.** Comparison of error distribution moments over disaggregated lidar orientation angles. No offset, same axes as LES domain (solid line), rotated 15° CCW from LES domain axes (dashed), rotated 30° CCW (dotted), and 45° CCW (dash-dot).





$$\text{(Mean)} \quad \mu \approx \frac{1}{N}\sum_{i=1}^{N} x_i$$

$$\text{(Centered Variance)} \quad \sigma^2 \approx s^2 = \frac{1}{N-1}\sum_{i=1}^{N}(x_i-\mu)^2$$

$$\text{(Standardized Skewness)} \quad \hat{\mu}_3 \approx \hat{m}_3 = \frac{N}{(N-1)(N-2)}\sum_{i=1}^{N}\left(\frac{x_i-\mu}{s}\right)^3$$

$$\text{(Standardized Excess Kurtosis)} \quad \hat{\mu}_4 \approx \hat{m}_4 = \frac{N(N+1)}{(N-1)(N-2)(N-3)}\sum_{i=1}^{N}\left(\frac{x_i-\mu}{s}\right)^4 - \frac{3(N-1)^2}{(N-2)(N-3)}$$

## Appendix C: Derivation of wind speed and direction error means

The random variable equation for wind speed error was derived in equation 19. Compute the expected value. We can further simplify if the component errors are assumed to have zero mean.

$$\mu(|\boldsymbol{U}_h|_{err}) = \mu\left[\frac{U}{|\boldsymbol{U}_h|}u_{err} + \frac{V}{|\boldsymbol{U}_h|}v_{err} + \frac{u_{err}^2 + v_{err}^2}{2|\boldsymbol{U}_h|}\right]$$

$$= \frac{U}{|\boldsymbol{U}_h|}\mu(u_{err}) + \frac{V}{|\boldsymbol{U}_h|}\mu(v_{err}) + \frac{\sigma^2(u_{err}) + \mu^2(u_{err}) + \sigma^2(v_{err}) + \mu^2(v_{err})}{2|\boldsymbol{U}_h|}$$

$$\approx \frac{\sigma^2(u_{err}) + \sigma^2(v_{err})}{2|\boldsymbol{U}_h|}$$

The approximated wind direction function 22 is the ratio of two random variables, each of which is a linear function of the component errors. Compute the expected value of the approximated form.

$$\mu(\Theta_{h,err}) \approx \mu\left(\frac{Vu_{err} - Uv_{err}}{|\boldsymbol{U}_h|^2 + Uu_{err} + Vv_{err}}\right)$$

$$= \mu(Vu_{err} - Uv_{err})\mu\left(\frac{1}{|\boldsymbol{U}_h|^2 + Uu_{err} + Vv_{err}}\right)$$

$$= [V\mu(u_{err}) - U\mu(v_{err})]\frac{1}{|\boldsymbol{U}_h|^2}\mu\left[1 - \left(\frac{U}{|\boldsymbol{U}_h|^2}u_{err} + \frac{V}{|\boldsymbol{U}_h|^2}v_{err}\right) + \left(\frac{U}{|\boldsymbol{U}_h|^2}u_{err} + \frac{V}{|\boldsymbol{U}_h|^2}v_{err}\right)^2 - \cdots\right]$$

$$\approx [V\mu(u_{err}) - U\mu(v_{err})]\frac{1}{|\boldsymbol{U}_h|^2}$$

$$= \frac{1}{|\boldsymbol{U}_h|}\left(\cos(-\Theta)\mu(u_{err}) - \sin(-\Theta)\mu(v_{err})\right)$$

If the component errors have zero mean, then the wind direction error is zero. If they are nonzero, large wind speeds can temper

the bias and high wind speeds compound it.



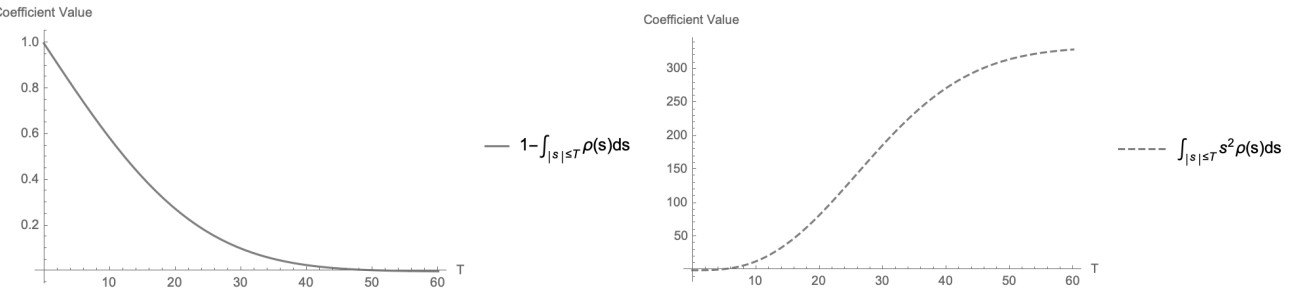

**Figure D1.** Growth and decay of the coefficients in the bounding terms of the radial velocity measurement error with increasing threshold distance $T$. The WindcubeV2 RWF (equation 6 with parameters in table 2) is used for illustration.

## Appendix D: Derivation of error bound on RWF weighted radial velocity measurement

Let $v_r(r_0)$ be the actual radial velocity at radius $r_0$ and $\overline{v}_r(r_0)$ the observed, range-gate-weighted radial velocity centered at $r_0$. Let $T > 0$ be an arbitrary threshold to split the integral.

We'll assume the $v_r(s)$ profile has at least two continuous derivatives. The range gate weighting function, $\rho(s)$, should
generally be non-negative, symmetric, and satisfy $\int_{-\infty}^{\infty} \rho(s)ds = 1$ by definition so we assume these properties as well.

Using the triangle inequality, integral mean value theorem, and Taylor series expansion, we have the following derivation.

$$
|\overline{v}_r(r_0) - v_r(r_0)| = \left| \int_{-\infty}^{\infty} \rho(s)v_r(r_0 + s)ds - v_r(r_0) \right|
$$

$$
\leq \left| \int_{|s|>T} \rho(s)v_r(r_0 + s)ds \right| + \left| \int_{|s|\leq T} \rho(s)v_r(r_0 + s)ds - v_r(r_0) \right|
$$

$$
\leq \max_{s>|T|}|v_r(r_0 + s)| \int_{|s|>T} \rho(s)ds + \left| \int_{|s|\leq T} \rho(s)v_r(r_0 + s)ds - v_r(r_0) \right|
$$

$$
= \max_{s>|T|}|v_r(r_0 + s)| \int_{|s|>T} \rho(s)ds + \left| \int_{|s|\leq T} \rho(s)\left[ v_r(r_0) + v_r'(r_0)s + \frac{1}{2}v_r''(r_0 + \xi(s))s^2 \right] ds - v_r(r_0) \right|
$$

$$
= \max_{s>|T|}|v_r(r_0 + s)| \left[ 1 - \int_{|s|\leq T} \rho(s)ds \right] + \left| v_r(r_0)\left( \int_{|s|\leq T} \rho(s)ds - 1 \right) + v_r'(r_0)\int_{|s|\leq T} \rho(s)sds + \frac{1}{2}v_r''(r_0 + \xi_*)\int_{|s|\leq T} \rho(s)s^2ds \right|
$$

$$
\leq \left[ 1 - \int_{|s|\leq T} \rho(s)ds \right]\left( \max_{s>|T|}|v_r(r_0 + s)| + |v_r(r_0)| \right) + \left[ \frac{1}{2}\int_{|s|\leq T} \rho(s)s^2ds \right]|v_r''(r_0 + \xi_*)|
$$

Where we've introduced $\xi : [-T, T] \to [-T, T]$ as part of the Taylor remainder. The relative sizes of the coefficients with choice of $T$ in the WindcubeV2 RWF (equation 6) are plotted in figure D1.





**Appendix E: Numeric computation of RWF convolution**

The numeric computation of the range-gate-weighted radial velocity involves approximating a convolution integral for a continuous weighted average. The estimate should ideally maintain the weighted average nature of the operation to prevent, e.g., underestimating the result by virtue of only incorporating a sub-unity set of weights. For this reason, previous implementations (Forsting et al., 2017; Lundquist et al., 2015) have implemented the RWF convolution as a discrete weighted average using

re-normalized RWF values at the corresponding points.

$$\int_{-\infty}^{\infty} \rho(s)v_r(r+s)ds \approx \int_{-T}^{T} \rho(s)v_r(r+s)ds \approx \frac{1}{\sum_i \rho(s_i)} \sum_k \rho(s_k)v_r(s_k)$$

The quadrature form amounts to a re-scaled midpoint rule when the nodes ($\{s_k\}$) are equispaced; rewrite to show explicitly.

$$\sum_k \frac{\rho(s_k)}{\sum_i \rho(s_i)} v_r(s_k) = \sum_k \frac{h\rho(s_k)}{h\sum_i \rho(s_i)} v_r(s_k) = \frac{1}{\sum_i h\rho(s_i)} \sum_k h\rho(s_k)v_r(s_k)$$

For nodes that are not equispaced, it suggests that the variable interval width $h_k$ should be incorporated into the approximation

such that it is still a rescaled midpoint rule.

$$\int_{-\infty}^{\infty} \rho(s)v_r(r+s)ds \approx \frac{1}{\sum_i h_i\rho(s_i)} \sum_k h_k\rho(s_k)v_r(s_k)$$

First, we'll truncate the domain over which we try to estimate the integral to a finite interval. The omitted contribution from the ends of the infinite integration interval can be bound small using that fact that the the tails of the RWF must vanish in order for the infinite integral $\int_{-\infty}^{\infty} \rho(s)ds = 1$ to converge.

$$\left| \frac{1}{\sum_i h_i\rho(s_i)} \sum_k h_k\rho(s_k)v_r(r+s_k) - \int_{-\infty}^{\infty} \rho(s)v_r(r+s)ds \right|$$

$$\leq \left| \frac{1}{\sum_i h_i\rho(s_i)} \sum_k h_k\rho(s_k)v_r(r+s_k) - \int_{-T}^{T} \rho(s)v_r(r+s)ds \right| + \left| \int_{|s|>T} \rho(s)v_r(r+s)ds \right|$$

$$\leq \left| \frac{1}{\sum_i h_i\rho(s_i)} \sum_k h_k\rho(s_k)v_r(r+s_k) - \int_{-T}^{T} \rho(s)v_r(r+s)ds \right| + \left[ \int_{|s|>T} \rho(s)ds \right] \max_{|s|>T} |v(r+s)|$$





The error bound on the numeric quadrature will be derived based on the midpoint rule. Consider just one subinterval of length $h_k$ with midpoint at $s_k$.

$940$

$$\left| \frac{1}{\sum_i h_i \rho(s_i)} h_k \rho(s_k) v_r(r+s_k) - \int_{s_k-h_k/2}^{s_k+h_k/2} \rho(s) v_r(r+s) ds \right|$$

$$= \left| \frac{1}{\sum_i h_i \rho(s_i)} h_k \rho(s_k) v_r(r+s_k) - \int_{s_k-h_k/2}^{s_k+h_k/2} \left[ \rho(s_k) v_r(r+s_k) + (\rho(s_k) v_r(r+s_k))'(s-s_k) + \frac{1}{2} (\rho(\xi_k) v_r(r+\xi_k(s)))''(s-s_k)^2 \right] ds \right|$$

$$= \left| \frac{1}{\sum_i h_i \rho(s_i)} h_k \rho(s_k) v_r(r+s_k) - \int_{s_k-h_k/2}^{s_k+h_k/2} \left[ \rho(s_k) v_r(r+s_k) + \frac{1}{2} (\rho(\xi_k) v_r(r+\xi_k(s)))''(s-s_k)^2 \right] ds \right|$$

$$= \left| \frac{1}{\sum_i h_i \rho(s_i)} h_k \rho(s_k) v_r(r+s_k) - \rho(s_k) v_r(r+s_k) h_k - \frac{1}{2} \int_{s_k-h_k/2}^{s_k+h_k/2} (\rho(\xi_k) v_r(r+\xi_k(s)))''(s-s_k)^2 ds \right|$$

$$\leq \left| \left( \frac{1}{\sum_i h_i \rho(s_i)} - 1 \right) h_k \rho(s_k) v_r(r+s_k) \right| + \left| \frac{1}{2} \int_{s_k-h_k/2}^{s_k+h_k/2} (\rho(\xi_k) v_r(r+\xi_k(s)))'' s^2 ds \right|$$

$945$

$$\leq \left| \left( \frac{1}{\sum_i h_i \rho(s_i)} - 1 \right) h_k \rho(s_k) v_r(r+s_k) \right| + \frac{h_k^3}{24} \max_{|\xi_k^* - s_k| \leq h_k} \left| (\rho(\xi_k) v_r(r+\xi_k^*))'' \right|$$

Sum the error accumulated over all the intervals. If the subintervals partition the full interval $[-T, T]$, then:

$$\left| \frac{1}{\sum_i h \rho(s_i)} \sum_k h \rho(s_k) v_r(r+s_k) - \int_{-T}^{T} \rho(s) v_r(r+s) ds \right|$$

$$\leq \sum_k \left| \frac{1}{\sum_i h \rho(s_i)} h_k \rho(s_k) v_r(r+s_k) - \int_{s_k-h_k/2}^{s_k+h_k/2} \rho(s) v_r(r+s) ds \right|$$

$$\leq \sum_k \left[ \left| \left( \frac{1}{\sum_i h_i \rho(s_i)} - 1 \right) h_k \rho(s_k) v_r(r+s_k) \right| + \frac{h_k^3}{24} \max_{|\xi_k - s_k| \leq h_k} \left| (\rho(\xi_k) v_r(r+\xi_k))'' \right| \right]$$

$950$

$$\leq \left| \frac{1}{\sum_i h_i \rho(s_i)} - 1 \right| \sum_k h_k \rho(s_k) |v_r(r+s_k)| + \frac{\sum_k h_k^3}{24} \max_{|\xi| \leq T} \left| (\rho(\xi) v_r(r+\xi))'' \right|$$

$$\leq \left| \frac{1}{\sum_i h_i \rho(s_i)} - 1 \right| \max_k |v_r(r+s_k)| \sum_k h_k \rho(s_k) + \frac{\sum_k h_k^3}{24} \max_{|\xi| \leq T} \left| (\rho(\xi) v_r(r+\xi))'' \right|$$

$$\leq \left| 1 - \sum_k h_k \rho(s_k) \right| \max_k |v_r(r+s_k)| + \frac{\sum_k h_k^3}{24} \max_{|\xi| \leq T} \left| (\rho(\xi) v_r(r+\xi))'' \right|$$

All together, the error of the numeric approximation of the integral may be bounded by:

$$\left| \frac{\sum_k h_k \rho(s_k) v_r(r+s_k)}{\sum_i h_i \rho(s_i)} - \int_{-\infty}^{\infty} \rho(s) v_r(r+s) ds \right|$$

$955$

$$\leq \left| 1 - \sum_k h_k \rho(s_k) \right| \max_k |v_r(r+s_k)| + \frac{\sum_k h_k^3}{24} \max_{|\xi| \leq T} \left| (\rho(\xi) v_r(r+\xi))'' \right| + \left[ \int_{|s|>T} \rho(s) ds \right] \max_{|s|>T} |v(r+s)| \qquad \text{(E1)}$$



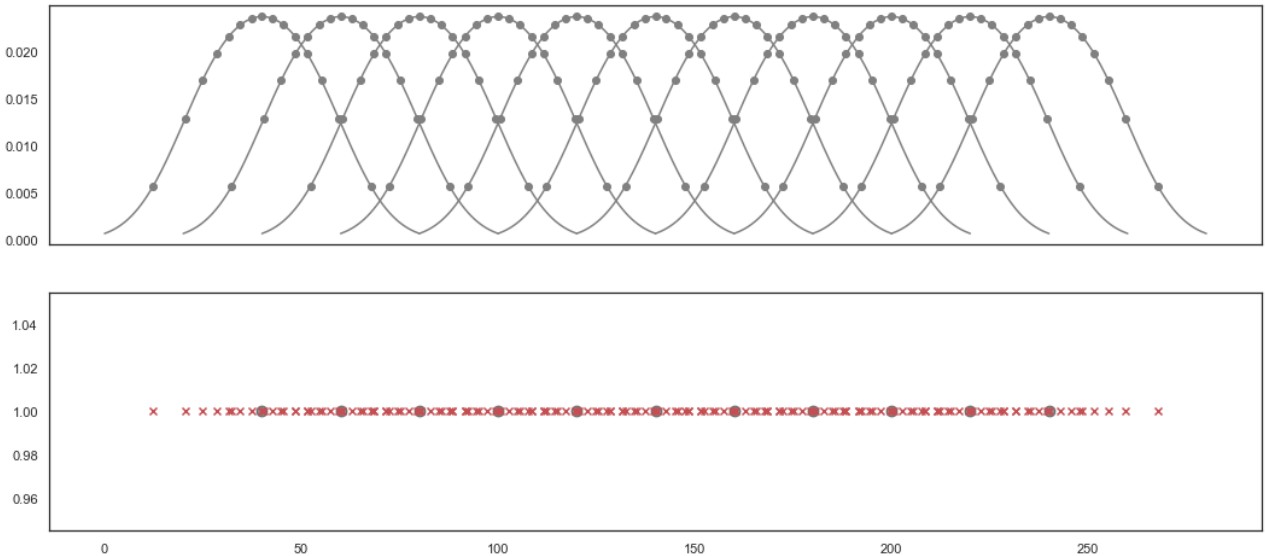

**Figure E1.** Using fifteen points per range gate, the node locations $\{s_k\}$ found by imposing $h_k\rho(s_k)$ to be constant for the WindcubeV2 RWF for as many nodes as possible. Top panel shows node locations on the overlapping RWF centered at each of the eleven range gates. Bottom panel shows the nodes (x's) and range gate centers (dots) along the one-dimensional beam length.

In addition to picking the interval $[-T, T]$ large enough so that the contribution from the tails is small and the usual second order error term from the midpoint rule, the points should be selected so that the midpoint approximation of integral of the RWF (i.e the first term) is close to one.

Choices for nodes include (1) equispaced (2) exponentially spaced (3) equal RWF area (4) equal midpoint area $h_k\rho(s_k)$.
Equispaced points are easy to determine and implement but do not maximize the utility of each point included. For a virtual lidar, which has to interpolate the winds for every node, inefficiently using points can be computationally expensive before the quadrature is even computed. The idea behind the other node distributions is to sample more heavily near the RWF center, where each point has greater impact on the result so that we reduce the number of 'low utility' nodes.

Exponentially spaced nodes (e.g. $\{-16, -8, -4, -2, -1, 0, 1, 2, 4, 16\}$) are easy to compute independent of the particular
RWF function like the equispaced nodes, but crudely achieves the goal of clustering points more heavily close to the RWF center. Forsting et al. (2017) recommend using nodes spaced so that the integral of the RWF between nodes is constant. The result is nodes clustered more heavily toward the center and more equal weighting on each node (not 'oversampling' the tails). Based on the midpoint form and error bound derived above, we propose an alternative in the same vein. We can explicitly set the $h_k\rho(s_k)$ to be constant so that every point has equal weight. Starting with $s_0 = 0$ and $h_0 = \frac{1}{N}$ where $N$ is the total (odd)
number of points, we can iteratively solve out for the symmetric nodes. In practice, the last nodes usually do not seem to be able to attain the same $h_k\rho(s_k) = 1/N$ weighting. Either the sum $\sum_k h_k\rho(s_k)$ must be reduced to less than one or the constant weighting relaxed for the end nodes. The latter is preferable to reduce the overall error.



Selecting nodes for computational expediency in the case of multiple range gates along a beam introduces further considerations than those for a single range gate. For the WindcubeV2 the intervals of dependency for the 20m-spaced range gates overlap (E1; nodes should be reused where possible rather than interpolating a separate set of points for each range gate. Methods using spacing by RWF area or imposing constant weighting are likely to produce unusual numbers that do not necessarily coincide between multiple range gates. Exponential spacing makes it easier to impose overlap in position. The cluster of points at the center of one range gate appear at the tail of another range gate and would have to be subselected to the desired points.

*Author contributions.* **Rachel Robey:** methodology, software, analysis and investigation, writing and editing **Julie K. Lundquist:** conceptualization, methodology, writing and editing

*Competing interests.* The authors declare that they have no conflict of interest.

*Disclaimer.* This report was prepared as an account of work sponsored by an agency of the United States Government. Neither the United States Government nor any agency thereof, nor any of their employees, makes any warranty, express or implied, or assumes any legal liability or responsibility for the accuracy, completeness, or usefulness of any information, apparatus, product, or process disclosed, or represents that its use would not infringe privately owned rights. Reference herein to any specific commercial product, process, or service by trade name, trademark, manufacturer, or otherwise does not necessarily constitute or imply its endorsement, recommendation, or favoring by the United States Government or any agency thereof. The views and opinions of authors expressed herein do not necessarily state or reflect those of the United States Government or any agency thereof.

This work was authored [in part] by the National Renewable Energy Laboratory, operated by Alliance for Sustainable Energy, LLC, for the U.S. Department of Energy (DOE) under Contract No. DE-AC36-08GO28308. Funding provided by the U.S. Department of Energy of Energy Efficiency and Renewable Energy Wind Energy Technologies Office. The views expressed in the article do not necessarily represent the views of the DOE or the U.S. Government. The U.S. Government retains and the the publisher, by accepting the article for publication, acknowledges that the U.S. Government retains a nonexclusive paid-up, irrevocable, worldwise license to publish or reproduce the published form of this work, or allow others to do so, for U.S. Government purposes.

*Acknowledgements.* Much thanks to Alex Rybchuk for use of his idealized convective boundary layer LES data, to Miguel Sanchez Gomez for his work to produce robust stable LES case runs, and to Raghavendra Krishnamurthy for his guidance with the WindcubeV2 velocity reconstruction.

This material is based upon work supported by the U.S. Department of Energy, Office of Science, Office of Advanced Scientific Computing Research, Department of Energy Computational Science Graduate Fellowship under Award Number DE-SC0021110.

This research has been supported by the US National Science Foundation (grand nos. AGS-1554055 and AGS-1565498).





We would like to acknowledge high-performance computing support from Cheyenne (doi:10.5065/D6RX99HX) provided by NCAR's Computational and Information Systems Laboratory, sponsored by the National Science Foundation.

This work utilized resources from the University of Colorado Boulder Research Computing Group, which is supported by the National Science Foundation (awards ACI-1532235 and ACI-1532236), the University of Colorado Boulder, and Colorado State University.



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
