# Peer review of "Behavior and Mechanisms of Doppler Wind Lidar Error in Varying Stability Regimes"

_Atmospheric Measurement Techniques, 2022_

## Referee Comment (RC2)

**Review of the manuscript AMT-2022-73, entitled "Behavior and Mechanisms of Doppler Wind Lidar Error in Varying Stability Regimes", by R. Robley and J. Lundquist**

This study focuses on the errors associated with the measurements performed with a pulsed profiling Lidar when estimating mean wind direction and speed for the atmospheric boundary layer under different atmospheric stability regimes. The problem is tackled from a numerical standpoint through the virtual lidar technique, namely by sampling the wind field simulated with the WRF-LES model through a Doppler Beam Swing (DBS) scan. Two main sources of error are identified, namely the horizontal heterogeneity and range-weighting function (RWF). These errors are investigated through the random-variable theory and convolution integrals, respectively.

The topic is definitely of interest and thoroughly examined in the manuscript. However, a major point that could be improved is the clarity of the language throughout the manuscript. Some statements may result a bit cumbersome and need to be read several times for a thorough understanding (more details in the following). Furthermore, motivations and discussions are qualitatively reported without including details and references, particularly in Sections 3 and 4.

The quality of the figures (in terms of labels and panel size) could be improved. Finally, from a technical standpoint, I have noticed some confusion in defining the temporal full-width half-maximum (FWHM) of the lidar pulse, range gate, and accumulation time, which are independent parameters (more details reported in the following). Given their importance for this work, I recommend defining them more clearly.

**Specific comments**

1.      **L164**: Earlier you referred to $\tau_m$ as "temporal range gate"; please avoid the introduction of new terminology unless strictly necessary. Also, the accumulation time is the time interval when the back-scattered signals are collected from a certain distance and ensemble-averaged in the Fourier space to single out the Doppler shift. Thus, it is independent of what here has been called "temporal range gate" (which is the spatial range gate divided by the speed of light); also, referring to Table 2, $\tau_m = 265$ns is too small to be an accumulation time (typically ranging from 0.5 s to few seconds). Please address this point.

2.      **L166**: To my understanding, Fig. 2b, c and d refer to time-averaged velocity profiles, which might be marginally affected by the shape of the RWF. The profiles of the Reynolds stresses may be more relevant for this discussion.

3.      **LL 292-294**: Using temporal FWHM of 165 ns (as reported in table 2), I obtain a range gate of (Frehlich & Cornman, 2002):

$$\sqrt{\ln 2}\, c \cdot \text{FWHM} = \sqrt{\ln 2} \times 0.29979 \frac{\text{m}}{\text{ns}} \times 165\text{ns} \approx 41\text{m}$$

If you used this relationship, please state it in the text with reference. Also, the effective range gate of the pulsed LiDAR system is defined as the temporal FWHM (multiplied by the speed of light) plus the range gate (Frehlich & Cornman, 2002) (265ns from table 2, corresponding to 79.5m). Please clarify this aspect.

4.      **L310**: In my opinion, an important point missing in this analysis, with respect to a real-scale experiment, is the decreasing of the backscatter coefficient moving away from the lidar, and, thus, the carrier-to-noise ratio (CNR). In other words, the presence of noise in the backscattered signal can severely impair the outcome from the lidar, and, thus, the velocity statistics. I understand that this effect is out of the scope of this analysis, but it could be mentioned when the error analysis is carried out.

5.      **LL323-331**: This part would be clearer with a figure reporting vertical profiles of statistical estimators of the error. Please remove Appendix A and put the relative figure here.

6.      **LL336-340**: From figure 5 (leftmost panel), I expected the mean error for the volume-averaged case to be lower than that for the 'tower' case considering the lower standard deviation. Can you please add some comments on this feature?

7.      **LL427-428**: Is this velocity still a function of time? If so, please add this detail.

8.      **LL444-445**: In general, the homogeneity is not violated due to the presence of turbulence. A flow can be homogeneous in a certain direction (i.e. having the same p.d.f. for all the sampled points) even in presence of turbulence. The lack of homogeneity is given by local spatio-temporal variations of the mean flow and turbulence statistics. Please rephrase this sentence accordingly, if you agree.

9.      **L506-507**: I am not sure why the RWF should impact the time-, streamwise- and spanwise-averaged velocity components for a homogeneous flow since the time and length scale of the flow are much greater than the probe volume. Please clarify.

10.      **Figure 12**: If the aim is to highlight the flow heterogeneity, then instantaneous flow visualizations cannot provide this information. Flow visualizations would work better. On the other hand, if you just want to show the positioning of the Windcube V2 within the domain, this figure works fine.

11.      **L653**: Please provide more details about the trend line.

12.      **LL 668-671**: This improvement is not so evident in Figure 15. If you want to highlight it, for instance, you can plot the median Normalized Wind Speed Error (plus-minus 25th percentile) as a function of the misalignment.

13.      **L681**: The term "noise" here is misleading. It can be replaced, for example, with "scattering" or "variability".

14.      **LL 702-704**: This sentence is unclear. It might not be correct to mention "mean error" referred to individual measurements. Further, it is unclear what the authors mean by saying "the time average approaches the potentially less biased 'ensemble' error from the selective sub-sample". Please clarify or rephrase it.

15.      **L705**: To my understanding, here you refer to the variance (i.e. the uncertainty) associated with the mean of the random process, which is a function both of the variability of the process and the number of samples. Please clarify this aspect.

16.      **L708-709**: This observation is qualitative and potentially not true. Please either remove this sentence or provide references.

17.      **L717**: This assumption is correct, but it relies entirely upon the estimate of the decorrelation time $\tau_c$. Please provide details about its calculation. Or, as an alternative, if you are interested in a more precise estimate of the standard deviation error, a good reference is Benedict & Gould (1996).

18.      **L739**: What do you mean with "background signal"? Do you refer to the time or space average?

19.      **L900**: This equality is unclear. Please clarify and/or rephrase it.

**Technical comments:**

20. **L15**: Which velocity components are named $u, v$?
21. **L21**: Please add the meaning of the acronym lidar.
22. **L31**: Please state clearly that the measured velocity is the along-beam, i.e. radial or line-of-sight, component.
23. **L105**: Also mention the conclusion reported in Sec. 5.
24. **L114**: For the sake of clarity, please put table 2 at this point of the discussion.
25. **L150**: Please mention that $\Delta p$ is the lidar range gate.
26. **LL194-195**: This sentence is unclear, please rephrase it.
27. **Table 2**: Please add one row reporting the azimuth angles used by the DBS scan.
28. **L297**: Please revise this sentence as it sounds unclear.
29. **Figure 5**: In the caption, instead of "dotted", you should refer to solid lines to indicate the volume-averaged reference.
30. **L358**: Please add a reference to figure 6 here.
31. **LL391-392**: To my understanding, the error metric is the same both for the pointwise and the volume-averaged reference. It would be more correct to say "[...] merging of the two error distribution profiles".
32. **L451**: Please provide a reference.
33. **L559**: I guess here you refer to figure 12.
34. **L644**: Please revise this sentence as it sounds unclear.
35. **LL659-661**: This sentence is unclear, please rephrase it.
36. **L706**: Please replace "variables" with "samples".
37. **L716**: Please provide any reference or derivation for this equation. Also, provide a brief description of the terms reported here, which have not been introduced before.
38. **L821**: This study offers a relevant comparison for the present work, so it should be discussed at the beginning of the previous Section.
39. **L884**: You can replace this Appendix with simple literature references you deem pertinent.
40. **L897**: Do you refer to Equation 22 here? If so, please state it clearly.

**References:**

Benedict, L. H., and R. D. Gould. (1996). Towards better uncertainty estimates for turbulence statistics. *Experiments in Fluids,* 22(2), 129-136.

Frehlich, R., & Cornman, L. (2002). Estimating spatial velocity statistics with coherent Doppler lidar. *Journal of Atmospheric and Oceanic Technology*, *19*(3), 355–366.

---

## Editor Comment (EC1)

Additional comments on *Behavior and Mechanisms of Doppler Wind Lidar Error in Varying Stability Regimes*: Clarification of WindCube v2 wind speed computation.

Andrew Black
April 28, 2022

*Section 3.2 Ten-minute time averaged velocities*

The WindCube v2 10-minute average wind speed reported by the device in the STA data is **a scalar average** of the 1 Hz horizontal wind speeds:

$$\overline{V_{scalar}} = \frac{1}{600}\sum_{i=1}^{600} V_i = \frac{1}{600}\sum_{i=1}^{600}\sqrt{u_i^2 + v_i^2}$$

The WindCube v2.1 uses a weighted linear combination of scalar and vector averaging:

$$\overline{V_{hybrid}} = \frac{1}{3}\overline{V_{vector}} + \frac{2}{3}\overline{V_{scalar}}$$

This is to point out that there **are not any WindCubes (except a few early v2.1's) reporting pure 10-minute vector averages in the STA data, as in your paper.** The vector-averaged data is available today in VSTA files on board the device, and can be computed from the 1 Hz LOS RTD data, but typical uses use the average wind speeds in the STA files.

Vector lidar and pointwise measurements (implicitly mimicking a sonic anemometer) or full vector wind field averages are shown (dashed lines, Figure 8, et al). Are the *pointwise* 10-minute averages computed using vector-averaged u, v, and w components? I assume yes, but this should be clarified.

It would be very interesting to compare the 10-minute scalar- and vector-averaged pointwise measurements to 10-minute scalar- and vector-averaged lidar measurements. Here's why:

In the Rosenbusch et al (2021) article, the differences between scalar and vector wind field reconstruction for pulsed DBS lidars (with WindCube scan geometry and timing) when compared to scalar-averaged cup anemometry were shown, theoretically, to depend on the correlation between *u, v,* and *w* turbulent components, thus likely strongly influenced by stability, just as you've divided your data. The data used in Rosenbusch was restricted to comparisons between scalar *cup* averages and scalar and vector lidar averages. The theory developed in Rosenbusch et al implies that 10-minute vector averages of pulsed DBS lidars with WindCube scan geometry should not exhibit systematic, WFR-caused biases when compared to vector-averaged pointwise measurements (though RWF biases may exist, as you observe).

On the other hand, it shows that the 10-minute scalar-averaged lidar measurements should exhibit systematic high biases when compared to scalar-averaged pointwise measurements, and that this bias

should vary in different stability regimes. As in your paper, this is shown in propagation of the turbulent decomposition through the WFR algorithm developed by Jennifer Newman. **This contradicts (or at least restricts to vector WFR) an observation in your paper:**

> Line 545-546 : *"For the most part, the error mean biases can be attributed to RWF effects and the velocity perturbation terms do have close to zero mean, but important deviations from that assumption do arise"*

The scalar WFR case should show systematic biases due to the velocity perturbation term, especially in the convective cases.

Your dataset is ready to make these scalar-to-vector, and scalar-to-scalar comparisons between the lidar and pointwise measurement. I think it would be a valuable addition to the paper. Adding the scalar-averaged pointwise ("cup-like") measurements would expand the scope of your results to another sensor type and constitute a more comprehensive first result using this simulation data. I believe it would increase the impact, as well, due to the ubiquity of cup anemometry in wind energy. Treatment of uncertainties for cup anemometry (and for lidar) is covered in multiple IEC standards (61400-12-1, -15-1, -15-2, 50-3, 50-4, et al). This topic of sensor uncertainty and error is of great importance for the wind energy industry, and I think your simulation framework is a breakthrough.

One last thought is that adding the 10-minute scalar averages would also allow for direct propagation of the 1 Hz errors to 10-minutes through the scalar averaging equation, a way to connect those two sections of the paper more strongly. This would require an interesting treatment of the covariance between the neighboring 1 Hz measurements, which share 1, 2, or 3 LOS measurements, essentially the covariance of a moving average (and not only the wind itself).

Best regards,
Andrew

---

## Author Comment (AC1)

**Response to CC #1 (Andrew Black)**

**Comment on amt-2022-73**

Andrew Black

Community comment on "Behavior and Mechanisms of Doppler Wind Lidar Error in Varying Stability Regimes" by Rachel Robey and Julie K. Lundquist, Atmos. Meas. Tech. Discuss., https://doi.org/10.5194/amt-2022-73-CC1, 2022

What interesting research, and seemingly powerful tools to use for years to come!

Figures 12 and 14 really bring the simulation data and lidar error to life. Could these be presented earlier in the paper? Perhaps swapping sections 2.1 and 2.2, and showing the 2D cross section of LES data with lidar beam overaly in the beginning to allow the reader to visualize the dataset.

On input from another reviewer, we have reordered some of the material to develop the random variable error model in Section 2. We've brought Figure 12 up along with it (now Fig. 4), so while not quite as early as you suggest, it hopefully allows for the reader to build intuition about the dynamics at play.

The ridgeline plots are very compelling. Could you add one that includes the all of the 10-minute averaged data? This would illustrate the reduction in errors through time averaging you describe.

While we decided not to take more space for a ridgeline plot of the 10-minute average, we have updated our discussion and figures of the effects of time-averaging to better show the trends at work. We have incorporated violin plots of the error distributions at different averaging intervals ranging up to 10 minutes, which performs a similar visualization. (Fig. 15)

In figures showing statistical moments by height, the x-axes are scaled to the data. In some cases, this highlights interesting trends, and they should remain, but I wonder if auto-scalaing obscures a conclusion: the errors are small. Deciding when errors are significant, and when they are insignificant is a key part of this paper.

Good point. When regenerating the figures, we've tried to strike a balance between making use of the panel space to clearly distinguish the trends and picking an appropriate scale that doesn't over-magnify small values (especially in the near-zero biases).

p15, L351 "inflection point" p176, L374 "will converge" Done, thanks!

In Section 4, did you consider illustrating some of the error trends as functions of atmospheric parameters like instantaneous (or 10-minute) wind shear, turbulence, etc? Perhaps you could pick only one measurement height to do this instead of the full profile, and then illustrate various errors as functions of upstream errors.

This is a lovely idea and we wish we had the data to try to visualize those dependencies. Unfortunately, the limitation in the number of LES test cases and their uniformity means the simulated flow fields don't span much of the atmospheric parameter space. We hope that by diagnosing the mechanisms behind the error, the reader can extrapolate likely behaviors in other conditions not included in our simulations.

In Fig 14, add color-coded trendlines for each stability class (instead of only the dashed white line). The positive bias in the low speed Strong CBL seems to be a key finding. Highlight it as best you can. Could you show the same graph but perhaps for the most and least biased heights?

Thank you for encouraging us to revisit this figure with more scrutiny. All else being equal, the error expression for the wind speed suggests that a higher wind speed should repress the positive bias (according to $1/|U_h|$). Having separated the trend by stability (Fig. 12) (and height, not shown), we do not feel we

can discern the expected trend from the other factors changing implicitly with wind speed within the LES test cases. We have adjusted our statement of this behavior accordingly (L685-692).

Section 4.5 illustrate the rapid decay of error with time averaging, it's very steep and interesting for wind energy folks who only ever think of 10-minute averages.

How can your illustrations complement your conclusions most strongly? In some cases you might want to clearly focus on one height instead of showing the whole profile.

Forgive me if any of my comments are addressed elsewhere in the paper. There is a lot to digest. This paper is so thorough and really excellent.

Thank you for the positive feedback. We have found the thoughts on how to better visually highlight the dynamics at play useful in revising the manuscript to best communicate the results from the virtual model.

---

## Author Comment (AC2)

**Response to Referee #1**

**Comment on amt-2022-73**

General comments:

This is an interesting study that tackles a fundamental problem for the Doppler wind lidar observations which is the better understanding of the measurements' error. The authors used large-eddy simulations for different atmospheric stability conditions and analyzed which mechanisms drive the errors under each condition. This study can be very useful for researchers studying turbulence as observed by a Doppler wind lidar. It could help them understand possible error biases or the cause of large errors. It also shows the importance of multiple scans so that the observations can be averaged over time to decrease the error. I believe this study is worth publishing. Nevertheless, the authors should make several changes in the manuscript in order to improve it's structure. In the current state, it is difficult to read through. Introduction, methodology, results and conclusions are mixed, the authors should set apart these chapters to improve the flow of the manuscript. In the current version, parts of the introduction are presented in the methodology, whereas parts of the methodology are introduced in the results and some results are presented but discussed in a later section. In the conclusions, a summary of the work with the main points of the results should be presented, however the authors make a comparison of their results with previous studies and furthermore they introduce new references. Regarding the abstract, the results should be supported by numbers so it will be evident that these are the results of the current study. Some terms should be explained better throughout the study, so it is clear to the reader what is the magnitude of terms such as the "strong winds" or the "large structures". The figures and their captions need corrections as well. All the important information should be included in the plot with the description written in the caption, rather than important information, such as the colour of the lines, only explained in the caption. The language in the manuscript is fluent. Please find my comments below.

We would like to thank the reviewer for their time reviewing the manuscript and thoughtful comments and appreciate the acknowledgement of the applicability of our findings for researchers working with Doppler wind lidar. The insights into how the material can be better presented were particularly useful.

**Scientific comments:**

Page 2 Line 27-29: The authors mention that lidar data offer an indirect representation of the flow field. Although it is true that the wind lidar observations most likely need extra steps to extract the wind components compared to sonic anemometers, it is possible to directly observe the wind components if the beams are aligned with the wind direction e.g. horizontal beams with no elevation angle alongside the wind direction or vertical beams for the vertical wind.

Measuring the along-beam component is more direct than the reconstruction required by a profiling lidar, but we would still view it as less direct than e.g. a 'point' measurement made by sonic anemometers. As modeled by the RWF the probe length of the lidar sampling process is 10s of meters long vs decimeters and can interact with shear in the flow.

P2 L45-47: Similarly to my previous comment. In the phrase "Questions about measurements of vertical velocities"; do the authors mean vertical velocities variances? Because vertical velocity can be directly measured by the Doppler wind lidar and in fact even the vertical velocity variance is the "easiest" turbulence parameter that can be estimated using wind lidar observations, see Bonin et al. 2016: "Improvement of vertical velocity statistics measured by a Doppler lidar through comparison with sonic anemometer observations".

"Questions about measurements of vertical velocities and the ability of wind profiling lidar to measure turbulence remain areas of active research (Sathe et al., 2011; Sathe and Mann, 2012; Sathe et al., 2015; Newman et al., 2016)."

We referred to the way the volume-averaging can complicate even the 'direct' measurement of the vertical velocity and the variance, which is addressed in the Bonin et al. 2016 paper. We have amended the statement to reflect the particular challenges in the turbulence measurements and included the paper as an additional reference.

(LL49-50) "Questions about the ability of wind profiling lidar to measure turbulence remain an active area of research (Sathe et al., 2011; Sathe and Mann, 2012; Sathe et al., 2015; Newman et al., 2016; Bonin et al., 2016)."

P2-3 L59-62: The sentence "Compared to field studies of instrument accuracy, studies with virtual instruments in LES ...." should be supported by some relevant references for such studies.

We have amended this sentence to acknowledge studies that have leveraged the unique opportunities presented by a virtual instrument.

(LL62-66) "Compared to field studies of instrument accuracy, studies with virtual instruments in LES have unencumbered access to full knowledge of the flow field. This knowledge enables control over the case parameters (terrain, forcing, boundaries), and so users can "deploy" instruments in ways that may not be physically or financially possible in reality (e.g. re-sampling the same flow field or testing many locations in a domain) (Muschinski et al.,1999; Stawiarski et al., 2013; Wainwright et al., 2014; Gasch et al., 2019).

P3 L67: The PALM model was developed by Raasch and Schroter, 2001: "A large-eddy simulation model performing on massively parallel computers". The reference should be added in this sentence. Added here, thanks!

P4 L78-79: The Skamarock, 2008: "A description of the Advanced Research WRF version" reference should be included in this sentence. Added.

P4 L82: The Chow et al., 2005 is not relevant here as they do not mention the WRF model in their study.

We have corrected this citation to Kirkil (2012).

P10-11 L248-270: The authors state that they selected cases with "strong" and "weak" convective boundary layers. In line 262, these cases are characterized well-mixed layers. However, in Figure Table 1 we can see that the for the weak cbl the abl height is 525 m. This value seems to correspond to a developing boundary layer and not a well mixed. The authors should comment on that and whether these values can occur only in an ideal case of the simulation with a flat, homogeneous terrain etc. Moreover, the vertical range of the Windcube is portrayed in Figure 3 but it is not mentioned in the text and even in Table 2 it is not directly shown. This information should be included along with an explanation of this selection. The value is higher than the ABL height under stable conditions (170 m). Wouldn't this affect the comparison? See also my comment regarding Figure 4.

Regarding the "weak" convective boundary layer: The convective cases drawn from Rybchuk (2021) are emulate conditions of the Project Prairie Grass campaign (Barad, 1958) which looked at diffusion of tracer gasses. The ABL height in the weak CBL case can and did occur in real conditions. The depth is a function not just of the surface forcing but also of the strength of the inversion aloft which allows for the 525m ABL to occur in a well-developed boundary layer. We have noted in the text that these cases reflect real, observed conditions and added a reference to the campaign along with the existing reference to Rybchuk (2021).

Regarding the vertical range of the Windcube: We have added a line to Table 2 (now Table 1) to make the range explicit (40-240m AGL). The range arises from values fixed by the instrument and typical operating parameters, i.e. the number and vertical spacing of the range gates. We have added this explanation to the text (LL201-203). The fact that the range exceeds the ABL height under stable conditions does affect the comparison. We refer to the boundary layer structure as part of the discussion of the error behavior (e.g. the peaking in the perturbation variances mid-boundary layer and tapering aloft in the stable BL while the

growing variances in the convective BLs only reach mid-boundary layer at the top of the range) but did not see fit to normalize by the boundary layer height since the geometry of the instrument is fixed.

P11 L266-267: The authors mention the limitation of the lidar range to include the entrainment zone. For the instrument, it is practically difficult to measure at this height due to the scarcity of aerosol in the zone. Do the authors refer to the instrument or the simulations? What are the limitations for a simulation?

Practical considerations like sufficient aerosols were outside the scope of our analysis, though important to actual operations. There is no reason the simulation / virtual lidar model couldn't be extended to higher ranges and the range limitation is introduced based on general Windcube instrument parameters (which do depend on considerations of aerosols and obtaining a usable signal) of the number and spacing of the range gates. Since the instrument geometry is more or less fixed for deployments across conditions, we have mimicked that geometry and range throughout the test cases (while ignoring, e.g. the possibility that a real instrument would have insufficient aerosol scattering to return a usable signal within that range).

P14: In Figure 4 panel (c), for the lower altitudes the median of the curve is not at zero as it also stated by the authors in P15 L356. The authors should explain at this point of the manuscript why this occur only at these particular levels of the SBL. It is also apparent from panel (c) that for the levels above the ABL height (170 m) the distribution becomes similar to the one near the surface. Any comments regarding this? Was this something the authors expected?

P15 L355: What could be the cause for overestimation during convective conditions?

For both these points, using the suggested re-ordering of the material to introduce the analytic error model in Section 2 allows the mechanisms behind these behaviors (RWF acting on the shear / vertical wind profile and the systematic positive bias term in the wind speed error) to be addressed as soon as they are shown.

The (moved-up) diagnosis of the causes of bias and variance in the error explains why the error above the stable ABL resembles that at the surface. The filtered turbulence under the scan-volume scale is similarly smaller at the surface and above the ABL (particularly the vertical velocity variances) which reduces the width of the distribution. Similar curvature in the background u velocity profile induces the negative biases in the shear layer in the surface but also aloft through the RWF.

A positive bias term (proportional the u,v variances / inversely proportional to the mean wind speed) gives rise to the overestimation in convective conditions when vertical velocity variances drive larger errors.

In the text, the discussion of these points and the connection of the error behavior to the vertical structure of the boundary layer turbulence (which were split between Sections 2 and 3 in the original manuscript) have been merged into a single, streamlined results section (Section 3).

P24 L506-507: The terms strong winds and strong shear are vague. Can you quantify these parameters? Is the underestimation expected above a specific threshold? Additionally, the argument that surface shear is one of the cause of the underestimation should be supported by some results in the form of Figures. Maybe the authors could add a secondary y axis with the wind speed and wind shear values at the different given heights in Figures 9, 10, 11. The estimation of the wind shear can be tricky but this claim should be supported by results.

"The most prominent influence of the RWF is near the surface layer (due to strong shear) and manifests as a negative shift in the mean error in the horizontal velocities. The weak CBL and stable BL, with strong winds and surface shear, exhibit the effect most strongly. "

(This material has been moved to Section 3.1 discussing the source of biases in the results. LL597-605)

The connection to the shear is mechanistically supported by our derivation of the bound (Eq. 16 in the edited manuscript) as well as previous studies (Clive, 2008; Courtney et al., 2014) that have derived the action of an RWF on idealized shear profiles. (These have been included in the discussion.)

The underestimation will occur for any concave function (like the semi-empirical log shear profile for U(z)); there is no threshold except what degree of bias is considered tolerable. The strong CBL, for example, does still have some shear near the surface but the resulting negative bias is <0.05 m/s, so would likely be considered negligible.

We have added Fig. 9 to help support the conclusion in the case-study results. Fig. 9 demonstrates that the bias due to the RWF in the full model and the bias resulting from applying the RWF to just the mean LES profile and shows the largest magnitudes in the bias in u (b) and v (d) occur in regions with the largest (nonlinear) shear.

P26 L540 & 548-550: Similarly to my previous comment, the term "large coherent structures" is vague. It is evident from Figure 12a that there are structures, upward motions followed by downward motions, of approximately 1 km size. On the contrary, for the stable boundary layer (Figure 12b) the size of the structures seems to be equal to few hundred meters. The lifecycle of such structures should be different. Do the authors categorize both structures' sizes as large? The authors also mention larger scale structures above the boundary layer in the SBL. Do they mean from 170 m up to 350 m that is shown in Figure 12b or in higher altitudes? Either way it should be clear to the reader what are the sizes of the structures and these should correspond to the figures presented in the manuscript.

"Without dissecting the mechanics more closely, we simply note that large coherent structure like turbulent plumes characteristic of the convective boundary layer can induce repeated, non-symmetric patterns in the relative perturbations of the beams, leading to non-zero means."

(This material has been moved to Section 3.1, LL590-596)

This is an important point about the size of the structures, which Fig 12 (now Fig. 4) was meant to help convey. The critical scale here is the size of the scan volume (which for the WindcubeV2 is ~42m across at the bottom range gate and ~255m across at the top, shown by the black lines in Fig. 4). "Large" coherent structures are meant to refer to features on the order of the scan volume or bigger so that their internal structure influences the sampling by opposing beams, which would include the 1km structures but potentially exclude some of the features in the lower stable BL.

We have explicitly called out "coherent structures large enough to span the scan volume" and denoted the scan radius in the text to help clarify this point.

**Technical comments:**

P1 L1: Lidars instead of lidar. Fixed.

P3: The full name for the abbreviation DBS should not be included in the caption of Figure 1, but rather in P4 L76.

The definition of the full name for the DBS abbreviation has been moved, along with checking the order and use of other abbreviations in the text.

In the first three instances Figure 1 is written with a capital F in the text, whereas in all the other instances figures in the text are written in parenthesis with a minor f. The citation of the figures should be consistent throughout the text.

Thanks for pointing this out. These instances along with all further references to figures and equations were changed to "Fig.", "Figures", and "Eq." to bring the text in line with journal style guidelines.

P3 L72: The full name for the abbreviation SOWFA should be given here. Fixed.

P3 L73: The full name for the abbreviation WRF-LES should be given here. Fixed.

P7 L156: The parameter "c" is defined after the equation 7, although it is also part of the equations 5 and 6. In order to avoid confusion, I suggest to move the definition before or after the equation 5.

Good catch. We have moved the definition in the text up to the first reference to the use of the speed of light before Eq. 4

P7 L161: The sentence "Using parameters .... can be made concrete" needs rephrasing. It is confusing in its' current state.

"Using parameters from the WindcubeV2 (Table 2), the shape and extent of the RWF (Eq. 6) can be made concrete (Fig. 2a)."

reworded to

(L170) "The RWF for the modeled WindcubeV2 (Fig. 2(a)) results from substituting the range gate and pulse parameters (Table 1) into the general, pulsed lidar equation (Eq. 6). The shape of the RWF peaks at the center and tapers symmetrically toward zero to either side."

P7 L167-170: The paragraph "The form of the pulsed lidar .... further distances by a pulsed lidar" is more suitable for the introductory section. The comparison of the RWF between pulsed and continuous lidar seems out of place in the methodology as only the pulsed lidar was used for this study.

We agree that the discussion here became more general than needed to serve the primary focus of the study. We've dropped the paragraph from the text.

P7 L172: Remove the word "found".

We have reworded the end of this sentence from

"…and the projection onto the beam direction found"

to

(L177) "…and then projected onto the beam direction".

Hopefully this makes it scan more cleanly for the reader.

P7 L173: The "Spe" in the parenthesis is a missing reference?

Yes; a reference used to split the scipy function (into triangulation and application of the interpolation weights to optimize repeated calls on the same grid) does not have a complete reference and therefore was mangled. We have moved that reference to the code and retained the complete technical scipy citation here.

P8 L185-188: The paragraph "Interpolation dominates ... subsequent developments" comments on the interpolation method and possible improvements in the data and methodology section. This paragraph is more suitable for a section like Conclusions/Discussion.

We have removed this paragraph and made a brief reference to the optimization of the interpolation as a target for future improvements in the conclusion (L948)

P8 L190-199: In the first paragraph of Section 2.1.2 the authors provide some general information regarding the lidar and the different scanning methods such as RHI and PPI. As these methods are not used in the particular study, they should not be mentioned in the data and method section. In my opinion, this paragraph should be removed entirely from the manuscript as it does not provide any valuable information for the study.

We had included this in a general approach to the virtual lidar model, but agree that the additional information in this paragraph is a distraction that doesn't serve the current study. We've cut out this portion of the paragraph and condensed it to simply refer the reader to Clifton (2015).

P10: In Figure 3 the explanation for the different lines (solid, dashed etc) representing the parameters is only included in the captions and not in the figures, hence it is not practical for the reader to study these figures, similarly for Figures 5, 6 etc.

We have reviewed each of the figures in the manuscript with this in mind and have included full legends reflecting each of the parameters used.

P13 L303: The claim that the components of equation 10 are commonly used should be supported by some examples-references.

We have included a reference to Cariou and Boquet (2010); the instrument itself returns the horizontal wind speed and direction and we believe they are fairly ubiquitous. (L414)

P13 L305-308: A figure showcasing the sign convention could be useful for the reader.

We've created a diagram (Fig. 5) to show the conventions used for the wind direction and sign of the direction error.

P14: The authors have introduced several parameters for the wind speed such as ulidar or horizontal wind speed, therefore it should be clear in Figure 4 which one is shown including its' name as defined by the authors.

We have re-labeled the figure axes with both the name and symbolic reference (|U_h|_err) which refers to the error in the magnitude of the horizontal winds (i.e. 'wind speed error'). Hopefully this addresses any possible confusion about the plotted quantity.

P14 L324: Remove "the". Fixed.

P15-P16: The caption of the Figures 5 and 6 mention dashed and dotted lines but only dashed and solid lines are depicted. The figures should be corrected and additionally there should be a legend in the figure with this information.

Thanks for catching this and we apologize for any confusion it may have caused in reviewing the article. The caption has been amended and the figure revised to include the line style distinction in the legend as well.

P15: Figures should be easily readable even when separated from the rest of the text. In Figure 8, the caption is linked to the parameters of Figure 5 which should be corrected. The parameters should be included in the caption and the legend of Figure 8 independently from Figure 5.

We have regenerated the figures to include comprehensive and consistent legends across these figures.

P17 L374: Converge instead of converges. Fixed.

P18 L401: The phrase "their respective height trends are similar to the previous section" should be accompanied by the respective values as a reminder for the reader. Not in the rearranged manuscript. We have tried to make sure to call out values and reiterate the recurring trends when we refer to them in the updated text.

P19-P33: In Chapter 4 the authors introduce several new equations. In my opinion, a manuscript flows better when all the equations and tools are presented in the data and methods chapter and subsequently the results are presented and discussed. So instead of presenting the figures in Chapter 3 and then using the equations to describe the results in Chapter 4, I believe it would be better if all the equations are already presented in Chapter 2 and the discussion of the results is moved to the corresponding figures. For example the explanation of the underestimation of the wind error although mentioned in Chapter 3 is explained much later in Chapter 4. By moving the equations to Chapter 2, the authors will also avoid repeating themselves.

Thank you for your thoughts on how best to organize the material. We had developed the analytic model for the error as a tool for understanding the output of the virtual tool and had thought it could be treated

concurrently with the results in the paper. Seeing the degree to which the analytic development grew and how much we rely on it in explaining the results, we agree that the flow of the article is improved by re-ordering the presentation.

We have moved the development of the analytic model for the lidar error into a new subsection of the Data and Methods (Section 2). The results using the virtual instrument have been condensed into a single section (Section 3) and streamlined, referencing back to the analytic model in Section 2 when needed for the discussion. We hope this removes ambiguity and repetition and makes the paper stronger and more clear to the reader.

P20: The caption of Figure 9 seems more like a part of the manuscript than a caption. It should be rephrased in a way to resemble a caption.

P21-22: It would be easier to interpret the results from Figures 9, 10, 11 if these figure were merged in multiple panels and thus it would be possible to use the same caption for all instead of linking to the caption of Figure 9 which makes the Figures unreadable independently.

To both these points:

We have revised this set of figures into a single, combined decomposition figure (Fig. 8) which uses the same caption. The revised figure shows the covariances and obviates the need for some of the explanatory text in the caption. The specification that only the non-offset lidar are used was moved to the main text.

P29 L616: The word "term" is used two times. Fixed.

P30: The height of the wind speed is not mentioned in Figure 14. It should be included as part of the xaxis title. Figure 14 (now Fig. 12) aggregates data from all heights. Since the original manuscript left room for uncertainty, we have added an explicit statement to the caption to clarify the point.

P32: The legend showcasing the parameters that correspond to the colour lines is missing in Figure 16.

We have replaced this figure to better illustrate the trends over the time-averaging window. The new version (Fig. 15) does include the color-coded legend.

P32: The panels in Figure 16 are not numbered with (a), (b), (c) etc and the authors refer to the different plots as left and right panels. I believe it will be easier to use the numbering. Similarly the Figures 5, 6, 8, 9, 10, 11, 13, 17, 18 and the ones presented in Appendices A, D, E also miss the numbering.

Thank you for pointing out the inconsistency and confusion in (not) labeling the panels in these figures. We've worked through each of the figures to clean them up and add labels in a consistent style where they were missing and have updated the captions to reflect the changes.

P36-39: The comparison of the authors' results with previous studies should be included in the sections of the results and not in the conclusions. The authors should summarize the key points of their study and not include new information in this chapter. Although it is possible to include some previously mentioned references in the conclusions, it is not recommended to introduce new references such as Klaas and Emeis, 2021 and Teschke and Lehmann 2017. These references should be introduced earlier in the manuscript.

The comparisons of how our findings relate to previous studies have been mentioned as part of the results where appropriate (now Section 3) and moved into a discussion section (Section 4). We have correspondingly shortened the conclusion to summarize our key points.

P38 L840: Add the word of - "The form of our error...." Fixed.

P52 L1117: Remove typo " ". Fixed.

P52 L1119: Remove typo "&ndash". Fixed.

---

## Author Comment (AC3)

**Response to Anonymous Referee #2**

**Review of the manuscript AMT-2022-73, entitled "Behavior and Mechanisms of Doppler Wind Lidar Error in Varying Stability Regimes", by R. Robey and J. Lundquist**

This study focuses on the errors associated with the measurements performed with a pulsed profiling Lidar when estimating mean wind direction and speed for the atmospheric boundary layer under different atmospheric stability regimes. The problem is tackled from a numerical standpoint through the virtual lidar technique, namely by sampling the wind field simulated with the WRF- LES model through a Doppler Beam Swing (DBS) scan. Two main sources of error are identified, namely the horizontal heterogeneity and range-weighting function (RWF). These errors are investigated through the random-variable theory and convolution integrals, respectively.

The topic is definitely of interest and thoroughly examined in the manuscript. However, a major point that could be improved is the clarity of the language throughout the manuscript. Some statements may result a bit cumbersome and need to be read several times for a thorough understanding (more details in the following). Furthermore, motivations and discussions are qualitatively reported without including details and references, particularly in Sections 3 and 4. The quality of the figures (in terms of labels and panel size) could be improved. Finally, from a technical standpoint, I have noticed some confusion in defining the temporal full-width half- maximum (FWHM) of the lidar pulse, range gate, and accumulation time, which are independent parameters (more details reported in the following). Given their importance for this work, I recommend defining them more clearly.

We would like to thank the reviewer for their thorough reading and constructive comments. The feedback has been helpful in ensuring the technical details are correct and accurately conveyed.

**Specific comments**

**L164**: Earlier you referred to $\tau m$ as "temporal range gate"; please avoid the introduction of new terminology unless strictly necessary. Also, the accumulation time is the time interval when the back-scattered signals are collected from a certain distance and ensemble-averaged in the Fourier space to single out the Doppler shift. Thus, it is independent of what here has been called "temporal range gate" (which is the spatial range gate divided by the speed of light); also, referring to Table 2, $\tau m$ = 265ns is too small to be an accumulation time (typically ranging from 0.5 s to few seconds). Please address this point.

Thank you for catching this. We made an unfortunate overload of the term 'accumulation time', which should, as you say, refer to time used to collect an ensemble of samples the average of which is used to diagnose the Doppler shift. The discussion of the "accumulation time" was mis-used here to refer to the temporal range gate. We apologize for the confusion the oversight caused and have fixed the wording to remove the mis-used language and clarify our meaning.

> "The choices of pulse and range-gate parameters in a coherent lidar system must balance the desire for spatial locality (reducing the width of the RWF) and the need for adequate accumulation time ($\tau_m$) to accurately resolve frequencies used for the radial velocity measurement."
>
> was changed to
>
> "The range-gate parameter in a coherent lidar system must balance the desire for spatial locality and the need for accurate frequencies used in measuring the radial velocity. The more signal points from the traveling pulse used, i.e. the longer the range gate, the more accurate the diagnosis of the frequencies but the longer the averaging volume along the beam." (LL173-175)

In this section, we wanted to make the distinction between the range gate parameter in time (the 'temporal range gate', $\tau_m$) and the corresponding spatial distance in a range gate (the 'spatial range gate', $\Delta p$). The part mis-using the "accumulation time" was intended to make note of how the choice of $\tau_m$ is subject to a trade-off: more points used in the FFT, i.e. a longer range gate, gives more accurate frequencies but

requires a larger probe volume / averaging over a longer spatial range gate along the beam which loses locality of the measurement.

**L166**: To my understanding, Fig. 2b, c and d refer to time-averaged velocity profiles, which might be marginally affected by the shape of the RWF. The profiles of the Reynolds stresses may be more relevant for this discussion

The panels in Fig. 2(b, c, d) are "instantaneous" line-of-sight velocity profiles (implicitly averaged over the grid/time step of the LES model) to which we have applied the RWF. We have edited the caption to make this explicit.

Because of limitations in the current virtual lidar model, we do not treat the effects of accumulation time / time averaging in the beam sampling; the measured line-of-sight velocity is taken directly from the RWF acting on a snapshot of the LES wind. Future/improved versions of the model would have to treat this point more carefully with the turbulent stresses as represented by the LES, but for this iteration of the model the raw line-of-sight velocity profile is appropriate to the underlying behavior.

**LL 292-294**: Using temporal FWHM of 165 ns (as reported in table 2), I obtain a range gate of (Frehlich & Cornman, 2002):
If you used this relationship, please state it in the text with reference. Also, the effective range gate of the pulsed LiDAR system is defined as the temporal FWHM (multiplied by the speed of light) plus the range gate (Frehlich & Cornman, 2002) (265ns from table 2, corresponding to 79.5m). Please clarify this aspect.

For the effective range gate / range resolution along the beam due to the RWF we directly solved for the FWHM of the RWF presented in the paper (Eq 6. with the parameters you give above, resulting in ~43 m). It is our understanding that, as summarized in Cariou and Boquet (2010, p 20), a later estimate of the effective range resolution by Banakh and Smalikho (1994) is indeed about one and a half times shorter than that found using the sum of the FWHM and range gate as in Frehlich & Cornman (2002). This estimate is in line with the FWHM of the RWF arising from the convolution. Hopefully this accounts for the disparity in our numbers.

We don't expect much sensitivity for range resolution values close to 40m. Comparing to preliminary runs using 40m high cylinders for the volume average truth, using the projected ~35m height didn't have an appreciable effect on the results.

**L310**: In my opinion, an important point missing in this analysis, with respect to a real- scale experiment, is the decreasing of the backscatter coefficient moving away from the lidar, and, thus, the carrier-to-noise ratio (CNR). In other words, the presence of noise in the backscattered signal can severely impair the outcome from the lidar, and, thus, the velocity statistics. I understand that this effect is out of the scope of this analysis, but it could be mentioned when the error analysis is carried out.

The CNR is out of scope, but can be significant to real measurements. We have noted its role and omission along with our disclaimer about the ignored factors in the quality of the radial velocity retrieval..

Added to L 137: "We similarly omit impacts of the carrier-to-noise ratio which can introduce additional uncertainty into the diagnosis of the radial velocity (Cariou and Boquet, 2010; Aitken et al.)."

**L323-331**: This part would be clearer with a figure reporting vertical profiles of statistical estimators of the error. Please remove Appendix A and put the relative figure here.

(Section 3 presenting the results has changed in rearranging the order of the paper. )

After due consideration, we have decided to leave the full set of moments with the orientation disaggregation in Appendix A. The results section is already quite figure heavy and we did not feel that it was the best use of space to show the skewness and kurtosis in the main text when they feature only briefly in the discussion.

**LL336-340**: From figure 5 (leftmost panel), I expected the mean error for the volume- averaged case to be lower than that for the 'tower' case considering the lower standard deviation. Can you please add some comments on this feature?

This behavior is related to the mechanics of the additional analysis we have added comparing the scalar-averaged wind speeds of lidar and pointwise (cup anemometer) measurements.

In expanding the wind speed error from which we obtain that the bias is proportional to the u,v error variance, the lidar-derived winds were expanded about the volume-averaged reference wind. This process doesn't really make sense to do in the same way with a point reference which is why the biases using the two references don't directly track from the error standard deviations in both cases. (now noted in L425)

If we think about expanding the pointwise measurement about the volume-average as well (or a Reynolds average) as explored in the added analysis (Section 2.3.5), then the pointwise wind speed will also have a positive bias / inflation relative to that of the volume-averaged wind speed. The biases are proportional to the variance of the fluctuations in the u and v measurements by the lidar or at the point. So part of the bias between the lidar and volume-average is canceled by the additional variability in the point measurement, rather than it being additive to the bias, giving less total bias compared to the point measurement.

**L427-428**: Is this velocity still a function of time? If so, please add this detail.

Yes; we will clarify here. We are letting the velocity vary in time but assuming the volume-average is about constant over the 5s duration of the scan.

**LL444-445**: In general, the homogeneity is not violated due to the presence of turbulence. A flow can be homogeneous in a certain direction (i.e. having the same p.d.f. for all the sampled points) even in presence of turbulence. The lack of homogeneity is given by local spatio-temporal variations of the mean flow and turbulence statistics. Please rephrase this sentence accordingly, if you agree.

This is an important point that we want to make sure is clear in the text.

We have used 'homogeneity' in the sense of the 'horizontal homogeneity assumption', meaning the assumption that winds are uniform/constant across the scan volume so that the reconstruction is exact. This phrasing is common in the literature when referring to the reconstruction of the 3D winds (see below). We recognize that our usage may cause confusion, especially if the reader thinks that we are talking about homogeneous turbulence.

We have adjusted our wording here (L324-327) to be explicit that we are referring to the horizontal homogeneity assumption and clarified usage of the phrase when it is introduced. The LES test cases were designed to be quasi-stationary and quasi-homogeneous (in the turbulence sense) so that the assumption of a *constant/uniform* wind across the horizontal scan volume is violated even in the homogeneous flow due to the momentary realization of the turbulence.

Quantifying error of lidar and sodar Doppler beam swinging measurements of wind turbine wakes using computational fluid dynamics

Remote Sensing of Complex Flows by Doppler Wind Lidar: Issues and Preliminary Recommendations

Turbulent kinetic energy estimates from profiling wind LiDAR measurements and their potential for wind energy applications - ScienceDirect

**L506-507**: I am not sure why the RWF should impact the time-, streamwise- and spanwise- averaged velocity components for a homogeneous flow since the time and length scale of the flow are much greater than the probe volume. Please clarify.

In our model, the sampling of the line-of-sight velocity using the RWF acts on a snapshot of LES flow field (which contains some implicit spatial and temporal averaging in the LES-discretized representation of the flow). But the key point with the RWF bias is not the variability it introduces (it's small, Fig. 8) but the

repeated action on the underlying, background wind profile. The volume average on a radial velocity profile with curvature is inducing a bias.

The mean bias effect persists even when acting on the time-, streamwise-, and span wise-averaged velocity because of the action of the RWF on curvature in the vertical profile of the velocity. In *Shear and  Turbulence effects on Lidar Measurements* (Courtney, 2014) an analysis of this behavior on analytic representation of shear profiles is done, showing the induction of the bias and which our findings support. Some of our analysis generalizes the behavior to any RWF fitting certain symmetry / unit integral properties.

We have placed an additional reference to (Courtney, 2014) and to (Clive, 2008) in the discussion of the RWF results (now Section 3.1, LL597-605) and added Fig. 9 which shows the bias due to the RWF bias in the full model coincides with the action of the RWF on just the mean LES profiles.

**Figure 12**: If the aim is to highlight the flow heterogeneity, then instantaneous flow visualizations cannot provide this information. Flow visualizations would work better. On the other hand, if you just want to show the positioning of the Windcube V2 within the domain, this figure works fine.

Referring back to our response to the comment on the use of "homogeneity" (**LL444-445**), Fig. 12 (now Fig. 4), is meant to visualize typical turbulent structures alongside the lidar scan geometry. There shouldn't be meaningful *turbulent* heterogeneities in the test case flow fields (they are designed to be quasi-homogeneous), but the character and scales of the turbulence interact with the lidar scan geometry to produce error. While Fig. 4 does not tell us exhaustively about the error, we think it helps to build intuition about the typical scales with respect to the lidar and the resulting interaction, which is then treated more systematically via the model distribution results.

**L653**: Please provide more details about the trend line.

The trend line is the result of plotting the average of the errors of data binned by wind speed (bins of 0.5 m/s). With closer scrutiny of this plot we have adjusted our statement about the decay of the bias (LL685-692) and removed potentially noisy points in the trend line for bins with fewer than 2500 points. All other things equal, the error expression for the wind speed suggests that a higher wind speed should repress the positive bias ($1/|U\_h|$) but separating the trend by stability (and height, not shown), we do not feel we can discern the expected trend from the other factors changing implicitly with wind speed.

**LL 668-671**: This improvement is not so evident in Figure 15. If you want to highlight it, for instance, you can plot the median Normalized Wind Speed Error (plus-minus 25th percentile) as a function of the misalignment.

Thanks for the feedback on this. Our key point is that we don't believe the orientation has a meaningful effect. We have added mean and standard deviation markings to the plot and amended the statement to consider the changes in the wind speed and direction error negligible with a small visible change in the (wind-direction-weighted) vertical velocity reconstruction.

**L681**: The term "noise" here is misleading. It can be replaced, for example, with "scattering" or "variability".

We replaced the use of "noise" with "variability" here. (Now in section 2.3.5)

**LL 702-704**: This sentence is unclear. It might not be correct to mention "mean error" referred to individual measurements. Further, it is unclear what the authors mean by saying "the time average approaches the potentially less biased 'ensemble' error from the selective sub- sample". Please clarify or rephrase it.

"The mean error in an individual measurement may benefit from time averaging, however, due to the time average approaches the potentially less biased ensemble error from the selective sub-sample (e.g. Fig. 7)."

Thanks for pointing this out. We agree about the "mean error" being a poor/incorrect way of referring to the error in this case. We wanted to make the distinction that the "mean" of measurements in the time series from a single instrument, i.e. the time-averaged measurement, can have an (unsystematic) bias and that

this "mean" improves with longer time averaging. This behavior really reflected by the variance of the time-averaged measurement error and this sentence only confuses the issue and we have dropped it.

**L705**: To my understanding, here you refer to the variance (i.e. the uncertainty) associated with the mean of the random process, which is a function both of the variability of the process and the number of samples. Please clarify this aspect.

(Now in Section 2.3.5 discussing the analysis of time-averaging effect. LL474-487)

Yes; the error spread is referring to the variance of the error which was discussed in earlier sections (and we have reworded to call it so explicitly).

This is exactly the analysis of the (time) mean of a random process. The 1-Hz measurement of the winds (the random process) is associated with a random variability/distribution connected to the character of the turbulent variability in the flow. The reduction in the variance of the mean of a random process is dependent on (inversely proportional to) the number of samples, N; however, the correlations in the time series lower the effective sample size. We have introduced (Lumley and Panofsky, 1964) to help cut more directly to our point about the expected scaling of the variance in a time-average.

**L708-709**: This observation is qualitative and potentially not true. Please either remove this sentence or provide references.

(Now in Section 2.3.5.)

"In the lidar, the pattern of turbulent structures that gave rise to a particular error continue to influence the error in the following samples as well so that the errors are quite similar."

We agree that the wording of this statement is perhaps too strong and depends on implicit assumptions about the turbulent scales, speed of the scanning cycle, and decorrelation times which may not always hold. Our purpose here was to make clear why the errors in the time-series cannot be assumed to be independent samples. We have deleted this sentence and simply left it at the preceding sentence, "Samples cannot simply be treated as independent since subsequent samples can be highly correlated." (L477-478)

**L717**: This assumption is correct, but it relies entirely upon the estimate of the decorrelation time $\tau c$. Please provide details about its calculation. Or, as an alternative, if you are interested in a more precise estimate of the standard deviation error, a good reference is Benedict & Gould (1996).

(Now Section 3.3, time averaging results.)

Thanks for the input here. The exact decorrelation time $\tau c$ was not intended to be a focal point but rather as a more conceptual step to reach the conclusion about the rate of reduction for the error variance/standard deviation. Upon reflection, we have decided to replace Fig. 16 (now Fig. 15) with a plot that we think better illustrates the trends we wanted to convey (i.e. the rate of the reduction in the standard deviation and decay of the wind speed over-estimate bias) and which does not depend on an estimate of the decorrelation time. Showing the reductions over time directly also allows the decorrelation rate to appear implicitly in its effect on the speed of the decay in the different stability cases.

(And thanks for the reference; we regret we won't have time to incorporate it into this work, but appreciate the recommendation.)

**L739**: What do you mean with "background signal"? Do you refer to the time or space average?

Yes; we're referring here to the time- and scan-volume-averaged vertical velocity. We've replaced "background signal" with "background, spatio-temporal average" to be clear. (L803)

**L900**: This equality is unclear. Please clarify and/or rephrase it.

We have reworked this derivation (Appendix B) to make the behavior and assumptions more explicit. We had originally immediately dropped the covariance term on the assumption that $Vu_{err} - Uv_{err}$ and $|\vec{U}_h|^2 + Uu_{err} + Vv_{err}$ were uncorrelated based on the reasoning that the error vector direction and magnitude are fairly uniformly distributed about the circle. We have kept the covariance term in the reworked derivation to treat it explicitly and called out the underlying assumption. We have also introduced a reference (Kendall, 1994) to skip directly to the approximated form for the expectation of a ratio instead of performing the Taylor series expansion ourselves, which keeps the focus instead on the expected behaviors of the numerator and denominator.

**Technical comments:**

**L15:** Which velocity components are named $u$, $v$? Changed to 'horizontal velocities' here since we do not meaningfully distinguish between the two.

**L21:** Please add the meaning of the acronym lidar. Done.

**L31:** Please state clearly that the measured velocity is the along-beam, i.e. radial or line-of-sight, component. Added.

**L105**: Also mention the conclusion reported in Sec. 5. Added.

**L114**: For the sake of clarity, please put table 2 at this point of the discussion. Moved up to be introduced with the RWF.

**L150**: Please mention that $\Delta p$ is the lidar range gate. Included before its first use in Eq. 4. (L158)

**LL194-195**: This sentence is unclear, please rephrase it. On recommendation of the other reviewer, we have removed the paragraph with this sentence as scanning lidar instruments are not directly relevant to the current study.

**Table 2**: Please add one row reporting the azimuth angles used by the DBS scan. Added.

**L297**: Please revise this sentence as it sounds unclear.

"The error incurred in any individual measurement does not necessarily represent general behaviors; deducing useful, generalizable trends entails focusing instead on distributions of the observation error."

changed to

"The error incurred in any individual measurement depends on the specific realization of turbulence during the measurement and is not necessarily representative of the full variability of possible error behavior. To deduce useful information about bias and typical error magnitudes that can be generalized to other measurements in the same conditions, we focus instead on the distribution of the observation error." (L497)

**Figure 5**: In the caption, instead of "dotted", you should refer to solid lines to indicate the volume-averaged reference. Thanks for catching this — reference to the solid and dashed lines is fixed and a legend added.

**L358**: Please add a reference to figure 6 here. This paragraph/sentence was broken up in the rearrangement of the manuscript.

**LL391-392**: To my understanding, the error metric is the same both for the pointwise and the volume-averaged reference. It would be more correct to say "[...] merging of the two error distribution profiles". We referred to the metrics of the distribution moments (mean/variance), but your wording is clearer and we have adopted it. (L731)

**L451**: Please provide a reference.

For mean and variance of linear combinations of random variables, we have added a reference:

Zwillinger, D. and Kokoska, S.: CRC Standard Probability and Statistics Tables and Formulae, Chapman & Hall/CRC, Boca Raton, Fla, 2000.

**L559**: I guess here you refer to figure 12.  Rearranged.

**L644**: Please revise this sentence as it sounds unclear.

(Now in section 3.2)

"For measurements made in relatively steady, slow winds it cannot be expected that bias will not emerge as it does in the strong CBL data."

changed to

"Measurements made in conditions of slow winds of fairly consistent direction, as in the strong CBL case, do not benefit from the cancellation expected in an ensemble over all instrument orientations and should take into account the possibility of a persistent bias arising in the wind direction." (LL676-679)

**LL659-661**: This sentence is unclear, please rephrase it.

"The additional decay is likely accountable to the inverse tangent, which curtails the size of the largest errors compared to the bounding estimate as well as improved correlation at the heights with the strongest winds reducing the component errors."

changed to

"In some cases, the decay in the error magnitudes is greater than the anticipated $1/U\_h$ bound. This may be in part because the inverse tangent in the full error expression (Eq. 21) should act to further curtail the size of the largest errors more than is captured in the bounding estimate (Eq. 22) and in part because of implicit correlation effects with height and wind speed." (LL696-698)

**L706**: Please replace "variables" with "samples".

The technical meaning of random variable was meant here; we have fixed "variable" to "random variable".

**L716**: Please provide any reference or derivation for this equation. Also, provide a brief description of the terms reported here, which have not been introduced before.

We have replaced our cursory discussion here reaching this inequality with a reference that cuts straight to the variance decay rate behavior (Lumley and Panofsky, 1964). We have fixed the inconsistent notation for the variance of the u velocity error and the remaining terms in the updated equation are defined in the preceding text.

**L821**: This study offers a relevant comparison for the present work, so it should be discussed at the beginning of the previous Section.

We have incorporated this reference (Teschke and Lehmann, 2017), which analytically treats the minimization of the error and bias with respect to the beam angle under simplifying assumptions, into the initial discussion of the beam angle. (L357)

L**884**: You can replace this Appendix with simple literature references you deem pertinent.

Good point. We have replaced the appendix entry with a general reference for the moments and unbiased estimators:

Zwillinger, D. and Kokoska, S.: CRC Standard Probability and Statistics Tables and Formulae, Chapman & Hall/CRC, Boca Raton, Fla, 2000.

and a reference for the adjusted Fisher-Pearson skewness and kurtosis coefficients used:

Joanes, D. N. and Gill, C. A.: Comparing Measures of Sample Skewness and Kurtosis, Journal of the Royal Statistical Society. Series D (The Statistician), 47, 183–189, 1998.

**L897**: Do you refer to Equation 22 here? If so, please state it clearly. Yes, fixed.

**References:**

Benedict, L. H., and R. D. Gould. (1996). Towards better uncertainty estimates for turbulence statistics. *Experiments in Fluids,* 22(2), 129-136.

Frehlich, R., & Cornman, L. (2002). Estimating spatial velocity statistics with coherent Doppler lidar. *Journal of Atmospheric and Oceanic Technology*, *19*(3), 355–366.

---

## Author Comment (AC4)

**Response to Additional Comments from Dr. Black**

Additional comments on *Behavior and Mechanisms of Doppler Wind Lidar Error in Varying Stability Regimes*: Clarification of WindCube v2 wind speed computation.

Andrew Black
April 28, 2022

We greatly appreciate Dr. Black's thoughtful and careful reading of our work and his efforts to make our results as topical and current as possible.

*Section 3.2 Ten-minute time averaged velocities*

The WindCube v2 10-minute average wind speed reported by the device in the STA data is **a scalar average** of the 1 Hz horizontal wind speeds:

The WindCube v2.1 uses a weighted linear combination of scalar and vector averaging:

This is to point out that there **are not any WindCubes (except a few early v2.1's) reporting pure 10-minute vector averages in the STA data, as in your paper.** The vector-averaged data is available today in VSTA files on board the device, and can be computed from the 1 Hz LOS RTD data, but typical uses use the average wind speeds in the STA files.

Thanks for pointing out the form of the internally-reported averaging. We have added these points about the instrument-reported averages to this section. (L539)

Vector lidar and pointwise measurements (implicitly mimicking a sonic anemometer) or full vector wind field averages are shown (dashed lines, Figure 8, et al). Are the *pointwise* 10-minute averages computed using vector-averaged u, v, and w components? I assume yes, but this should be clarified.

Yes; all (lidar, volume average, and and pointwise) time averages were computed as vector averages. We've updated the notation to be explicit (Eq. 26,27,33) since we've introduced the use of scalar averages in expanding the analysis.

It would be very interesting to compare the 10-minute scalar- and vector-averaged pointwise measurements to 10-minute scalar- and vector-averaged lidar measurements. Here's why:

In the Rosenbusch et al (2021) article, the differences between scalar and vector wind field reconstruction for pulsed DBS lidars (with WindCube scan geometry and timing) when compared to scalar-averaged cup anemometry were shown, theoretically, to depend on the correlation between *u, v,* and *w* turbulent components, thus likely strongly influenced by stability, just as you've divided your data. The data used in Rosenbusch was restricted to comparisons between scalar *cup* averages and scalar and vector lidar averages. The theory developed in Rosenbusch et al implies that 10-minute vector averages of pulsed DBS lidars with WindCube scan geometry should not exhibit systematic, WFR- caused biases when compared to vector-averaged pointwise measurements (though RWF biases may exist, as you observe).

From what we understand, the theory in Rosenbusch et al (2021) implying the 10-minute vector averages of the lidar and pointwise measurements experience no bias is based on the idea that the flow is homogeneous and the velocity component perturbations vanish entirely in the average. Under the vector-average, then, both measurements represent the Reynolds averaged wind velocity. This is a good approximation under some conditions with small turbulent structures, but based on our results may not necessarily hold in other conditions.

In our analysis and model results, we expect the positive bias in a wind speed measurement to decay like $T^{-1}$ over a vector time-average, so that it diminishes rapidly to negligible values in conditions like the stable

BL or even the weak CBL (Fig. 14). In the strong CBL, however, the initial bias was large and the coherent structures and slower winds lead to longer correlation times, making the decay of the bias slower. Especially for the top range gates in the strong CBL, we still see evidence of non-negligible bias (0.1-0.2 m/s) in the vector average after 10-minutes (Fig. 14). A longer average would be needed in these conditions to drive the bias down to negligible levels as suggested by the treatment in Rosenbusch et al (2021).

We've noted this in our analysis on the topic and the consequences of the behavior for the hybrid weighting scheme (Section 3.3, LL785-799).

On the other hand, it shows that the 10-minute scalar-averaged lidar measurements should exhibit systematic high biases when compared to scalar-averaged pointwise measurements, and that this bias should vary in different stability regimes. As in your paper, this is shown in propagation of the turbulent decomposition through the WFR algorithm developed by Jennifer Newman. **This contradicts (or at least restricts to vector WFR) an observation in your paper:**

> Line 545-546 : *"For the most part, the error mean biases can be attributed to RWF effects and the velocity perturbation terms do have close to zero mean, but important deviations from that assumption do arise"*

The scalar WFR case should show systematic biases due to the velocity perturbation term, especially in the convective cases.

Yes; this statement was meant to refer to the wind component errors (which we've clarified in the reorganized text). We noted the systematic bias in the 1-Hz wind speed error, along with the diminishment of the bias under the ten-minute vector time-average. The bias in the 1-Hz error is perpetuated under the linear scalar-average.

Your dataset is ready to make these scalar-to-vector, and scalar-to-scalar comparisons between the lidar and pointwise measurement. I think it would be a valuable addition to the paper. Adding the scalar- averaged pointwise ("cup-like") measurements would expand the scope of your results to another sensor type and constitute a more comprehensive first result using this simulation data. I believe it would increase the impact, as well, due to the ubiquity of cup anemometry in wind energy. Treatment of uncertainties for cup anemometry (and for lidar) is covered in multiple IEC standards (61400-12-1, - 15-1, -15-2, 50-3, 50-4, et al). This topic of sensor uncertainty and error is of great importance for the wind energy industry, and I think your simulation framework is a breakthrough.

One last thought is that adding the 10-minute scalar averages would also allow for direct propagation of the 1 Hz errors to 10-minutes through the scalar averaging equation, a way to connect those two sections of the paper more strongly. This would require an interesting treatment of the covariance between the neighboring 1 Hz measurements, which share 1, 2, or 3 LOS measurements, essentially the covariance of a moving average (and not only the wind itself).

Best regards,

Andrew

Thank you for these suggestions and insights about expanding our analysis and the motivation and interest behind particular comparisons. We were gratified to hear that our data presents an opportunity to dig into some of these dynamics and we have expanded our analysis to address comparison to 'cup' (scalar-averaged, pointwise) wind speeds.

In the rearranged manuscript, these additions appear in the time-averaging section of the analytic model error (1.3.5) and in the time-averaging results section (3.3).

A full-scale analysis of the covariances could likely make up its own paper so we've had to stick to preliminaries. We hope that the availability of our data can facilitate future investigations and analysis that are out of scope for the present paper.

---

## Author Comment (AC5)

**General comments from the author on the revision:**

We would like to thank the reviewers for their thoughtful and thorough responses and comments which have helped to strengthen the manuscript. In the process of revising and in light of some of the reviewers' points, we have made some additional edits to clarify and clean up the manuscript.

- In responding to the review comments, the organization and ordering of the material emerged as an issue. We have sought to address this by moving the analytic treatment of the error distributions (which appeared along with the results in Section 3 describing the underlying mechanisms) to a new subsection in Section 2 (Data and Methods). The restructuring required some large changes in the manuscript to accommodate the new ordering, combining repeated or dispersed statements into the new sections, and to help and with the flow. New or modified text has been added, but we have tried to avoid changing or introducing new technical material in the rearrangement.
  - Fig. 1-3 are the same
  - Fig. 4 moved (previously was Fig. 12)
  - Fig. 5 is a new diagram by reviewer request
  - Fig. 6 moved (previously was Fig. 4)
  - Fig. 7 is "new", summary of u and v moments from appendix by request
  - Fig. 8 combines the error decomposition from previous Fig. 9, 10, 11
  - Fig. 9 is new, compares RWF acting on mean profile to bias in full model (to address reviewer questions)
  - Fig. 10 is moved (previously was Fig. 13)
  - Fig. 11 is moved (previously was Fig. 5)
  - Fig. 12 is moved (previously was Fig. 14)
  - Fig. 13 is moved (previously was Fig. 15)
  - Fig. 14 is moved (previously was Fig. 8)
  - Fig. 15 replaces what was Fig. 16, highlights trends in time-averaging demonstrates error decay rates instead of showing reduction just in a 10-minute average (thank you for reviewer feedback)
  - Fig. 16 is new, part of the added analysis of comparison to cup measurements
  - Fig. 17 is new, also part of the added analysis
  - Fig. 18 combines plots of the vertical velocity reconstruction error (prev Fig. 6,17)
  - Fig. 19 was previously Fig. 19
  - (previous Fig. 7, showing the KDEs of a single time-series measurement to illustrate the selective sampling from the ensemble distribution was removed as a lower-priority figure for space and flow reasons)
- A community comment suggested additional analysis in comparing vector- and scalar-averaged lidar and point measurements. The bulk of the additions regarding this analysis have been made to the time-averaging theory (Section 2.3.5) and time-averaging results (Section 3.3, Fig. 16 and 17) to provide the background and present the results.
- In the derived wind speed error form, we have replaced our first-order expansion with an expansion using second order Taylor expansions in u and v to put it in the same form as mathematical analogs in existing literature (e.g. Courtney 2014 and Rosenbusch 2021). The change of form makes the correspondence with previous work clearer, especially when summarizing and discussing the comparison of the lidar and cup scalar averages of the wind speed. The analysis and conclusions from the original form hold equally for the alternative expansion.
- It came to our attention while clarifying some of the terminology based on reviewer comments that the mathematical notation could also be adjusted to support important distinctions. We have adjusted the notation to try to make each of the mathematical objects clear and sought to make the notational forms consistent.
  - We have changed the radial wind measured by the lidar at the beams from $u_E$ to $v_{r,E}$ to make the relationship to $v_r$ used for the radial velocity in the RWF model clearer and to avoid any confusion with the velocity perturbations at the beams (was $u_E'$)
  - We have made a notational distinction between the perturbations from the volume-averaged wind (over the lidar scan volume) at each of the beam locations ($u_E'' = u(east\ beam) - U$) (now using $u_E''$ instead of $u_E'$ ) turbulent fluctuations from a Reynolds average ($u' = u(x,t) - \bar{u}$). This supports our discussion of the ways the behavior can differ from Reynolds turbulent fluctuations when using a local volume-average, the distinction in the filtering of the turbulent velocity variances, and in the added material on the comparison of scalar wind speed averages.

- The decomposition of the vertical velocity error (w_err) was plotted against the wind-direction-corrected full model rather than the V1, equally weighted model. This has been corrected and changes the solid, 'full model' error lines in the panels for the vertical velocity in the decomposition in Fig. 8. (was Fig. 9,10,11)
- Correction of small typos, edits to improve clarity and flow, and bringing the text into compliance with journal standards (use of Fig., Eq., and hyphenation in particular)
- Figures have been regenerated as pdfs where possible. Panel labeling, legends, and use of mathematical terms in axis labels have been adjusted in some figures for consistency across the manuscript.

**Response to Referee #1**

**Comment on amt-2022-73**

General comments:

This is an interesting study that tackles a fundamental problem for the Doppler wind lidar observations which is the better understanding of the measurements' error. The authors used large-eddy simulations for different atmospheric stability conditions and analyzed which mechanisms drive the errors under each condition. This study can be very useful for researchers studying turbulence as observed by a Doppler wind lidar. It could help them understand possible error biases or the cause of large errors. It also shows the importance of multiple scans so that the observations can be averaged over time to decrease the error. I believe this study is worth publishing. Nevertheless, the authors should make several changes in the manuscript in order to improve it's structure. In the current state, it is difficult to read through. Introduction, methodology, results and conclusions are mixed, the authors should set apart these chapters to improve the flow of the manuscript. In the current version, parts of the introduction are presented in the methodology, whereas parts of the methodology are introduced in the results and some results are presented but discussed in a later section. In the conclusions, a summary of the work with the main points of the results should be presented, however the authors make a comparison of their results with previous studies and furthermore they introduce new references. Regarding the abstract, the results should be supported by numbers so it will be evident that these are the results of the current study. Some terms should be explained better throughout the study, so it is clear to the reader what is the magnitude of terms such as the "strong winds" or the "large structures". The figures and their captions need corrections as well. All the important information should be included in the plot with the description written in the caption, rather than important information, such as the colour of the lines, only explained in the caption. The language in the manuscript is fluent. Please find my comments below.

We would like to thank the reviewer for their time reviewing the manuscript and thoughtful comments and appreciate the acknowledgement of the applicability of our findings for researchers working with Doppler wind lidar. The insights into how the material can be better presented were particularly useful.

**Scientific comments:**

Page 2 Line 27-29: The authors mention that lidar data offer an indirect representation of the flow field. Although it is true that the wind lidar observations most likely need extra steps to extract the wind components compared to sonic anemometers, it is possible to directly observe the wind components if the beams are aligned with the wind direction e.g. horizontal beams with no elevation angle alongside the wind direction or vertical beams for the vertical wind.

Measuring the along-beam component is more direct than the reconstruction required by a profiling lidar, but we would still view it as less direct than e.g. a 'point' measurement made by sonic anemometers. As modeled by the RWF the probe length of the lidar sampling process is 10s of meters long vs decimeters and can interact with shear in the flow.

P2 L45-47: Similarly to my previous comment. In the phrase "Questions about measurements of vertical velocities"; do the authors mean vertical velocities variances? Because vertical velocity can be directly measured by the Doppler wind lidar and in fact even the vertical velocity variance is the "easiest" turbulence parameter that can be estimated using wind lidar observations, see Bonin et al. 2016: "Improvement of vertical velocity statistics measured by a Doppler lidar through comparison with sonic anemometer observations".

"Questions about measurements of vertical velocities and the ability of wind profiling lidar to measure turbulence remain areas of active research (Sathe et al., 2011; Sathe and Mann, 2012; Sathe et al., 2015; Newman et al., 2016)."

We referred to the way the volume-averaging can complicate even the 'direct' measurement of the vertical velocity and the variance, which is addressed in the Bonin et al. 2016 paper. We have amended the statement to reflect the particular challenges in the turbulence measurements and included the paper as an additional reference.

(LL49-50) "Questions about the ability of wind profiling lidar to measure turbulence remain an active area of research (Sathe et al., 2011; Sathe and Mann, 2012; Sathe et al., 2015; Newman et al., 2016; Bonin et al., 2016)."

P2-3 L59-62: The sentence "Compared to field studies of instrument accuracy, studies with virtual instruments in LES ...." should be supported by some relevant references for such studies.

We have amended this sentence to acknowledge studies that have leveraged the unique opportunities presented by a virtual instrument.

(LL62-66) "Compared to field studies of instrument accuracy, studies with virtual instruments in LES have unencumbered access to full knowledge of the flow field. This knowledge enables control over the case parameters (terrain, forcing, boundaries), and so users can "deploy" instruments in ways that may not be physically or financially possible in reality (e.g. re-sampling the same flow field or testing many locations in a domain) (Muschinski et al.,1999; Stawiarski et al., 2013; Wainwright et al., 2014; Gasch et al., 2019).

P3 L67: The PALM model was developed by Raasch and Schroter, 2001: "A large-eddy simulation model performing on massively parallel computers". The reference should be added in this sentence. Added here, thanks!

P4 L78-79: The Skamarock, 2008: "A description of the Advanced Research WRF version" reference should be included in this sentence. Added.

P4 L82: The Chow et al., 2005 is not relevant here as they do not mention the WRF model in their study.

We have corrected this citation to Kirkil (2012).

P10-11 L248-270: The authors state that they selected cases with "strong" and "weak" convective boundary layers. In line 262, these cases are characterized well-mixed layers. However, in Figure Table 1 we can see that the for the weak cbl the abl height is 525 m. This value seems to correspond to a developing boundary layer and not a well mixed. The authors should comment on that and whether these values can occur only in an ideal case of the simulation with a flat, homogeneous terrain etc. Moreover, the vertical range of the Windcube is portrayed in Figure 3 but it is not mentioned in the text and even in Table 2 it is not directly shown. This information should be included along with an explanation of this selection. The value is higher than the ABL height under stable conditions (170 m). Wouldn't this affect the comparison? See also my comment regarding Figure 4.

Regarding the "weak" convective boundary layer: The convective cases drawn from Rybchuk (2021) are emulate conditions of the Project Prairie Grass campaign (Barad, 1958) which looked at diffusion of tracer gasses. The ABL height in the weak CBL case can and did occur in real conditions. The depth is a function not just of the surface forcing but also of the strength of the inversion aloft which allows for the 525m ABL to occur in a well-developed boundary layer. We have noted in the text that these cases reflect real, observed conditions and added a reference to the campaign along with the existing reference to Rybchuk (2021).

Regarding the vertical range of the Windcube: We have added a line to Table 2 (now Table 1) to make the range explicit (40-240m AGL). The range arises from values fixed by the instrument and typical operating parameters, i.e. the number and vertical spacing of the range gates. We have added this explanation to the text (LL201-203). The fact that the range exceeds the ABL height under stable conditions does affect the comparison. We refer to the boundary layer structure as part of the discussion of the error behavior (e.g. the peaking in the perturbation variances mid-boundary layer and tapering aloft in the stable BL while the

growing variances in the convective BLs only reach mid-boundary layer at the top of the range) but did not see fit to normalize by the boundary layer height since the geometry of the instrument is fixed.

P11 L266-267: The authors mention the limitation of the lidar range to include the entrainment zone. For the instrument, it is practically difficult to measure at this height due to the scarcity of aerosol in the zone. Do the authors refer to the instrument or the simulations? What are the limitations for a simulation?

Practical considerations like sufficient aerosols were outside the scope of our analysis, though important to actual operations. There is no reason the simulation / virtual lidar model couldn't be extended to higher ranges and the range limitation is introduced based on general Windcube instrument parameters (which do depend on considerations of aerosols and obtaining a usable signal) of the number and spacing of the range gates. Since the instrument geometry is more or less fixed for deployments across conditions, we have mimicked that geometry and range throughout the test cases (while ignoring, e.g. the possibility that a real instrument would have insufficient aerosol scattering to return a usable signal within that range).

P14: In Figure 4 panel (c), for the lower altitudes the median of the curve is not at zero as it also stated by the authors in P15 L356. The authors should explain at this point of the manuscript why this occur only at these particular levels of the SBL. It is also apparent from panel (c) that for the levels above the ABL height (170 m) the distribution becomes similar to the one near the surface. Any comments regarding this? Was this something the authors expected?

P15 L355: What could be the cause for overestimation during convective conditions?

For both these points, using the suggested re-ordering of the material to introduce the analytic error model in Section 2 allows the mechanisms behind these behaviors (RWF acting on the shear / vertical wind profile and the systematic positive bias term in the wind speed error) to be addressed as soon as they are shown.

The (moved-up) diagnosis of the causes of bias and variance in the error explains why the error above the stable ABL resembles that at the surface. The filtered turbulence under the scan-volume scale is similarly smaller at the surface and above the ABL (particularly the vertical velocity variances) which reduces the width of the distribution. Similar curvature in the background u velocity profile induces the negative biases in the shear layer in the surface but also aloft through the RWF.

A positive bias term (proportional the u,v variances / inversely proportional to the mean wind speed) gives rise to the overestimation in convective conditions when vertical velocity variances drive larger errors.

In the text, the discussion of these points and the connection of the error behavior to the vertical structure of the boundary layer turbulence (which were split between Sections 2 and 3 in the original manuscript) have been merged into a single, streamlined results section (Section 3).

P24 L506-507: The terms strong winds and strong shear are vague. Can you quantify these parameters? Is the underestimation expected above a specific threshold? Additionally, the argument that surface shear is one of the cause of the underestimation should be supported by some results in the form of Figures. Maybe the authors could add a secondary y axis with the wind speed and wind shear values at the different given heights in Figures 9, 10, 11. The estimation of the wind shear can be tricky but this claim should be supported by results.

"The most prominent influence of the RWF is near the surface layer (due to strong shear) and manifests as a negative shift in the mean error in the horizontal velocities. The weak CBL and stable BL, with strong winds and surface shear, exhibit the effect most strongly. "

(This material has been moved to Section 3.1 discussing the source of biases in the results. LL597-605)

The connection to the shear is mechanistically supported by our derivation of the bound (Eq. 16 in the edited manuscript) as well as previous studies (Clive, 2008; Courtney et al., 2014) that have derived the action of an RWF on idealized shear profiles. (These have been included in the discussion.)

The underestimation will occur for any concave function (like the semi-empirical log shear profile for U(z)); there is no threshold except what degree of bias is considered tolerable. The strong CBL, for example, does still have some shear near the surface but the resulting negative bias is <0.05 m/s, so would likely be considered negligible.

We have added Fig. 9 to help support the conclusion in the case-study results. Fig. 9 demonstrates that the bias due to the RWF in the full model and the bias resulting from applying the RWF to just the mean LES profile and shows the largest magnitudes in the bias in u (b) and v (d) occur in regions with the largest (nonlinear) shear.

P26 L540 & 548-550: Similarly to my previous comment, the term "large coherent structures" is vague. It is evident from Figure 12a that there are structures, upward motions followed by downward motions, of approximately 1 km size. On the contrary, for the stable boundary layer (Figure 12b) the size of the structures seems to be equal to few hundred meters. The lifecycle of such structures should be different. Do the authors categorize both structures' sizes as large? The authors also mention larger scale structures above the boundary layer in the SBL. Do they mean from 170 m up to 350 m that is shown in Figure 12b or in higher altitudes? Either way it should be clear to the reader what are the sizes of the structures and these should correspond to the figures presented in the manuscript.

"Without dissecting the mechanics more closely, we simply note that large coherent structure like turbulent plumes characteristic of the convective boundary layer can induce repeated, non-symmetric patterns in the relative perturbations of the beams, leading to non-zero means."

(This material has been moved to Section 3.1, LL590-596)

This is an important point about the size of the structures, which Fig 12 (now Fig. 4) was meant to help convey. The critical scale here is the size of the scan volume (which for the WindcubeV2 is ~42m across at the bottom range gate and ~255m across at the top, shown by the black lines in Fig. 4). "Large" coherent structures are meant to refer to features on the order of the scan volume or bigger so that their internal structure influences the sampling by opposing beams, which would include the 1km structures but potentially exclude some of the features in the lower stable BL.

We have explicitly called out "coherent structures large enough to span the scan volume" and denoted the scan radius in the text to help clarify this point.

**Technical comments:**

P1 L1: Lidars instead of lidar. Fixed.

P3: The full name for the abbreviation DBS should not be included in the caption of Figure 1, but rather in P4 L76.

The definition of the full name for the DBS abbreviation has been moved, along with checking the order and use of other abbreviations in the text.

In the first three instances Figure 1 is written with a capital F in the text, whereas in all the other instances figures in the text are written in parenthesis with a minor f. The citation of the figures should be consistent throughout the text.

Thanks for pointing this out. These instances along with all further references to figures and equations were changed to "Fig.", "Figures", and "Eq." to bring the text in line with journal style guidelines.

P3 L72: The full name for the abbreviation SOWFA should be given here. Fixed.

P3 L73: The full name for the abbreviation WRF-LES should be given here. Fixed.

P7 L156: The parameter "c" is defined after the equation 7, although it is also part of the equations 5 and 6. In order to avoid confusion, I suggest to move the definition before or after the equation 5.

Good catch. We have moved the definition in the text up to the first reference to the use of the speed of light before Eq. 4

P7 L161: The sentence "Using parameters .... can be made concrete" needs rephrasing. It is confusing in its' current state.

"Using parameters from the WindcubeV2 (Table 2), the shape and extent of the RWF (Eq. 6) can be made concrete (Fig. 2a)."

reworded to

(L170) "The RWF for the modeled WindcubeV2 (Fig. 2(a)) results from substituting the range gate and pulse parameters (Table 1) into the general, pulsed lidar equation (Eq. 6). The shape of the RWF peaks at the center and tapers symmetrically toward zero to either side."

P7 L167-170: The paragraph "The form of the pulsed lidar .... further distances by a pulsed lidar" is more suitable for the introductory section. The comparison of the RWF between pulsed and continuous lidar seems out of place in the methodology as only the pulsed lidar was used for this study.

We agree that the discussion here became more general than needed to serve the primary focus of the study. We've dropped the paragraph from the text.

P7 L172: Remove the word "found".

We have reworded the end of this sentence from

"…and the projection onto the beam direction found"

to

(L177) "…and then projected onto the beam direction".

Hopefully this makes it scan more cleanly for the reader.

P7 L173: The "Spe" in the parenthesis is a missing reference?

Yes; a reference used to split the scipy function (into triangulation and application of the interpolation weights to optimize repeated calls on the same grid) does not have a complete reference and therefore was mangled. We have moved that reference to the code and retained the complete technical scipy citation here.

P8 L185-188: The paragraph "Interpolation dominates … subsequent developments" comments on the interpolation method and possible improvements in the data and methodology section. This paragraph is more suitable for a section like Conclusions/Discussion.

We have removed this paragraph and made a brief reference to the optimization of the interpolation as a target for future improvements in the conclusion (L948)

P8 L190-199: In the first paragraph of Section 2.1.2 the authors provide some general information regarding the lidar and the different scanning methods such as RHI and PPI. As these methods are not used in the particular study, they should not be mentioned in the data and method section. In my opinion, this paragraph should be removed entirely from the manuscript as it does not provide any valuable information for the study.

We had included this in a general approach to the virtual lidar model, but agree that the additional information in this paragraph is a distraction that doesn't serve the current study. We've cut out this portion of the paragraph and condensed it to simply refer the reader to Clifton (2015).

P10: In Figure 3 the explanation for the different lines (solid, dashed etc) representing the parameters is only included in the captions and not in the figures, hence it is not practical for the reader to study these figures, similarly for Figures 5, 6 etc.

We have reviewed each of the figures in the manuscript with this in mind and have included full legends reflecting each of the parameters used.

P13 L303: The claim that the components of equation 10 are commonly used should be supported by some examples-references.

We have included a reference to Cariou and Boquet (2010); the instrument itself returns the horizontal wind speed and direction and we believe they are fairly ubiquitous. (L414)

P13 L305-308: A figure showcasing the sign convention could be useful for the reader.

We've created a diagram (Fig. 5) to show the conventions used for the wind direction and sign of the direction error.

P14: The authors have introduced several parameters for the wind speed such as ulidar or horizontal wind speed, therefore it should be clear in Figure 4 which one is shown including its' name as defined by the authors.

We have re-labeled the figure axes with both the name and symbolic reference (|U_h|_err) which refers to the error in the magnitude of the horizontal winds (i.e. 'wind speed error'). Hopefully this addresses any possible confusion about the plotted quantity.

P14 L324: Remove "the". Fixed.

P15-P16: The caption of the Figures 5 and 6 mention dashed and dotted lines but only dashed and solid lines are depicted. The figures should be corrected and additionally there should be a legend in the figure with this information.

Thanks for catching this and we apologize for any confusion it may have caused in reviewing the article. The caption has been amended and the figure revised to include the line style distinction in the legend as well.

P15: Figures should be easily readable even when separated from the rest of the text. In Figure 8, the caption is linked to the parameters of Figure 5 which should be corrected. The parameters should be included in the caption and the legend of Figure 8 independently from Figure 5.

We have regenerated the figures to include comprehensive and consistent legends across these figures.

P17 L374: Converge instead of converges. Fixed.

P18 L401: The phrase "their respective height trends are similar to the previous section" should be accompanied by the respective values as a reminder for the reader. Not in the rearranged manuscript. We have tried to make sure to call out values and reiterate the recurring trends when we refer to them in the updated text.

P19-P33: In Chapter 4 the authors introduce several new equations. In my opinion, a manuscript flows better when all the equations and tools are presented in the data and methods chapter and subsequently the results are presented and discussed. So instead of presenting the figures in Chapter 3 and then using the equations to describe the results in Chapter 4, I believe it would be better if all the equations are already presented in Chapter 2 and the discussion of the results is moved to the corresponding figures. For example the explanation of the underestimation of the wind error although mentioned in Chapter 3 is explained much later in Chapter 4. By moving the equations to Chapter 2, the authors will also avoid repeating themselves.

Thank you for your thoughts on how best to organize the material. We had developed the analytic model for the error as a tool for understanding the output of the virtual tool and had thought it could be treated

concurrently with the results in the paper. Seeing the degree to which the analytic development grew and how much we rely on it in explaining the results, we agree that the flow of the article is improved by re-ordering the presentation.

We have moved the development of the analytic model for the lidar error into a new subsection of the Data and Methods (Section 2). The results using the virtual instrument have been condensed into a single section (Section 3) and streamlined, referencing back to the analytic model in Section 2 when needed for the discussion. We hope this removes ambiguity and repetition and makes the paper stronger and more clear to the reader.

P20: The caption of Figure 9 seems more like a part of the manuscript than a caption. It should be rephrased in a way to resemble a caption.

P21-22: It would be easier to interpret the results from Figures 9, 10, 11 if these figure were merged in multiple panels and thus it would be possible to use the same caption for all instead of linking to the caption of Figure 9 which makes the Figures unreadable independently.

To both these points:

We have revised this set of figures into a single, combined decomposition figure (Fig. 8) which uses the same caption. The revised figure shows the covariances and obviates the need for some of the explanatory text in the caption. The specification that only the non-offset lidar are used was moved to the main text.

P29 L616: The word "term" is used two times. Fixed.

P30: The height of the wind speed is not mentioned in Figure 14. It should be included as part of the xaxis title. Figure 14 (now Fig. 12) aggregates data from all heights. Since the original manuscript left room for uncertainty, we have added an explicit statement to the caption to clarify the point.

P32: The legend showcasing the parameters that correspond to the colour lines is missing in Figure 16.

We have replaced this figure to better illustrate the trends over the time-averaging window. The new version (Fig. 15) does include the color-coded legend.

P32: The panels in Figure 16 are not numbered with (a), (b), (c) etc and the authors refer to the different plots as left and right panels. I believe it will be easier to use the numbering. Similarly the Figures 5, 6, 8, 9, 10, 11, 13, 17, 18 and the ones presented in Appendices A, D, E also miss the numbering.

Thank you for pointing out the inconsistency and confusion in (not) labeling the panels in these figures. We've worked through each of the figures to clean them up and add labels in a consistent style where they were missing and have updated the captions to reflect the changes.

P36-39: The comparison of the authors' results with previous studies should be included in the sections of the results and not in the conclusions. The authors should summarize the key points of their study and not include new information in this chapter. Although it is possible to include some previously mentioned references in the conclusions, it is not recommended to introduce new references such as Klaas and Emeis, 2021 and Teschke and Lehmann 2017. These references should be introduced earlier in the manuscript.

The comparisons of how our findings relate to previous studies have been mentioned as part of the results where appropriate  (now Section 3) and moved into a discussion section (Section 4). We have correspondingly shortened the conclusion to summarize our key points.

P38 L840: Add the word of - "The form of our error...." Fixed.

P52 L1117: Remove typo " ".  Fixed.

P52 L1119: Remove typo "&ndash". Fixed.

---

## Referee Report (RR1)

General comments:

The authors have made some significant improvements upon the structure of the manuscript. The different sections are better separated and furthermore they added a new section "Discussion" which seems very helpful to the reader for this kind of manuscript with so much information. The authors have also improved the quality and description of the figures. In my opinion, the "Conclusions" section needs a little more work as the current version feels incomplete. I suggest to include the values for the most important findings of the study in this section. Regarding the scientific context, I do not have any additional comments. It is a study that could be particularly useful for researchers working with wind lidar measurements. Please find my technical comments below.

Technical comments:

P8 Table 1: The second line displays the azimuth angle, however the name of the variable is elevation angle. It should be corrected.

P8 Table 1: Above the table 1, there are two references "Cariou and Boquet (2010); Bodini et al. (2019)", which are not clear to me to which part of the text they refer to.

P8 L196-199: This sentence needs rephrasing as it describes a simple functionality of the lidar in a confusing way. In addition, I am not sure that the term "complex options" needs to be supported by the reference "Clifton et al., 2015".

P9 L205: Maybe "during" is more correct than "through" in this context.

P13 L304: Correct the typo "explicitly".

P14 L314-315: One sentence is written inside a parenthesis "(Different notions…representation to measure.)". If the parenthesis was included in the previous sentence written with fewer words or remain as it is without the brackets, the text would flow better.

P17 L414: Replace "An" with "A".

P17 L414-416: The first two sentences of the paragraph "An common representation...reconstructed velocity components" could be simplified as it is a bit confusing in its' current state.

P22 L539-542: The sentence "Analysis of the limiting...(Rosenbusch et al., 2021)." seems to either miss some information or have some extra words that are not needed. Maybe "when" has to be removed and "whereas" has to be replaced by "on the contrary"?

P27 L619: Is the word "across" necessary in this sentence?

P31 L707-708: The sentence "As in Rahlves...in the stable case, Fig. 13)." likely misses a conjunction to connect the verbs.

P31 L710: Add "to" after "due".

P32 L729: Do you mean "In select flat conditions,…" instead of "In select, flat conditions"?

P36 L790-793: No need to use parenthesis for this text.

P41 Conclusions: In the conclusions, the authors summarize their work without providing the values for the most significant results. I believe that the section with conclusions works better when it can be read as a stand-alone text. In that regard, I recommend to include some values of the results in the conclusions similarly to the abstract.

---

## Author Response (AR2)

**Response to Referee #2**

**Comment on amt-2022-73**

General comments:

The authors have made some significant improvements upon the structure of the manuscript. The different sections are better separated and furthermore they added a new section "Discussion" which seems very helpful to the reader for this kind of manuscript with so much information. The authors have also improved the quality and description of the figures. In my opinion, the "Conclusions" section needs a little more work as the current version feels incomplete. I suggest to include the values for the most important findings of the study in this section. Regarding the scientific context, I do not have any additional comments. It is a study that could be particularly useful for researchers working with wind lidar measurements. Please find my technical comments below.

We are glad to hear the changes have been effective in improving the manuscript and making the material accessible to the reader. We would like to thank the reviewer for their time reviewing the edits and for the additional suggestions.

Technical comments:

P8 Table 1: The second line displays the azimuth angle, however the name of the variable is elevation angle. It should be corrected. Fixed.

P8 Table 1: Above the table 1, there are two references "Cariou and Boquet (2010); Bodini et al. (2019)", which are not clear to me to which part of the text they refer to.

Since citations are given in the text when the Windcube parameters in the table are introduced, we have omitted the references describing the instrument from the table.

P8 L196-199: This sentence needs rephrasing as it describes a simple functionality of the lidar in a confusing way. In addition, I am not sure that the term "complex options" needs to be supported by the reference "Clifton et al., 2015".

We've tried to simplify the idea about the scan being performed quickly. We added "as in e.g. Clifton et al., 2015" to make clear that the reference is included to point the reader to further information about other possible complex scan strategies.

P9 L205: Maybe "during" is more correct than "through" in this context.

Rotating through the scan reads more cleanly to us than rotating during the scan, but we will happily leave it up to the editor if the other wording is better.

P13 L304: Correct the typo "explicitly". Fixed.

P14 L314-315: One sentence is written inside a parenthesis "(Different notions…representation to measure.)". If the parenthesis was included in the previous sentence written with fewer words or remain as it is without the brackets, the text would flow better. We've omitted the parentheses.

P17 L414: Replace "An" with "A". This sentence was reworked.

P17 L414-416: The first two sentences of the paragraph "An common representation...reconstructed velocity components" could be simplified as it is a bit confusing in its current state.

We've reworded these sentences to try to make them simpler and more clear.

P22 L539-542: The sentence "Analysis of the limiting...(Rosenbusch et al., 2021)." seems to either miss some information or have some extra words that are not needed. Maybe "when" has to be removed and "whereas" has to be replaced by "on the contrary"?

We've broken the sentence in two and reworded it to render the inequalities in words as well.

P27 L619: Is the word "across" necessary in this sentence?

We wanted to make clear that the range of lengths referred to the diameter. We've stated that explicitly instead of saying 'across'.

P31 L707-708: The sentence "As in Rahlves...in the stable case, Fig. 13)." likely misses a conjunction to connect the verbs.

We didn't see where a conjunction might be needed but reworded the sentence slightly to hopefully make it read more clearly.

P31 L710: Add "to" after "due". Fixed.

P32 L729: Do you mean "In select flat conditions,..." instead of "In select, flat conditions"? Fixed.

P36 L790-793: No need to use parenthesis for this text. Parentheses have been removed.

P41 Conclusions: In the conclusions, the authors summarize their work without providing the values for the most significant results. I believe that the section with conclusions works better when it can be read as a stand-alone text. In that regard, I recommend to include some values of the results in the conclusions similarly to the abstract.

We have reworked the conclusion (P 41 L 931-949) so that it stands better alone as a summary of the key findings and corresponding values, reflecting the information in the abstract.